**Subject Category:**
Biology (whole organism)

palaeontology/evolution/ecology

dinosaur, Appalachia, Cretaceous, fauna, tyrannosaur, teeth

**Author for correspondence:**
Chase Doran Brownstein
e-mail: chasethedinosaur@gmail.com

# New records of theropods from the latest Cretaceous of New Jersey and the Maastrichtian Appalachian fauna

## Chase Doran Brownstein

Department of Collections and Exhibitions, Stamford Museum and Nature Center, Stamford, CT, USA

CDB, 0000-0003-4514-9565

The faunal changes that occurred in the few million years before the Cretaceous–Palaeogene extinction are of much interest to vertebrate palaeontologists. Western North America preserves arguably the best fossil record from this time, whereas terrestrial vertebrate fossils from the eastern portion of the continent are usually limited to isolated, eroded postcranial remains. Examination of fragmentary specimens from the American east, which was isolated for the majority of the Cretaceous as the landmass Appalachia, is nonetheless important for better understanding dinosaur diversity at the end of the Mesozoic. Here, I report on two theropod teeth from the Mount Laurel Formation, a lower-middle Maastrichtian unit from northeastern North America. One of these preserves in detail the structure of the outer enamel and resembles the dentition of the tyrannosauroid *Dryptosaurus aquilunguis* among latest Cretaceous forms in being heavily mediolaterally compressed and showing many moderately developed enamel crenulations. Along with previously reported tyrannosauroid material from the Mt Laurel and overlying Cretaceous units, this fossil supports the presence of non-tyrannosaurid tyrannosauroids in the Campanian–Maastrichtian of eastern North America and provides evidence for the hypothesis that the area was still home to relictual vertebrates through the end of the Mesozoic. The other tooth is assignable to a dromaeosaurid and represents both the youngest occurrence of a non-avian maniraptoran in eastern North America and the first from the Maastrichtian reported east of the Mississippi. This tooth, which belonged to a 3–4 m dromaeosaurid based on size comparisons with the teeth of taxa for which skeletons are known, increases the diversity of the Maastrichtian dinosaur fauna of Appalachia.

Along with previously reported dromaeosaurid teeth, the Mt Laurel specimen supports the presence of mid-sized to large dromaeosaurids in eastern North America throughout the Cretaceous.

## 1. Introduction

The extinction of the non-avian dinosaurs at the end of the Mesozoic Era is a topic that has continued to intrigue vertebrate palaeontologists (e.g. [1–7]). However, a poor global terrestrial record from the Maastrichtian has hindered attempts to assess the diversity dynamics of important groups like the Dinosauria during this period (e.g. [4,6]). Western North America preserves arguably the most well-characterized vertebrate record from the last 20 Myr of the Mesozoic Era [6], whereas that from the eastern portion of the continent is far more obscure. During the majority of the Late Cretaceous, eastern and western North America were separated, the former existing as a landmass called Appalachia. Appalachian dinosaur faunas included intermediate-grade tyrannosauroids [8,9], basal hadrosaurids and non-hadrosaurid hadrosauroids [10–12], nodosaurids [13,14] and ornithomimosaurs [15–19].

Despite the amount of knowledge of Cretaceous faunal change to be gleaned from the fossil record of Appalachia, the assemblages of this landmass have remained fundamentally understudied since the mid-nineteenth century (e.g. [17,19–21]). The scarcity of terrestrial sedimentary units known from the eastern half of the USA has also contributed to the obscurity of Appalachian faunas compared to western North American ones [8,9,17,20,21]. Only in the past few years have come indications of faunal changes in the latest Cretaceous (late Campanian–Maastrichtian) of the American east, and all from isolated, fragmentary finds. Although a ceratopsian tooth from the uppermost Maastrichtian of Mississippi [22] and possible lambeosaurine bones from the upper Maastrichtian of New Jersey [13] have revealed that faunal exchanges probably occurred between Appalachia and Laramidia following the regression of the Western Interior Seaway in the latest Campanian–earliest Maastrichtian, the timing of these events remains poorly constrained. Other fossils from the Maastrichtian of the American east, such as the holotype of the late-surviving non-tyrannosaurid tyrannosauroid *Dryptosaurus aquilunguis*, suggest the area may have continued to act as a refugium for some vertebrates. Further sampling of enigmatic assemblages from the Maastrichtian, such as those of the eastern USA, is therefore important for understanding faunal change in latest Mesozoic North America.

In the Campanian–Maastrichtian of New Jersey, a set of formations corresponds to a period of transgressions and regressions of the Atlantic Ocean (e.g. [13,23–25]). The majority of these Cretaceous units are known for producing marine vertebrate and invertebrate fossils [13], although some, such as the Woodbury and New Egypt formations, are notable for producing some of the first partial dinosaur skeletons from the Americas (e.g. [13,17,20,26,27]). One of the most fossiliferous of these formations is the Mount Laurel Formation, which is either uppermost Campanian or lowermost Maastrichtian [24] and in New Jersey has produced the remains of several groups of dinosaurs, including hadrosaurids, tyrannosauroids and ornithomimosaurs [13,28]. Because of the sheer diversity of the community represented in the Mt Laurel, the formation serves as a window into Campanian–Maastrichtian eastern North American faunas. However, the terrestrial fossils it produces are often eroded postcranial fragments [28].

Here, I describe two theropod teeth from the Mt Laurel Formation of New Jersey. These include a large tooth assignable to a 6–8 m tyrannosauroid and a smaller, heavily recurved one assignable to a 3–4 m dromaeosaurid. These teeth are among the most diagnostic records of theropods from the Mt Laurel Formation, allowing for a more precise understanding of the faunal composition and ecology of the eastern seaboard during the Maastrichtian, a globally under-sampled time period [4,5].

## 2. Results

### 2.1. Geological setting

Both theropod teeth described here were collected from sediments of the Mount Laurel Formation [13], a marine deposit that represents a regression of the Atlantic Ocean during the Late Cretaceous and is the oldest unit included in the Monmouth Group [13,24]. The tyrannosauroid tooth described here, NJSM GP 12456, was recovered from Big Brook (figure 1a), a highly fossiliferous locality famous for producing an extensive marine fauna [13,17]. At Big Brook, the stratigraphic column is exposed along

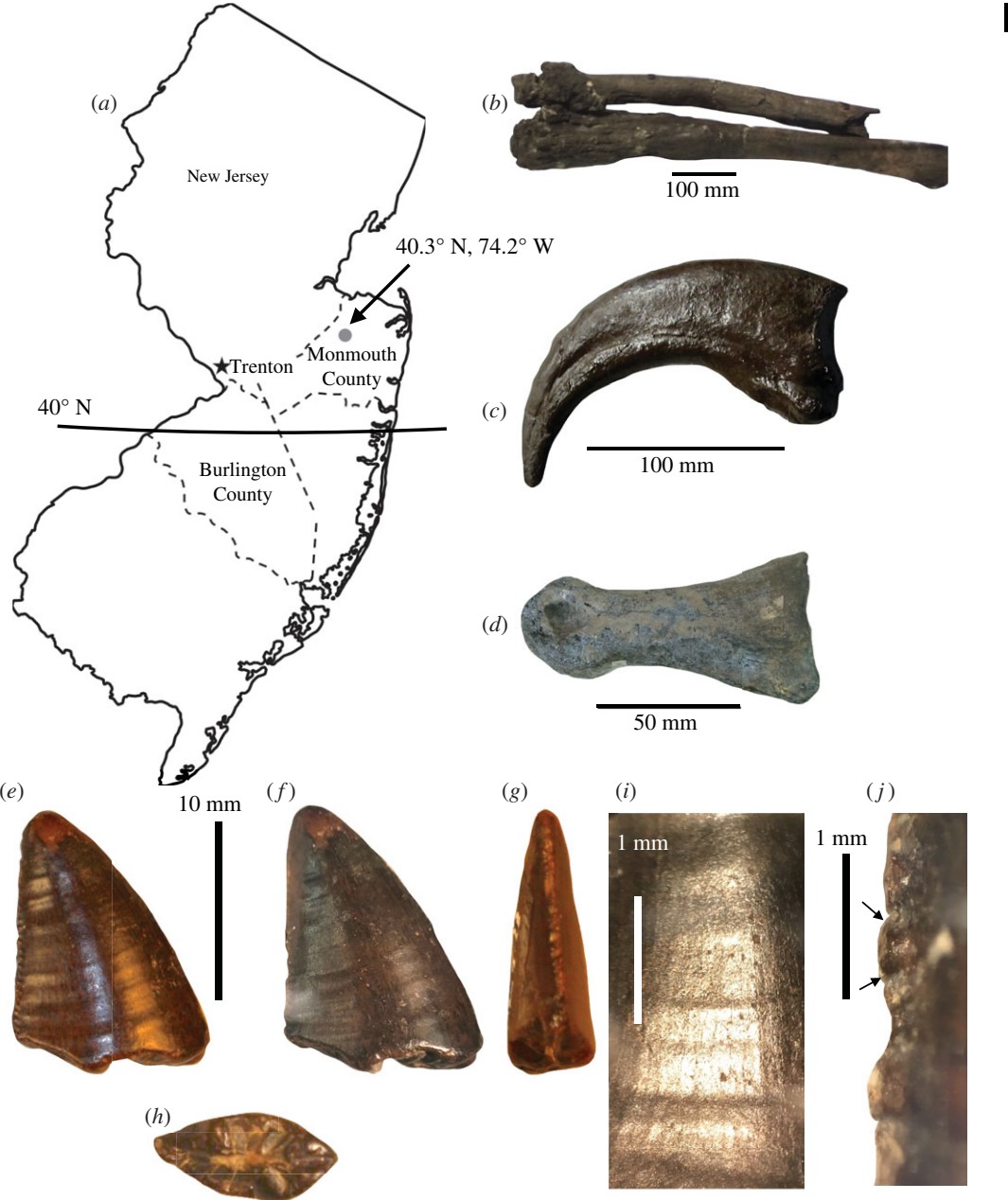

**Figure 1.** Locality information, Maastrichtian eastern North American dinosaurs and Mount Laurel tyrannosauroid tooth anatomy. (*a*) Map of New Jersey showing the location of Burlington and Monmouth counties and the Big Brook site (grey dot), (*b*) possible partial Lambeosaurine forelimb, (*c*) cast of the manual ungual of *Dr. aquilunguis* and (*d*) pedal phalanx of an ornithomimosaur. NJSM GP 14256 in labial (*e*,*f*), distal (*g*) and basal (*h*) views with a close-up of the enamel wrinkles (*i*) and distal denticles (*j*) (arrows show convexity). Map is public domain, access to photograph the cast of the manual ungual of *Dr. aquilunguis* courtesy of the Yale Peabody Museum (peabody.yale.edu).

the banks, with the Wenonah Formation grading into the Mt Laurel Formation such that the boundary between the two is indistinguishable [13]. The contact between the Mt Laurel and the overlying Navesink Formation is an unconformity [24,25]. The Mt Laurel Formation appears as grey to dark brown, pebbly quartz sands. The Big Brook tyrannosauroid tooth (figure 1*e*–*h*) is unusual among the terrestrial vertebrate teeth collected from the site in possessing a well-preserved enamel surface. NJSM GP 12456 preserves both its outermost enamel layer and many of its denticles, whereas other terrestrial vertebrate fossils from Big Brook are known for being heavily water-worn and lacking morphological details.

NJSM GP 22949, the dromaeosaurid tooth, was recovered from Mt Laurel deposits in Burlington County, New Jersey (figure 1*a*). In this area, which makes up a portion of the southwestern-most range of the Monmouth Group, the sands of the Mt Laurel are more glauconitic than farther north and are intermixed with iron compounds [13]. The thickness of this unit is also far greater to the southwest of its range (e.g. [13]).

## 2.2. Tyrannosauroid tooth

Dinosauria Owen 1842
Theropoda Marsh 1881
Coelurosauria von Huene 1921
Tyrannosauroidea Osborn 1905
Tyrannosauroidea indet.

## 2.3. Material

New Jersey State Museum collections (NJSM GP) 14256, the partial tooth of a large theropod dinosaur (figure 1*e*–*h*).

## 2.4. Locality and horizon

Mt Laurel Formation sediments at Big Brook, Monmouth County, New Jersey, latest Campanian to early Maastrichtian [13,24,25].

## 2.5. Identification

NJSM GP 14256 (figure 1*e*–*h*) closely resembles the dentition of tyrannosauroid theropods in several ways. The tooth resembles those of adult tyrannosauroids in its size, which is closely comparable to tyrannosauroid crowns known from both western and eastern North America [9,29–33]. In addition to its size, the Mt Laurel tooth resembles those of tyrannosauroids to the exclusion of other theropods known from Late Cretaceous North America in possessing a combination of packed denticles (2–2.5 mm$^{-1}$) on its distal carina (15+ mm), the presence of denticles along both carinae, its slight, rather than pronounced, curvature, the presence of numerous transverse undulations (density = 2 mm$^{-1}$) on its main surface, the presence of slightly biconvex denticle outlines for denticles all along the tooth (figure 1*j*) and its smooth but slightly irregular surface texture (figure 1*e*–*h*) [9,29–33]. However, despite the size of the Mt Laurel tooth, NJSM GP 14256 is notably unlike the teeth of tyrannosaurids, for which incrassate teeth (basal width to length ratio greater than 0.6) are a synapomorphy (e.g. [33,34]). Instead, NJSM GP 14256 is narrow (CBW/CBL ∼ 0.54) and possesses a lens-shaped basal cross-section, indicative that it came from a tyrannosauroid outside Tyrannosauridae. Among large Late Cretaceous tyrannosauroids, only *Dr. aquilunguis* from the Maastrichtian New Egypt Formation of New Jersey is known to possess a combination of ziphodont dentition and tyrannosaurid-like features of the denticles and tooth surface (e.g. [9]). NJSM GP 14256 is also comparable to the mediolaterally compressed teeth of *Dr. aquilunguis* in its dimensions, curvature and enamel crenulations [9]. Given the Mt Laurel tooth's very close spatio-temporal proximity to the holotype of *Dryptosaurus*, I suggest the tooth belongs to a closely related form.

Given the number of measurements unable to be taken from this tooth due to its incompleteness, I did not include it in a morphometric analysis. However, phylogenetic analysis of the tooth within the dataset of Hendrickx & Mateus [35] found NJSM GP 14256 to be the sister taxon of *Tyrannosaurus rex* in a clade united by four character states. These are characters 94 (biconvex apical denticles present on distal carinae of lateral teeth), 100 (subequal number of denticles apically than at mid-crown portion of distal carinae on lateral teeth), 103 (interdenticular space between mid-crown denticles on distal carinae of lateral teeth broad) and 105 (interdenticular sulci between mid-crown denticles on distal carinae of lateral teeth present, long and well developed) [35]. The clade comprised NJSM GP 14256 and the three other derived tyrannosaurs included in the analysis (*Alioramus*, *Tyrannosaurus* and '*Raptorex*') is united by the presence of a subsymmetric crown with a centrally positioned distal carina in distal view (char. 83). Characters uniting the tyrannosauroid clade include 3, 5, 19, 27, 37, 38, 41 and 48 in the list of Hendrickx & Mateus [35]. The strict consensus topology (tree length = 688, consistency index = 0.290, retention index=0.446) is shown in figure 2.

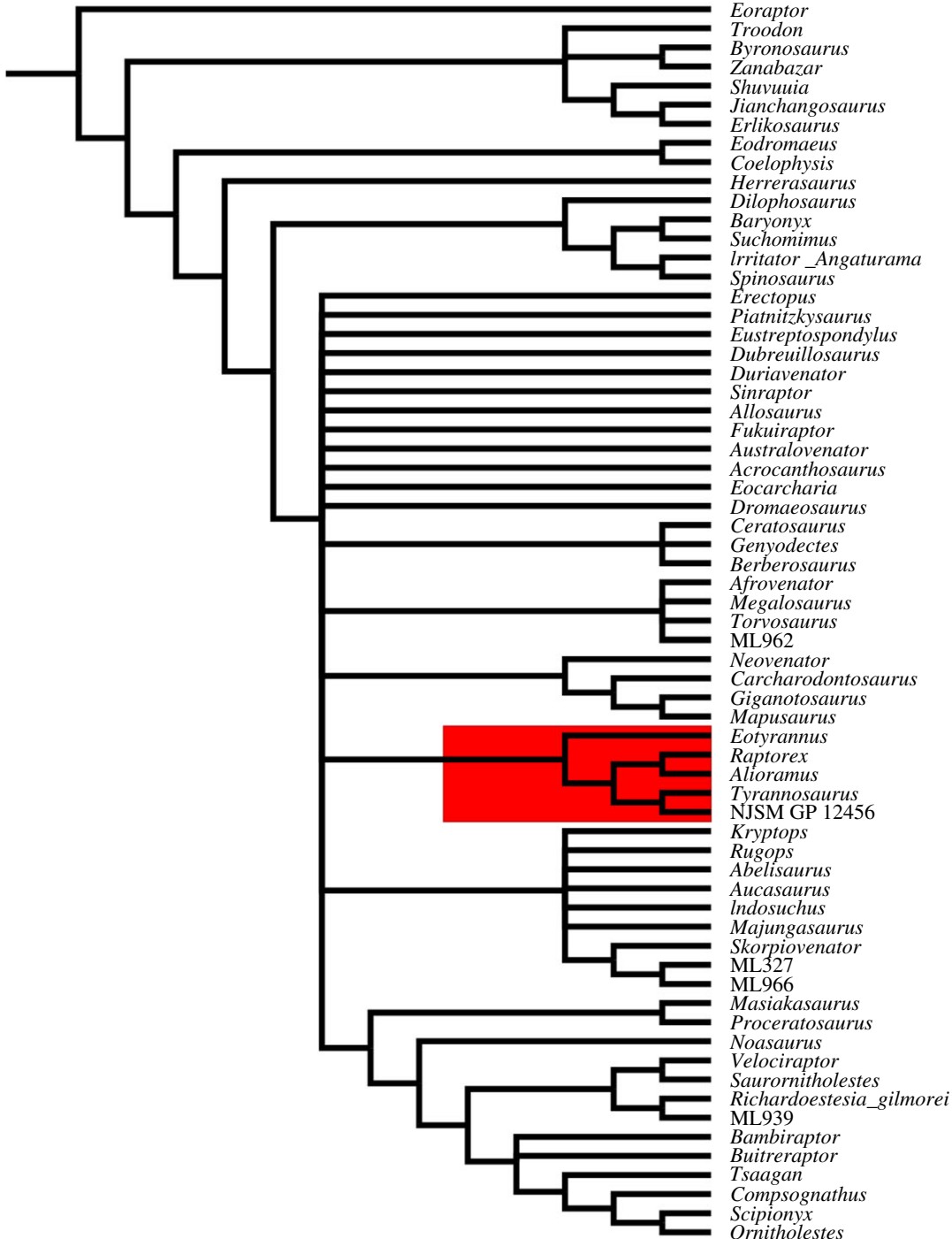

**Figure 2.** Support for the assignment of NJSM GP 14256 in Tyrannosauroidea. Phylogenetic topology of theropod teeth, with Tyrannosauroidea highlighted in red.

## 2.6. Description

NJSM GP 14256 (figure 1) is the apical half of the tooth of a theropod dinosaur. Measurements of the specimen may be found in table 1. The tooth is well preserved for a terrestrial fossil collected from one of the marine deposits of the Cretaceous Atlantic Coastal Plain, preserving details of the outer enamel layer and denticle morphology. Unfortunately, the basal half of the crown and the entirety of the root of the tooth are not preserved. This is probably due to erosion, as the tooth is broken transversely and heavily rounded at its preserved base (figure 1*e,f*).

The tooth displays the ziphodont condition in being labiolingually compressed and only slightly recurved at its tip. The preserved mesial carina is slightly convex, whereas the distal carina is flat.

**Table 1.** Measurements of teeth described in this study (in mm).

| specimen | CH | CBL | CBW | AL | CA | DB | DC | DA |
|---|---|---|---|---|---|---|---|---|
| NJSM GP 14256 | 15 (est. 25) | 8.99 | 4.90 | n.a. | n.a. | 2 | 2 | 2.5 |
| NJSM GP 22949 | 15.5 | 8.0 | 2.0 | 18.00 | 55.5 | 6 | 5 | 5 |

The labial and lingual portions of the enamel are well preserved (figure 1*g*), bearing transverse undulations that develop out of the distal margin of the tooth to become bands [36]. In NJSM GP 14256, these undulations (= marginal bands) are relatively prominent, although they are less prominent than in carcharodontosaurids (e.g. [36,37]). The labial and lingual surfaces of the tooth are slightly convex, as in most other theropod dinosaurs [38]. The apex of the tooth bears a slight wear facet on its lingual surface. The tooth is lenticular in basal cross-section.

The distal carina preserves many denticles (figure 1*e,f,h*), which are small, dense (6 mm$^{-1}$), and apicobasally straightened. The denticles are interspersed with diminutive interdenticular sulci [29]. These are encompassed by the apical ends of the denticles. These denticles maintain a similar density along the entirety of the distal carina. However, their density may have changed along the missing portion of the tooth. The mesial carina preserves a few denticles, although these are too worn for much morphological description. These denticles appear to be similar in size to those on the distal carina.

## 2.7. Dromaeosaurid tooth

Dinosauria Owen 1842
Theropoda Marsh 1881
Coelurosauria von Huene 1921
Maniraptora Gauthier 1986
Dromaeosauridae Matthew & Brown 1922
Saurornitholestinae Sues 1978
cf. Saurornitholestinae indet.

## 2.8. Material

NJSM GP 22949, well-preserved, complete isolated tooth.

## 2.9. Locality and horizon

Mt Laurel Formation sediments in Burlington County, New Jersey, latest Campanian to early Maastrichtian [13,24,25].

## 2.10. Identification

NJSM GP 22949 is identified as the lateral tooth of a dromaeosaurid theropod based on the following combination of features: (i) its extreme apicobasal curvature created by its concave distal carina and an apex that is clearly distally offset from the distal edge of the crown base, (ii) the presence of apically hooked distal denticles, (iii) the absence of mesiodistal constriction along the crown base, and (iv) distal denticles that decrease in size towards the apex of the tooth (figure 3*a–e*) [29,33,39,40]. NJSM GP 22949 is smaller than the majority of Appalachian theropod teeth assigned to tyrannosauroids, in which crown heights surpass 50 mm [8,9,16,18,41]. However, the tooth is notably larger than most North American dromaeosaurid teeth, which are often less than 10 mm in height and mostly measure around 5 mm in that dimension [29,33,40,42–44]. *Dakotaraptor*, which possessed crowns up to approximately 25 mm in crown height, represents the exception among Maastrichtian dromaeosaurids [40,45]. Instead, the Mt Laurel dromaeosaurid tooth is more comparable in size to the teeth of the 3–4 m *Deinonychus* and an indeterminate specimen from the Tar Heel Formation of North Carolina [42,46]. NJSM GP 22949 is also distinguished from Appalachian tyrannosauroids in lacking subquadrangular distocentral denticles [38]. The first discriminant analysis on the Mt Laurel dromaeosaurid tooth supports this hypothesis by placing the specimen within the convex hull formed by the teeth of *Velociraptor* and not in the convex hull formed by the teeth of tyrannosaurs or troodontids (figure 4*a*).

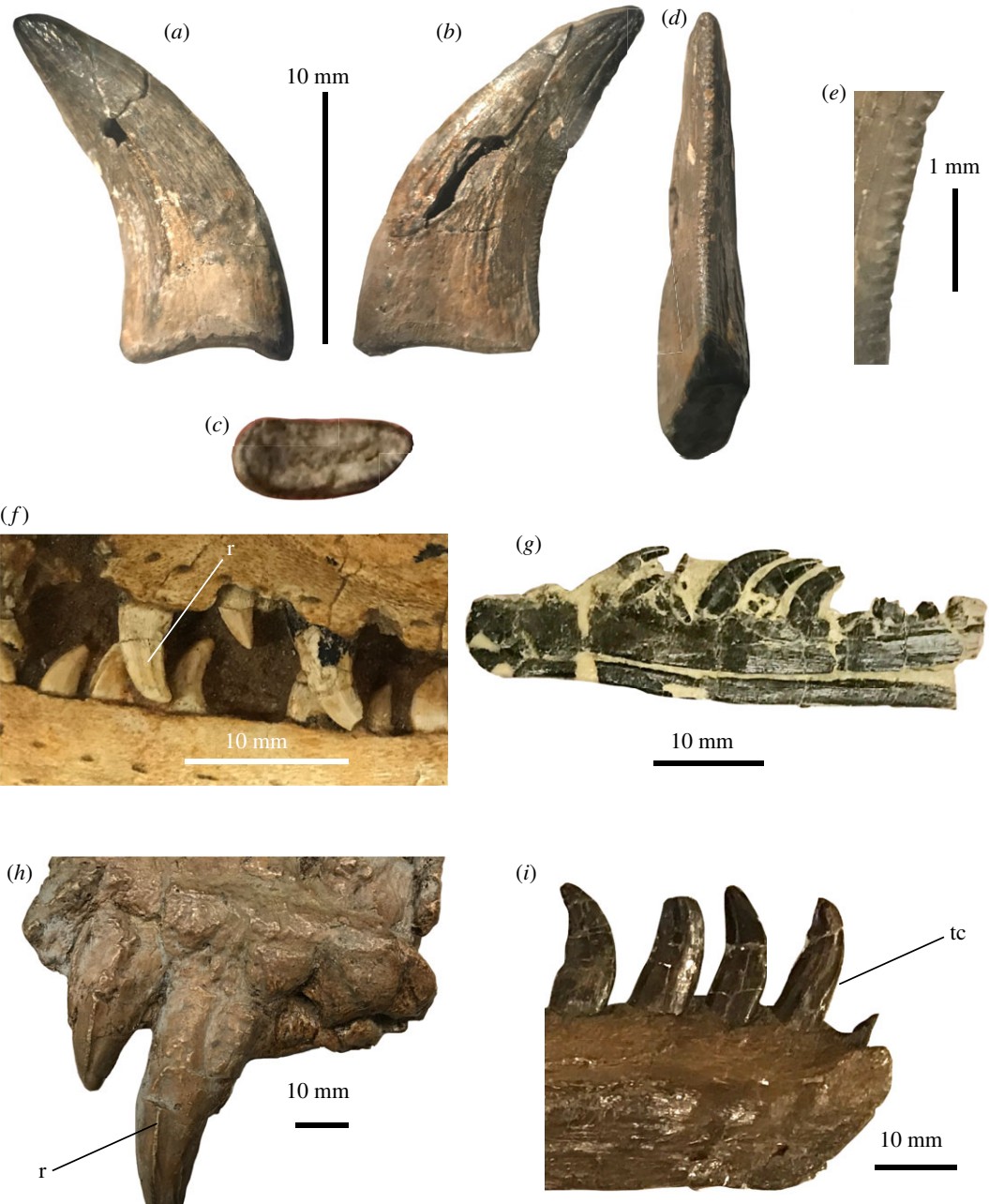

**Figure 3.** Anatomy of the Mt Laurel dromaeosaurid tooth. NJSM GP 22949 in labial (*a*), lingual (*b*), basal (*c*) and distal (*d*) views, with a close-up (*e*) of the distal denticles. (*f*) Teeth of *Velociraptor*, (*g*) dentary and teeth of '*Bambiraptor*', (*h*) premaxilla and teeth of *Utahraptor* and (*i*) dentary and teeth of *Dromaeosaurus*. r, enamel ridge; tc, twisted mesial carina.

NJSM GP 22949 is notable for being similar to the teeth of western North American saurornitholestine dromaeosaurids (figure 3*a–e,g*) [40] and somewhat unlike those previously discovered from the American east [18,19,47]. Teeth assigned to *Saurornitholestes* have been described from the Cretaceous of the southeastern USA [18,47]. These teeth are extremely small (less than 6 mm), far less recurved than NJSM GP 22949, and have proportionally large denticles that are more strongly apically hooked (e.g. fig. 1 in [47]). One tooth from Alabama, ALMNH (Alabama Museum of Natural History) 2001.1, measures 4.9 mm in crown height and preserves seven distal denticles and eight mesial denticles per millimetre. ALMNH 2001.1 is less recurved than NJSM GP 22949 and far less elongate in labial and lingual views (see [47]). Some saurornitholestine teeth from South Carolina [18] also lack the slender condition in NJSM GP 22949, where the mesiodistal width of the heavily recurved tooth crown is

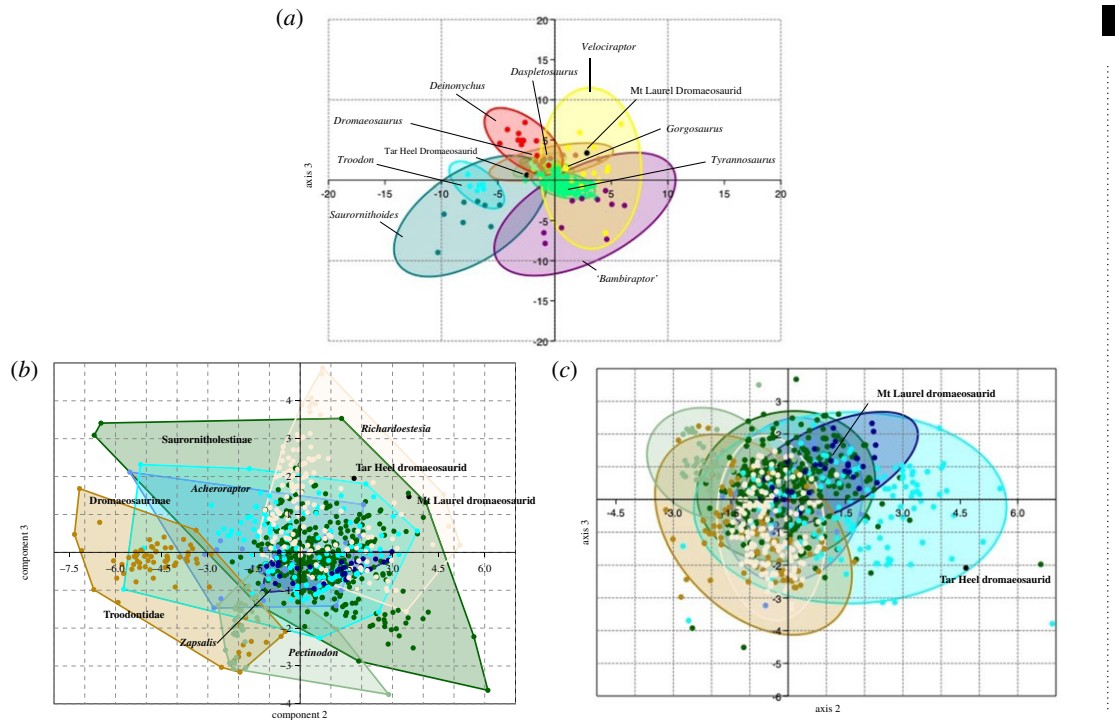

**Figure 4.** Principal components and discriminant analyses of the Mt Laurel dromaeosaurid tooth. (*a*) Discriminant components analysis of coelurosaurian teeth including a large dromaeosaurid tooth from North Carolina and the Mt Laurel dromaeosaurid tooth [46,50]. (*b*) PCA of North American paravian teeth including the Mt Laurel dromaeosaurid tooth [40]. Principal component 1 accounted for (*c*) discriminant analysis of North American paravian teeth including the Mt Laurel dromaeosaurid tooth [40].

much smaller than the crown height. Teeth from the Ellisdale site of New Jersey include dromaeosaurid crowns [41,48]. However, large crowns from Ellisdale are not 'slender' like NJSM GP 22949, are larger in size and bear distal denticles that are considerably more apically hooked than the Mt Laurel tooth and western saurornitholestines [48]. A dromaeosaurid tooth from North Carolina is slightly larger than NJSM GP 22949, but is less recurved and far less slender [46].

When compared with dromaeosaurid teeth from outside eastern North America, NJSM GP 22949 most closely resembles the teeth of western North American saurornitholestine dromaeosaurids. Despite the fact that NJSM GP 22949 was placed in the convex hull formed by the teeth of *Velociraptor* in the first discriminant analysis conducted, the tooth is unlike those of the western North American velociraptorine *Acheroraptor temertyorum* [49] or the Mongolian *Velociraptor mongoliensis* [39; figure 3*f*) in lacking strongly developed striations along its crown surface, in being more strongly recurved, and in being far more slender (lower CBL/CH value) (figure 3*a,b,f*) [49,50]. In contrast with *Dromaeosaurus albertensis*, and 'dromaeosaurine' teeth from western North America, the mesial carina in NJSM GP 22949 is not twisted onto the mesiolingual face of the crown, the distal denticles are apically hooked, and the tooth is more strongly recurved (figure 3*h,i*; [39,40]). The Mt Laurel tooth is far less robust and has far less-developed carinae than the teeth of *Utahraptor* (figure 3*h*). The Mt Laurel tooth is also smaller, much more strongly recurved, and possesses denticles more apically hooked than those of the giant Maastrichtian dromaeosaurid *Dakotaraptor steini* [45]. NJSM GP 22949 lacks the 'figure 8' basal cross-section seen in the teeth of *Deinonychus* [42,46]. Although the strongly recurved maxillary teeth of *Deinonychus* [39,42] are somewhat comparable with NJSM GP 22949, the differing basal cross-sections among these specimens from the slightly asymmetrical morphology of the teeth in *Deinonychus* distinguish *Deinonychus antirrhopus* and the Mt Laurel form. The discriminant analysis of the Larson & Currie [40] dataset supports saurornitholestine affinities for NJSM GP 22949, classifying the tooth as a saurornitholestine crown with 10 hits in the confusion matrix and the jackknifed classification placing the tooth in the 's' (saurornitholestine) group.

NJSM GP 22949 resembles the teeth of western North American Maastrichtian saurornitholestines in having a slender, tall outline in labial and lingual views (the 'Lancian' saurornitholestine morphotype of [40]). The tooth is closely comparable with the crowns of the juvenile saurornitholestine *Bambiraptor*

*feinbergi*, which are extremely recurved and slender and possess apically hooked denticles (figure 3*g*). A discriminant analysis of the Late Cretaceous western North American paravian tooth dataset of Larson & Currie [40] found NJSM GP 22949 to nest within the convex hulls formed by four tooth morphotypes (Saurornitholestinae, Dromaeosaurinae, *Zapsalis* and *Atrociraptor*). NJSM GP 22949 is also quantifiably unlike YPM VPPU.021397, the large dromaeosaurid tooth from the Campanian of North Carolina [46], plotting far from the southeastern North American specimen in both morphometric analyses (figure 4). Thus, NJSM GP 22949 is most comparable to the crowns of a saurornitholestine-like dromaeosaurid. Saurornitholestines are small-bodied dromaeosaurids [39], and so NJSM GP 22949 is important for indicating members of this group may have achieved relatively large body sizes for dromaeosaurs.

## 2.11. Description

NJSM GP 22949 is the complete crown of a dromaeosaurid dinosaur. Measurements of this specimen are in table 1. This tooth is heavily recurved, displaying the ziphiform condition. The crown possesses an ovoid basal cross-section. In distal view, the middle portion of tooth is convex labially, although the crown becomes labiolingually straightened towards its apex. The labial and lingual surfaces are flattened, and the lack of a root attached to this crown indicates it was shed. Although both the mesial and distal carinae are preserved, the mesial denticles have been mostly eroded away, and precise denticle counts for the mesial carina are unable to be taken. Some portions of the tooth crown are cracked, and the outer enamel layer is poorly preserved towards the distal end of the specimen. Small portions of the middle of the crown are missing. The distal profile of NJSM GP 22949 is strongly concave. The preserved portions of the outer enamel layer are smooth, although at the apex, several slightly developed ridges appear. These ridges could represent features of the original morphology of the tooth or be damage from feeding or taphonomic processes. The distal carina preserves a large number of apically hooked denticles that become smaller towards the apex of the crown. These denticles are separated by interdenticular sulci that, along with the serrations, project slightly onto the tooth surface. Unfortunately, the shape and density of the mesial denticles could not be determined, as the mesial carina is heavily eroded in NJSM GP 22949.

# 3. Discussion

The two theropod teeth described here add to one of the most complete Maastrichtian faunas from eastern North America. The dromaeosaurid tooth NJSM GP 22949 is biogeographically significant for being the first occurrence of this clade in the Mount Laurel Formation and more generally the Maastrichtian of eastern North America. Until now, tyrannosauroids and ornithomimosaurs were the only known theropods from the late Campanian–Maastrichtian of this area [5,9,13,17,20,25], with the latest records of dromaeosaurids in the American east hailing from mid-Campanian units in the Carolinas [18,46] and the Ellisdale site of New Jersey [41]. Although NJSM GP 22949 is most comparable to the crowns of mid-sized to large dromaeosaurids like *De. antirrhopus* and *Da. steini* [42,45] and to small tyrannosauroids [18,33,41] in its dimensions, it is most closely allied with dromaeosaurids in the morphometric analyses conducted (figure 4) and in many key features of its morphology.

The tyrannosauroid tooth NJSM GP 14256 supports the presence of *Dryptosaurus*-like tyrannosauroids in the early Maastrichtian of New Jersey. Isolated teeth and postcranial material from the Mount Laurel Formation were previously assigned to *Dryptosaurus* sp. based on little more than their geographical proximity to the site where the holotype of this taxon was recovered [13,17,20], but no detailed description of late Campanian to early Maastrichtian tyrannosauroids from New Jersey has appeared in the literature. The well-preserved nature of NJSM GP 14256 thus allows for the formal recognition of the presence of non-tyrannosaurid tyrannosauroids in the latest Campanian to earliest Maastrichtian of the Atlantic Coastal Plain. Furthermore, the excellent condition of the outer enamel layer of NJSM GP 12456 allows for further documentation of the dental anatomy of Appalachian tyrannosauroids, the isolated teeth of which are often found highly abraded among stream deposits (e.g. [17]). The presence of non-tyrannosaurid tyrannosauroids in the Mt Laurel Formation is expected, given the presence of *Dr. aquilunguis* and non-tyrannosaurid tyrannosauroids of similar phylogenetic position in both the middle-late Maastrichtian Navesink and New Egypt formations [9,13] and early Campanian Marshalltown Formation [41]. However, that NJSM GP 14256, originally discovered in 1984, is only described now attests to the understudied nature of these deposits.

The late recognition of dromaeosaurids in the Maastrichtian sediments of New Jersey is notable, given that the dinosaurs of the Mt Laurel and other Cretaceous units in the Atlantic Coastal Plain have been studied for over a century and a half (e.g. [13,17,20,26,27]). Teeth from locations like the Ellisdale site of the Marshalltown Formation of New Jersey originally assigned to tyrannosaurs have more recently been reclassified as the crowns of dromaeosaurids [13,20,41], so it is entirely possible that the lack of diversity in dinosaur faunas from the Maastrichtian of the Atlantic Coastal Plain reflects the misidentification of these isolated fossils. Only further work on Appalachian fossils will allow for more comprehensive revision of the identification of fossils from this area.

During the Cretaceous, terrestrial faunas became more regionalized as the break-up of supercontinents like Gondwana and Laurasia and the inundation of smaller landmasses like North America and Europe occurred (e.g. [37,51–54]). In particular, the assemblages of Appalachia and Laramidia, which became isolated from each other as the Western Interior Seaway flooded the middle of North America (e.g. [51,53,55,56]), have been recognized as highly distinctive (e.g. [13,17,18,20,41,53]). In the past 30 years, a handful of discoveries from the eastern margin of North America have indicated some faunal interchange occurred between Laramidian and Appalachian dinosaur communities during the latest Cretaceous. These include the tooth of a possible chasmosaurine ceratopsid from the latest Maastrichtian of Mississippi and the fragmentary forelimb material of possible lambeosaurines from the Maastrichtian of New Jersey and earliest Maastrichtian of Nunavut, Canada [20,22,57]. At the same time, Appalachian faunas continued to harbour endemic forms like intermediate tyrannosauroids, represented by *Dr. aquilunguis* and comparable taxa through the Campanian–Maastrichtian transition in the Atlantic Coastal Plain ([9,13]; this paper). The latter finds suggest that eastern North America may have continued to act as a sort of refugium up until the end of the Mesozoic. Faunal interchange in the last 10 Myr of the Mesozoic seems to have occurred throughout the Northern Hemisphere. Phylogenetic evidence posits that the presence of *Tyrannosaurus rex* in the Maastrichtian of the western USA and Canada represents a dispersal of Asian tyrannosaurids into the Americas [34,58]. A dispersal event between Asian and North American faunas during the Maastrichtian may have occurred over Beringia [59], given the similarity of Maastrichtian polar faunas from Russia, Alaska and lower latitudes in the USA and Canada (e.g. [4,60–62]). Along with previous discoveries, the tyrannosauroid tooth described here supports the 'refugium' model for latest Cretaceous eastern North America, wherein taxa more closely allied with middle Cretaceous forms (e.g. non-tyrannosaurid tyrannosauroids like *Dryptosaurus* and *Appalachiosaurus*) [8,9,34] persisted in relative isolation as more phylogenetically advanced forms evolved in Laurasia.

## 4. Conclusion

Two theropod teeth from the Campanian–Maastrichtian Mount Laurel Formation of New Jersey are described in detail. The dromaeosaurid tooth, which plots with western North American saurornitholestine teeth in principal components and discriminant analyses, is the youngest record of a non-avian maniraptoran from eastern North America and the first from the latest Campanian–Maastrichtian of the American east. This tooth provides another record of a mid-sized to large dromaeosaurid in the Cretaceous of eastern North America. However, this tooth is more allied with those of saurornitholestines and velociraptorines than with *Deinonychus*, dromaeosaurines or large dromaeosaurid teeth previously described from Appalachia, tentatively suggesting that several types of dromaeosaurids might have grown to relatively large sizes in the Cretaceous of the eastern USA and indicating mid-sized to largish dromaeosaurids were a usual component of Appalachian faunas. The tyrannosauroid tooth is the first specimen to suggest the presence of *Dr. aquilunguis* or a closely related tyrannosauroid in the Mt Laurel ecosystem, further supporting the refugium model for Appalachian vertebrate evolution.

## 5. Methods

### 5.1. Measurements and nomenclature

Measurements of both teeth were taken in accordance with the methodology of Smith *et al.* [50] and Larson & Currie [40]. The dimensions of the Mt Laurel teeth were determined using digital callipers. I follow the nomenclature of Hendrickx *et al.* [38] when describing the two teeth on which this paper focuses.

## 5.2. Principal components analyses

In order to quantitatively test the identity of the dromaeosaurid tooth among theropod dinosaurs, I included it in principal components and discriminant analyses conducted in the program PAST v. 3.18 [63]. In order to assess the morphological similarity of the Mt Laurel teeth to theropod clades present in the Cretaceous of the northern hemisphere, I used a modified version [46] of the dataset of Smith *et al.* [50] that includes tooth data on tyrannosauroids, troodontids and dromaeosaurids. A principal components analysis (PCA) was run on this dataset, which included data on 15 measurements: crown height (CH), crown base length (CBL), crown base width (CBW), apicobasal length (length of the tooth along the longest apicobasal axis) and serration density per 5 mm for the basal (DB), mid-crown (DC) and apical (DA) distal carina. An additional PCA was conducted using the dataset of Larson & Currie [40] in order to better assess the similarity of the Mt Laurel dromaeosaurid specimen to other North American paravian teeth. This PCA assessed for five measurements: CH, CBL, CBW, and the mesial (MD) and distal (DD) denticles per millimetre. The summary statistics and loadings from the results of the PCAs conducted are included in the electronic supplementary material.

## 5.3. Discriminant analyses

To further assess the affinities of the Mt Laurel dromaeosaurid tooth, I performed a discriminant analysis on the tooth datasets of Smith *et al.* ([50], modified in [46]) and Larson & Currie [40]. This analysis creates a morphospace by maximally separating objects sorted into predetermined groups. This analysis was also run in PAST v. 3.18 [63], and the loadings and confusion matrices can be found in the electronic supplementary material.

## 5.4. Phylogenetic analysis

To test the referral of the incomplete tyrannosauroid tooth to that family, I coded the specimen for the phylogenetic matrix of Hendrickx & Mateus [35], a dataset of theropod dentition that includes 64 taxa/specimens coded for 141 characters. The matrix was entered into the phylogenetics program TNT 1.5 [64] for a phylogenetic analysis. The matrix was first analysed using the 'New Technology Search', with default parameters for ratchet, tree drift, tree fuse and sectorial search. A total of 10 trees of length 688 were retained. These topologies were then subjected to traditional branch swapping, which allows for a more extensive exploration of each tree island. This found more than 99 999 most parsimonious topologies of 688 steps. These were summarized in a strict consensus topology (consistency index = 0.290; retention index = 0.446).

Ethics. All specimens examined are in licensed public repositories. Access to the New Jersey State Museum was provided by Dana Ehret, David Parris and Rodrigo Pellegrini. Access to the American Museum of Natural History was provided by Carl Mehling, and access to the Yale Peabody Museum was provided by Daniel Brinkman.

Data accessibility. All data are in the electronic supplementary material.

Competing interests. The author declares no competing interests.

Funding. The author received no funding for this work.

Acknowledgements. I thank Dana Ehret for access to the collections of the New Jersey State Museum, Carl Mehling for access to the collections of the American Museum of Natural History and Daniel Brinkman for access to the collections of the Yale Peabody Museum of Natural History. I thank Howard Falcon-Lang, Christophe Hendrickx, Terry Gates, Kevin Padian and four anonymous reviewers for their comments, which greatly improved the quality of this paper.

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
