## [Reviewer comments · Royal Society Open Science]

Review History

RSOS-190533.R0 (Original submission)

Review form: Reviewer 1

Is the manuscript scientifically sound in its present form?

No

Are the interpretations and conclusions justified by the results?

No

Is the language acceptable?

Yes

Is it clear how to access all supporting data?

Yes

Do you have any ethical concerns with this paper?

No

Have you any concerns about statistical analyses in this paper?

Yes

Recommendation?

Major revision is needed (please make suggestions in comments)

Comments to the Author(s)

This manuscript hinges upon the claim that you have a non-tyrannosaurid tyrannosauroid with affinities to eastern taxa (ie. *Dryptosaurus*) and a dromaeosaurid with western affinities in the same formation. Therefore, the manuscript requires both these identifications and these different affinities to be well-supported.

The text in the descriptions does a good job of comparing these teeth to other comparable taxa (I should note, however, that a twisted mesial carina on lateral teeth is only documented from *D. albertensis* and not from other similar isolated teeth or *Utahraptor*).

However, your data and analyses either do not support your main claim or are insufficient to do so. Starting with your dataset, your measurements of NJSMP GP 12456 for DB, DC, and DA do not appear congruent with the specimen figured in Fig. 1J, though do seem to match the scale if Fig. 1E. In Fig. 1J, measurements closer to 4 would appear to be more accurate than 2, 2, and 2.5. The scale bars for the photos and your measurements should be double-checked. You also have a substantial amount of missing data for your new specimens in your dataset, and you haven't discussed how this was handled in your PCA. The default in PAST is to fill missing data with column average substitution, which in the Smith et al. (2005) dataset will bias all of your specimens towards the tyrannosaur part of your plot (as there are more tyrannosaurs in the dataset than any other group). Given that all three of the new teeth plot in areas of your plots where no other theropod teeth plot (partway between dromaeosaurs and tyrannosaurs), it appears that column average substitution of your dataset is having a substantial contribution to the position of your specimens on these plots. As well, eastern taxa are notably excluded. *Dryptosaurus* is not plotted in Fig. 2A, and the Tar Heel dromaeosaurid is not plotted in Fig. 4B and 4C, making the argument that either of these teeth have eastern or western affinities insufficiently supported by these analyses. Additions of those specimens and the exclusion of variables with missing data are necessary for these plots to be informative.

In the phylogenetic analysis, you only list shared derived characters between your tyrannosaur specimen and *T. rex*, but do not list shared derived characters that unite all tyrannosauroids. However, even this analysis does not support your argument that the specimen is a non-tyrannosaurid tyrannosauroid with affinities with *Dryptosaurus*. Also, one of the four characters you do list, 94) biconvex apical distal denticles, is not mentioned in your description at all nor is it visible in Fig. 1.

As well, Fig. 1B, 1C, and 1D depict fossils that are not described in the manuscript. If these specimens are figured in this manuscript, they should be described systematically in the text. Alternatively, these figures could easily be excluded from the manuscript. The entire datasets used for analysis, including the Smith et al. (2005) and Larson and Currie (2013) should be reproduced in the supplemental data for ease of replication.

This description is of specialist interest given the new records from eastern North America, but any discussion of faunal interchange is premature. In your description, you have demonstrated that you have teeth of a dromaeosaurid and a tyrannosauroid from this formation, but your analyses in their current form are insufficient to demonstrate that these teeth have any biogeographic affinities.

Review form: Reviewer 2 (Terry A. Gates)

Is the manuscript scientifically sound in its present form?

No

Are the interpretations and conclusions justified by the results?

Yes

Is the language acceptable?

Yes

Is it clear how to access all supporting data?

Yes

Do you have any ethical concerns with this paper?

No

Have you any concerns about statistical analyses in this paper?

Yes

Recommendation?

Major revision is needed (please make suggestions in comments)

Comments to the Author(s)

The author of this paper has identified two theropod dinosaur teeth from the east coast of North America they attribute to Tyrannosauroidae and Dromaeosauridae.

Further the author suggests ways that these teeth contribute to our knowledge of faunal interchange between eastern and western North America at the Campano-Maastrichtian boundary.

This paper is important for the acknowledgement of the taxa found in this New Jersey formation. Outside of this finding there is much more difficulty in acquiring significance. This problem of lack of greater significance is not the fault of the author but the nature of the Appalachian dinosaur record. Fossiliferous formations are poorly dated, fossils are not terribly abundant, and when known are typically in rough condition. Because there is not a proper framework for the dinosaurs, determining faunal interchange is problematic...you need to know what is around when, and the organism's phylogenetic relationships. Without this it is difficult to say anything concrete about paleobiology. Especially, in these days when doing rigorous biogeographic analyses is so easy. In my opinion you can't really say anything about faunal interchange until a solid, dated fossil record is assembled.

I am glad to see the author using quantitative techniques to identify the teeth. However, I have some issues with the methods and/or the results presented.

First, I firmly do not think that a phylogenetic analysis should be used on a single tooth, much less a single partial tooth. The phylogenetic analysis of teeth is not a proper use of phylogenetics in my opinion at any rate. To my mind this part of the paper should be eliminated.

Second, the author should use a more recent ordinal analysis of teeth. This is not a deal breaker, but the primary dataset chosen is 13 years old (or 5 yrs old in the case of Larson and Currie) and many others have been produced since then.

Third, the tyrannosaur tooth should absolutely be removed from the PCA analysis. It is a fragmentary tooth, therefore any measurements taken are not going to be the correct measurement and will provide erroneous results. Or there will be too little data to say much about.

Fourth, the dromaeosaurid tooth falls outside the dromaeosaurids in the Smith/Brownstein matrix. And is just part of the miasma of teeth in the Larson and Currie matrix. I would suggest pulling individual taxa from the analyses and talk about comparisons with them instead of consistently saying it is a Saurornitholestesine theropod. Remember that the PCA is using data you input. If there is more information supporting your taxonomic claim that is not included in the quantitative data, then you should discuss this in the text. In PAST you can change the symbol and the color, meaning that you can label individual dinosaur taxa and the higher level taxonomic groups at the same time. This should be done to help you see which species the Tar Heel and Mt Laurel specimens are mostly similar to.

Fifth, this is very important! When using PCA on variables that are measurements, the first axis is almost invariably a reflection of size, not shape. Therefore, you need to show axes 2 and 3. You are welcome to retain axis 1 and 2, but 2 and 3 must be shown in addition. You may also want to consider Centering or z-transforming your data because it can help remove the effect of size, bring outliers closer together, and reflect shape better.

I have provided a marked PDF with more comments and suggestions (Appendix A).

Hopefully these suggestions help the paper and the author.

Sincerely,
Terry Gates

Decision letter (RSOS-190533.R0)

10-Jul-2019

Dear Mr Brownstein:

Manuscript ID RSOS-190533 entitled "New records of theropods from New Jersey inform faunal interchange in Maastrichtian North America" which you submitted to Royal Society Open Science, has been reviewed. The comments from reviewers are included at the bottom of this letter.

In view of the criticisms of the reviewers, the manuscript has been rejected in its current form. However, a new manuscript may be submitted which takes into consideration these comments.

Please note that resubmitting your manuscript does not guarantee eventual acceptance, and that your resubmission will be subject to peer review before a decision is made.

Once you have revised your manuscript, go to <https://mc.manuscriptcentral.com/rsos> and login to your Author Center. Click on "Manuscripts with Decisions," and then click on "Create a

Resubmission" located next to the manuscript number. Then, follow the steps for resubmitting your manuscript.

Your resubmitted manuscript should be submitted by 07-Jan-2020. If you are unable to submit by this date please contact the Editorial Office.

on behalf of Dr Robert Sansom (Associate Editor) and Kevin Padian (Subject Editor)
openscience@royalsociety.org

Subject Editor Comments to Authors:

Both reviewers (and I) appreciate the great amount of work that has gone into this manuscript and the level of painstaking analysis. That said, neither reviewer is happy with the level of biogeographic inference, and this is largely because they are not happy with the taxonomic identifications and use of phylogenetic inference. Without these the paper becomes considerably more limited in scope, and that would be a different paper.

I will be glad to entertain a resubmission if the advice of the reviewers is followed, or, alternatively, if you can refute their concerns to their satisfaction. Best wishes.

Reviewers' Comments to Author:

Reviewer: 1

Comments to the Author(s)

This manuscript hinges upon the claim that you have a non-tyrannosaurid tyrannosauroid with affinities to eastern taxa (ie. *Dryptosaurus*) and a dromaeosaurid with western affinities in the same formation. Therefore, the manuscript requires both these identifications and these different affinities to be well-supported.

The text in the descriptions does a good job of comparing these teeth to other comparable taxa (I should note, however, that a twisted mesial carina on lateral teeth is only documented from *D. albertensis* and not from other similar isolated teeth or *Utahraptor*).

However, your data and analyses either do not support your main claim or are insufficient to do so. Starting with your dataset, your measurements of NJSM GP 12456 for DB, DC, and DA do not appear congruent with the specimen figured in Fig. 1J, though do seem to match the scale if Fig. 1E. In Fig. 1J, measurements closer to 4 would appear to be more accurate than 2, 2, and 2.5. The scale bars for the photos and your measurements should be double-checked. You also have a substantial amount of missing data for your new specimens in your dataset, and you haven't discussed how this was handled in your PCA. The default in PAST is to fill missing data with column average substitution, which in the Smith et al. (2005) dataset will bias all of your

specimens towards the tyrannosaur part of your plot (as there are more tyrannosaurs in the dataset than any other group). Given that all three of the new teeth plot in areas of your plots where no other theropod teeth plot (partway between dromaeosaurs and tyrannosaurs), it appears that column average substitution of your dataset is having a substantial contribution to the position of your specimens on these plots. As well, eastern taxa are notably excluded. *Dryptosaurus* is not plotted in Fig. 2A, and the Tar Heel dromaeosaurid is not plotted in Fig. 4B and 4C, making the argument that either of these teeth have eastern or western affinities insufficiently supported by these analyses. Additions of those specimens and the exclusion of variables with missing data are necessary for these plots to be informative.

In the phylogenetic analysis, you only list shared derived characters between your tyrannosaur specimen and *T. rex*, but do not list shared derived characters that unite all tyrannosauroids. However, even this analysis does not support your argument that the specimen is a non-tyrannosaurid tyrannosauroid with affinities with *Dryptosaurus*. Also, one of the four characters you do list, 94) biconvex apical distal denticles, is not mentioned in your description at all nor is it visible in Fig. 1.

As well, Fig. 1B, 1C, and 1D depict fossils that are not described in the manuscript. If these specimens are figured in this manuscript, they should be described systematically in the text. Alternatively, these figures could easily be excluded from the manuscript. The entire datasets used for analysis, including the Smith et al. (2005) and Larson and Currie (2013) should be reproduced in the supplemental data for ease of replication.

This description is of specialist interest given the new records from eastern North America, but any discussion of faunal interchange is premature. In your description, you have demonstrated that you have teeth of a dromaeosaurid and a tyrannosauroid from this formation, but your analyses in their current form are insufficient to demonstrate that these teeth have any biogeographic affinities.

Reviewer: 2

Comments to the Author(s)

The author of this paper has identified two theropod dinosaur teeth from the east coast of North America they attribute to Tyrannosauroidae and Dromaeosauridae.

Further the author suggests ways that these teeth contribute to our knowledge of faunal interchange between eastern and western North America at the Campano-Maastrichtian boundary.

This paper is important for the acknowledgement of the taxa found in this New Jersey formation. Outside of this finding there is much more difficulty in acquiring significance. This problem of lack of greater significance is not the fault of the author but the nature of the Appalachian dinosaur record. Fossiliferous formations are poorly dated, fossils are not terribly abundant, and when known are typically in rough condition. Because there is not a proper framework for the dinosaurs, determining faunal interchange is problematic...you need to know what is around when, and the organism's phylogenetic relationships. Without this it is difficult to say anything concrete about paleobiology. Especially, in these days when doing rigorous biogeographic analyses is so easy. In my opinion you can't really say anything about faunal interchange until a solid, dated fossil record is assembled.

I am glad to see the author using quantitative techniques to identify the teeth. However, I have some issues with the methods and/or the results presented.

First, I firmly do not think that a phylogenetic analysis should be used on a single tooth, much less a single partial tooth. The phylogenetic analysis of teeth is not a proper use of phylogenetics in my opinion at any rate. To my mind this part of the paper should be eliminated.

Second, the author should use a more recent ordinal analysis of teeth. This is not a deal breaker, but the primary dataset chosen is 13 years old (or 5 yrs old in the case of Larson and Currie) and many others have been produced since then.

Third, the tyrannosaur tooth should absolutely be removed from the PCA analysis. It is a fragmentary tooth, therefore any measurements taken are not going to be the correct measurement and will provide erroneous results. Or there will be too little data to say much about.

Fourth, the dromaeosaurid tooth falls outside the dromaeosaurids in the Smith/Brownstein matrix. And is just part of the miasma of teeth in the Larson and Currie matrix. I would suggest pulling individual taxa from the analyses and talk about comparisons with them instead of consistently saying it is a Saurornitholestesine theropod. Remember that the PCA is using data you input. If there is more information supporting your taxonomic claim that is not included in the quantitative data, then you should discuss this in the text. In PAST you can change the symbol and the color, meaning that you can label individual dinosaur taxa and the higher level taxonomic groups at the same time. This should be done to help you see which species the Tar Heel and Mt Laurel specimens are mostly similar to.

Fifth, this is very important! When using PCA on variables that are measurements, the first axis is almost invariably a reflection of size, not shape. Therefore, you need to show axes 2 and 3. You are welcome to retain axis 1 and 2, but 2 and 3 must be shown in addition. You may also want to consider Centering or z-transforming your data because it can help remove the effect of size, bring outliers closer together, and reflect shape better.

I have provided a marked PDF with more comments and suggestions.

Hopefully these suggestions help the paper and the author.

Sincerely,
Terry Gates

Author's Response to Decision Letter for (RSOS-190533.R0)

See Appendix B.

RSOS-191206.R0

Review form: Reviewer 2 (Terry A. Gates)

Is the manuscript scientifically sound in its present form?

No

Are the interpretations and conclusions justified by the results?

No

Is the language acceptable?

Yes

Do you have any ethical concerns with this paper?

No

Have you any concerns about statistical analyses in this paper?

No

Recommendation?

Accept with minor revision (please list in comments)

Comments to the Author(s)

The paper is much improved since I last reviewed it. There are comments posted in another annotated pdf attached to this review (Appendix C).

Important changes that still need to be made.

First and foremost--the phylogenetic analysis of teeth. You must use the super matrix of Hendrickx and Mateo, not just the tooth matrix. Please recode and rerun the analysis. Also, you should add Dryptosaurus and potentially one other tyrannosaurid to the matrix so that you might get some resolution as to the phylogenetic affinities of your tooth. Without this there is little that can be said except it is a tyrannosauroid tooth.

Put some or all of the Discriminant Analysis results in the paper, not the SM. We need to see the data.

The last sentence of the abstract does not make sense. I suggest deleting it, but you can try to modify it.

Good luck with further revisions.

Terry Gates

Review form: Reviewer 3

Is the manuscript scientifically sound in its present form?

Yes

Are the interpretations and conclusions justified by the results?

Yes

Is the language acceptable?

Yes

Do you have any ethical concerns with this paper?

No

Have you any concerns about statistical analyses in this paper?

No

Recommendation?

Accept with minor revision (please list in comments)

Comments to the Author(s)

This revision of the original paper is much better present and the interpretations are supported. I think it is nearly ready for publication. I made minor corrections to the MS on the PDF (Appendix D). Please be careful with the use of the clade names. Tyrannosaurids = members of Tyrannosauridae. I noted these throughout. Also, put the description before the identification of each tooth section.

Review form: Reviewer 4 (Thomas Carr)

Is the manuscript scientifically sound in its present form?

No

Are the interpretations and conclusions justified by the results?

Yes

Is the language acceptable?

Yes

Do you have any ethical concerns with this paper?

No

Have you any concerns about statistical analyses in this paper?

No

Recommendation?

Accept with minor revision (please list in comments)

Comments to the Author(s)

Dear Editor,

I submit to you my review of MS RSOS-191206, "New records of theropods from New Jersey inform faunal interchange in Maastrichtian North America" by Brownstein. This article is a long-overdue consideration of the possibility of faunal exchange between Laramidia and Appalachia during the latest Cretaceous. Virtually all paleogeographic maps of this time show Laramidia and Appalachia joined across what is now western Canada, where previously they were decisively separated by the Western Interior Seaway (WIS). However, the complete absence of Appalachian dinosaurs from the Hell Creek Formation and its equivalents sets the null hypothesis that faunal exchange, at least from east to west, did not occur. In his manuscript, Brownstein compares these hypotheses from the perspective of the east.

If dispersal went in the opposite direction (west to east), then fossils of, say, *Tyrannosaurus rex* and *Triceratops* should be present in late Maastrichtian sedimentary rocks of Appalachia. Indeed, if *T. rex* did disperse from Central Asia to Laramidia, becoming nearly instantaneously widespread across that landmass, then Appalachia should have been occupied in the same

fashion. But that isn't what the fossil record shows: certainly, opportunity for faunal exchange presented itself at 76, 74, and 68 million years ago, but there is no evidence of it in Laramidia, and the basal nature of *Appalachiosaurus* (Campanian) and *Dryptosaurus* (Maastrichtian) are consistent with that pattern. In the end, the author does conclude that the tooth specimens that he describes provide evidence for the refugium hypothesis, a conclusion that I agree with.

I suggest publication with minor revision; in addition to the specific comments below, I think the author should more clearly couch the article in terms of the refugium versus exchange models. In the introduction and conclusion, he leans heavily on the exchange model and that needs to be balanced, in both places, by a skeptical assessment of the case for the exchange model. An explicit statement of the evidence in favor of the refugium hypothesis must also be included in the abstract and the title must be adjusted accordingly.

The author may know my identity: Thomas D. Carr

Sincerely,
 Thomas D. Carr, PhD
 Associate Professor of Biology
 Carthage College
 2001 Alford Park Drive
 Kenosha, WI 53140
 p/262-551-5887

Specific comments:

Abstract: the phrase "fine detail" does not need to be hyphenated.

The phrase "heavily mediolaterally compressed" is too wordy and misleading; the word "narrow" with a width to length ratio (in per cent) would suffice. My quibble is this: narrow teeth (w:l ratio less than 60%) is merely the plesiomorphic condition for tyrannosauroids. Since narrow teeth are the ancestral condition, there is no process of "compression" to account for their form.

Replace "this fossil supports" with "this fossil is evidence for".

Replace the word "common" with "widespread".

Introduction

Page 9, line 60: delete "rectangular".

Page 10, lines 74 to 74: move "come" up to follow "have".

Page 11, lines 97, 98: replace the em-dashes with "to".

Page 11, line 111: replace "border" with "boundary".

Tyrannosauroid tooth

Identification

Page 13, line 147: insert a space between the parentheses.

Page 13, lines 151-152: replace "highly mediolaterally compressed" with "narrow" and give a width to length ratio (mentioning, of course, that the tooth is incomplete and so the ratio is an overestimate).

Page 13, line 155: replace "high mediolateral compression" with "narrow".

Page 13, line 158: delete "pers. obs. of".

Page 14, line 161: mention that the crown is incomplete and that you only have the apical third to half, otherwise your inability to take all measurements, as said here, is unclear.

Page 14, line 169: insert a space between the parentheses.

Table 1: the specimen number “NJSM GP 12456” does not match the number in the text (NJSM GP 14256); which is correct?

Is there a caption that explains the abbreviations? If not, please include.

Why is the basal width to basal length ratio not included?

Page 15, line 181: modern erosion or ancient erosion? Clarify.

Page 15, line 183: a single tooth cannot be “ziphodont”, which is a term used for the entire dentition; “ziphiform” is appropriate for a single tooth; replace “compressed” with “narrow”.

Page 15, line 184: in which direction is the mesial carina convex: labial or lingual? Please clarify.

Page 15, line 185: replace “run” with “extent”.

Page 15: are the apical denticles worn down; i.e., through use, as produced the subtle wear facet on the lingual surface? Please include that observation, if you can.

Page 15, line 193: replace “interspersed with diminutive” with “separated by small”.

Figures: I did not receive any figure captions with the manuscript, so I cannot comment on those.

Dromaeosaurid tooth

Identification

Page 16, line 219: separate the parentheses with a space.

Page 17, line 223: explicitly state the crown height here.

Page 17, line 238: separate the parentheses with a space.

Page 18, lines 248, 251: placing a word or phrase between apostrophes implies that you are disowning what you are saying; since this clearly isn't the case, please delete them – they are distracting and self-contradictory. If you are insecure about using slender as a qualitative term, you could quantify the condition of slender with a ratio of, say, basal length to crown height or the like.

Page 18, line 249: give the ratio!

Page 18, line 252: is the term “western” necessary? Or do you mean exclusively Laramidia?

Clarify.

Page 18, line 254: give a ratio to quantify this contrast in slenderness.

Page 18, line 258: delete “conducted”.

Page 18, line 262: separate the parentheses with spaces.

Page 19, line 274: replace “and” with “from”.

Page 19, line 281: the name *B. feinbergi* should not be enclosed in apostrophes.

Page 19, line 288: replace “analysis” with “analyses”; delete “in which...included”.

Description

Page 20, line 295: replace “ziphodont” with “ziphiform”.

Page 20, line 308: replace “project” with “extend”.

Discussion

Page 21, line 315: insert “the” ahead of “Maastrichtian”.

Page 21, line 321: is “largish” a real word? Please fix!

Page 23: I'm skeptical of the claim for interchange – why can't the ceratopsian and lambeosaurine material simply reflect descendants of those clades before the inundation of the WIS? For example, lambeosaurines do have Santonian representatives in Asia. My impression is that the opportunities for exchange between Asia and Laramidia were generally episodic, whereas it was sustained during the late Maastrichtian. Regardless, the evidence for exchange between Laramidia and Appalachia isn't strong even though it is a worthwhile hypothesis to test.

Page 23, line 381: separate the parentheses with a space.

Page 24, line 395: argh – “largish” again! Please fix.

Methods

Page 25, line 406: change “provide support for the assignment” to “quantitatively test the identity”.

Page 26, line 429: change “provide additional support for” to “test”.

Figure 3: Perhaps it is in the caption, but please make it clear what the “ML” specimens are.

Decision letter (RSOS-191206.R0)

23-Sep-2019

Dear Mr Brownstein

On behalf of the Editor, I am pleased to inform you that your Manuscript RSOS-191206 entitled "New records of theropods from New Jersey inform faunal interchange in Maastrichtian North America" has been accepted for publication in Royal Society Open Science subject to minor revision in accordance with the referee suggestions. Please find the referees' comments at the end of this email.

The reviewers and Subject Editor have recommended publication, but also suggest some minor revisions to your manuscript. Therefore, I invite you to respond to the comments and revise your manuscript. The reviewers have provided comments in PDF format to assist you - as these appear to be large files, and sometimes the ScholarOne mail server strips messages of large attachments, please let me know if you do not receive these.

- Ethics statement

- Data accessibility

<http://datadryad.org/submit?journalID=RSOS&manu=RSOS-191206>

- Competing interests

- Authors' contributions

- Acknowledgements

- Funding statement

Because the schedule for publication is very tight, it is a condition of publication that you submit the revised version of your manuscript before 02-Oct-2019. Please note that the revision deadline will expire at 00.00am on this date. If you do not think you will be able to meet this date please let me know immediately.

- 1) A text file of the manuscript (tex, txt, rtf, docx or doc), references, tables (including captions) and figure captions. Do not upload a PDF as your "Main Document".
- 2) A separate electronic file of each figure (EPS or print-quality PDF preferred (either format should be produced directly from original creation package), or original software format)

- 3) Included a 100 word media summary of your paper when requested at submission. Please ensure you have entered correct contact details (email, institution and telephone) in your user account
- 4) Included the raw data to support the claims made in your paper. You can either include your data as electronic supplementary material or upload to a repository and include the relevant doi within your manuscript
- 5) All supplementary materials accompanying an accepted article will be treated as in their final form. Note that the Royal Society will neither edit nor typeset supplementary material and it will be hosted as provided. Please ensure that the supplementary material includes the paper details where possible (authors, article title, journal name).

on behalf of Dr Robert Sansom (Associate Editor) and Kevin Padian (Subject Editor)
openscience@royalsociety.org

Associate Editor Comments to Author (Dr Robert Sansom):

The reviewers are impressed with this revised version and suggest only minor revisions. I apologise for the long delay in getting this back to you; I have never had more difficulty in procuring reviewers than for this submission.

Reviewer comments to Author:
Reviewer: 2

Comments to the Author(s)

The paper is much improved since I last reviewed it. There are comments posted in another annotated pdf attached to this review.

Important changes that still need to be made.

First and foremost--the phylogenetic analysis of teeth. You must use the super matrix of Hendrickx and Mateo, not just the tooth matrix. Please recode and rerun the analysis. Also, you should add *Dryptosaurus* and potentially one other tyrannosaurid to the matrix so that you might get some resolution as to the phylogenetic affinities of your tooth. Without this there is little that can be said except it is a tyrannosauroid tooth.

Put some or all of the Discriminant Analysis results in the paper, not the SM. We need to see the data.

The last sentence of the abstract does not make sense. I suggest deleting it, but you can try to modify it.

Good luck with further revisions.

Terry Gates

Reviewer: 3

Comments to the Author(s)

This revision of the original paper is much better present and the interpretations are supported. I think it is nearly ready for publication. I made minor corrections to the MS on the PDF. Please be careful with the use of the clade names. Tyrannosaurids = members of Tyrannosauridae. I noted these throughout. Also, put the description before the identification of each tooth section.

Reviewer: 4

Comments to the Author(s)

Dear Editor,

I submit to you my review of MS RSOS-191206, "New records of theropods from New Jersey inform faunal interchange in Maastrichtian North America" by Brownstein. This article is a long-overdue consideration of the possibility of faunal exchange between Laramidia and Appalachia during the latest Cretaceous. Virtually all paleogeographic maps of this time show Laramidia and Appalachia joined across what is now western Canada, where previously they were decisively separated by the Western Interior Seaway (WIS). However, the complete absence of Appalachian dinosaurs from the Hell Creek Formation and its equivalents sets the null hypothesis that faunal exchange, at least from east to west, did not occur. In his manuscript, Brownstein compares these hypotheses from the perspective of the east.

If dispersal went in the opposite direction (west to east), then fossils of, say, *Tyrannosaurus rex* and *Triceratops* should be present in late Maastrichtian sedimentary rocks of Appalachia. Indeed, if *T. rex* did disperse from Central Asia to Laramidia, becoming nearly instantaneously widespread across that landmass, then Appalachia should have been occupied in the same fashion. But that isn't what the fossil record shows: certainly, opportunity for faunal exchange presented itself at 76, 74, and 68 million years ago, but there is no evidence of it in Laramidia, and the basal nature of *Appalachiosaurus* (Campanian) and *Dryptosaurus* (Maastrichtian) are consistent with that pattern. In the end, the author does conclude that the tooth specimens that he describes provide evidence for the refugium hypothesis, a conclusion that I agree with.

I suggest publication with minor revision; in addition to the specific comments below, I think the author should more clearly couch the article in terms of the refugium versus exchange models. In the introduction and conclusion, he leans heavily on the exchange model and that needs to be balanced, in both places, by a skeptical assessment of the case for the exchange model. An explicit statement of the evidence in favor of the refugium hypothesis must also be included in the abstract and the title must be adjusted accordingly.

The author may know my identity: Thomas D. Carr

Sincerely,
 Thomas D. Carr, PhD
 Associate Professor of Biology
 Carthage College
 2001 Alford Park Drive
 Kenosha, WI 53140
 p/262-551-5887

Specific comments:

Abstract: the phrase "fine detail" does not need to be hyphenated.
 The phrase "heavily mediolaterally compressed" is too wordy and misleading; the word "narrow" with a width to length ratio (in per cent) would suffice. My quibble is this: narrow teeth (w:l ratio less than 60%) is merely the plesiomorphic condition for tyrannosauroids. Since narrow teeth are the ancestral condition, there is no process of "compression" to account for their form. Replace "this fossil supports" with "this fossil is evidence for".
 Replace the word "common" with "widespread".

Introduction

Page 9, line 60: delete "rectangular".
 Page 10, lines 74 to 74: move "come" up to follow "have".
 Page 11, lines 97, 98: replace the em-dashes with "to".
 Page 11, line 111: replace "border" with "boundary".

Tyrannosauroid tooth

Identification

Page 13, line 147: insert a space between the parentheses.
 Page 13, lines 151-152: replace "highly mediolaterally compressed" with "narrow" and give a width to length ratio (mentioning, of course, that the tooth is incomplete and so the ratio is an overestimate).
 Page 13, line 155: replace "high mediolateral compression" with "narrow".
 Page 13, line 158: delete "pers. obs. of".
 Page 14, line 161: mention that the crown is incomplete and that you only have the apical third to half, otherwise your inability to take all measurements, as said here, is unclear.
 Page 14, line 169: insert a space between the parentheses.

Table 1: the specimen number "NJSMP GP 12456" does not match the number in the text (NJSMP GP 14256); which is correct?

Is there a caption that explains the abbreviations? If not, please include.
 Why is the basal width to basal length ratio not included?

Page 15, line 181: modern erosion or ancient erosion? Clarify.
 Page 15, line 183: a single tooth cannot be "zipodont", which is a term used for the entire dentition; "zipiform" is appropriate for a single tooth; replace "compressed" with "narrow".
 Page 15, line 184: in which direction is the mesial carina convex: labial or lingual? Please clarify.
 Page 15, line 185: replace "run" with "extent".
 Page 15: are the apical denticles worn down; i.e., through use, as produced the subtle wear facet on the lingual surface? Please include that observation, if you can.
 Page 15, line 193: replace "interspersed with diminutive" with "separated by small".

Figures: I did not receive any figure captions with the manuscript, so I cannot comment on those.

Dromaeosaurid tooth

Identification

Page 16, line 219: separate the parentheses with a space.

Page 17, line 223: explicitly state the crown height here.

Page 17, line 238: separate the parentheses with a space.

Page 18, lines 248, 251: placing a word or phrase between apostrophes implies that you are disowning what you are saying; since this clearly isn't the case, please delete them – they are distracting and self-contradictory. If you are insecure about using slender as a qualitative term, you could quantify the condition of slender with a ratio of, say, basal length to crown height or the like.

Page 18, line 249: give the ratio!

Page 18, line 252: is the term “western” necessary? Or do you mean exclusively Laramidia?

Clarify.

Page 18, line 254: give a ratio to quantify this contrast in slenderness.

Page 18, line 258: delete “conducted”.

Page 18, line 262: separate the parentheses with spaces.

Page 19, line 274: replace “and” with “from”.

Page 19, line 281: the name *B. feinbergi* should not be enclosed in apostrophes.

Page 19, line 288: replace “analysis” with “analyses”; delete “in which...included”.

Description

Page 20, line 295: replace “ziphodont” with “ziphiform”.

Page 20, line 308: replace “project” with “extend”.

Discussion

Page 21, line 315: insert “the” ahead of “Maastrichtian”.

Page 21, line 321: is “largish” a real word? Please fix!

Page 23: I'm skeptical of the claim for interchange – why can't the ceratopsian and lambeosaurine material simply reflect descendants of those clades before the inundation of the WIS? For example, lambeosaurines do have Santonian representatives in Asia. My impression is that the opportunities for exchange between Asia and Laramidia were generally episodic, whereas it was sustained during the late Maastrichtian. Regardless, the evidence for exchange between Laramidia and Appalachia isn't strong even though it is a worthwhile hypothesis to test.

Page 23, line 381: separate the parentheses with a space.

Page 24, line 395: argh – “largish” again! Please fix.

Methods

Page 25, line 406: change “provide support for the assignment” to “quantitatively test the identity”.

Page 26, line 429: change “provide additional support for” to “test”.

Figure 3: Perhaps it is in the caption, but please make it clear what the “ML” specimens are.

Author's Response to Decision Letter for (RSOS-191206.R0)

See Appendix E.

Decision letter (RSOS-191206.R1)

22-Oct-2019

Dear Mr Brownstein,

I am pleased to inform you that your manuscript entitled "New records of theropods from the latest Cretaceous of New Jersey and the Maastrichtian Appalachian fauna" is now accepted for publication in Royal Society Open Science.

Kind regards,

on behalf of the Associate Editor and Professor Kevin Padian (Subject Editor)
openscience@royalsociety.org

Appendix A

ROYAL SOCIETY OPEN SCIENCE

New records of theropods from New Jersey inform faunal interchange in Maastrichtian North America

Journal:	Royal Society Open Science
Manuscript ID	RSOS-190533
Article Type:	Research
Date Submitted by the Author:	26-Mar-2019
Complete List of Authors:	Brownstein, Chase; Stamford Museum and Nature Center,
Subject:	Palaeontology < EARTH SCIENCES, evolution < BIOLOGY, ecology < BIOLOGY
Keywords:	Dinosaur, Biogeography, Appalachia, Cretaceous
Subject Category:	Biology (whole organism)

**Author-supplied statements**

Relevant information will appear here if provided.

***Ethics***

*Does your article include research that required ethical approval or permits?:*

This article does not present research with ethical considerations

*Statement (if applicable):*

CUST_IF_YES_ETHICS :No data available.

***Data***

*It is a condition of publication that data, code and materials supporting your paper are made publicly*
*available. Does your paper present new data?:*

Yes

*Statement (if applicable):*

The data is included in the supplementary material.

***Conflict of interest***

I/We declare we have no competing interests

*Statement (if applicable):*

CUST_STATE_CONFLICT :No data available.

***Authors' contributions***

I am the only author on this paper

*Statement (if applicable):*

CUST_AUTHOR_CONTRIBUTIONS_TEXT :No data available.

**New records of theropods from New Jersey inform faunal interchange in Maastrichtian**
**North America**

Chase Doran Brownstein

Research Associate, Dept. of Collections and Exhibitions, Stamford Museum and Nature Center,
Stamford, CT, chasethedinosaur@gmail.com

Abstract.

The faunal changes that occurred in the few million years before the Cretaceous-Paleogene
extinction are of much interest to vertebrate paleontologists. Western North America preserves
arguably the best fossil record from this time, whereas terrestrial vertebrate fossils from the
eastern portion of the continent are usually limited to isolated, eroded postcranial remains. Here,
I report on two theropod teeth from the Mount Laurel Formation, a lower-middle Maastrichtian
unit from northeastern North America. One of these preserves in fine-detail the structure of the
outer enamel and closely resembles the dentition of the tyrannosauroid *Dryptosaurus*
*aquilunguis*. The other is assignable to a dromaeosaurid and represents both the youngest
occurrence of a non-avian maniraptoran in eastern North America and the first from the
Maastrichtian reported east of the Mississippi. Unlike other dromaeosaurid teeth from the
northeastern portion of the continent, the Mt. Laurel specimen is closely comparable with
western North American ones. Taken together, these fossils suggest a complex biogeography for
dinosaurs in the American east. Although the tyrannosauroid tooth provides an additional record
of non-tyrannosaurid tyrannosauroids in the Maastrichtian of eastern North America, the
dromaeosaurid specimen indicates faunal interchange between eastern and western North
America began to take place as early as the early Maastrichtian, several million years earlier than
previous discoveries indicated. Along with recent research on coeval dispersals between Asia and
western North America, the description of the new Mt. Laurel teeth emphasizes that dinosaur
biogeography during the Maastrichtian was characterized by extensive faunal interchange.

Keywords: Dinosaur; Appalachia; Cretaceous; Biogeography; Tyrannosaur; Teeth

Introduction.

The extinction of the non-avian dinosaurs at the end of the Mesozoic Era is a topic that has continued to intrigue vertebrate paleontologists (e.g., Alvarez et al., 1980; Sloan et al., 1986; Sarjeant and Currie, 2001; Le Loeuff 2012, Brusatte et al., 2012, 2015; Sakamoto et al., 2016). However, a poor global terrestrial record from the Maastrichtian has hindered attempts to assess the diversity dynamics of important groups like the Dinosauria during this period (e.g., Le Loeuff, 2012; Brusatte et al., 2015). Western North America preserves arguably the most well-characterized vertebrate record from the last 20 million years of the Mesozoic Era (Brusatte et al., 2015), whereas that from the eastern portion of the continent is far more obscure. During the majority of the Late Cretaceous, eastern and western North America were separated, the former existing as a rectangular landmass called Appalachia. Appalachian dinosaur faunas included intermediate-grade tyrannosaurs (Carr et al., 2005; Brusatte et al., 2011), basal hadrosaurids and derived non-hadrosaurid hadrosauroids (Langston, 1960; Prieto-Márquez et al., 2006; Prieto-Márquez et al., 2016), nodosaurids (Gallagher, 1993; Burns and Ebersole, 2016), and ornithomimosaurids (Miller, 1967; Schwimmer et al., 1993; Weishampel and Young, 1996; Schwimmer et al., 2015; Brownstein, 2018a).

Despite the amount of knowledge of Cretaceous faunal change to be gleaned from the fossil record of Appalachia, the assemblages of this landmass have remained fundamentally understudied since the mid-19th century (e.g., Weishampel & Young, 1996; Gallagher, 1997; Weishampel, 2006; Brownstein, 2018a). The scarcity of terrestrial sedimentary units known from the eastern half of the United States has also contributed to the obscurity of Appalachian faunas compared to western North American ones (Weishampel and Young, 1996; Gallagher, 1997;

Weishampel, 2006). Only in the past few years have indications of complex faunal changes in the
latest Cretaceous of the American east come, and all from isolated, fragmentary finds. Although
a ceratopsian tooth from the uppermost Maastrichtian of Mississippi (Farke and Phillips, 2017)
and lambeosaurine bones from the upper Maastrichtian of New Jersey (Gallagher, 1993) have
hinted that faunal exchanges occurred between Appalachia and Laramidia following the
regression of the Western Interior Seaway in the latest Campanian-earliest Maastrichtian, the
timing of these events remains poorly constrained. Characterizing faunal dispersals leading up to
the K-Pg extinction is essential for modeling changes to terrestrial vertebrate communities that
occurred before the event. Thus, further sampling of enigmatic assemblages from the

In the Campanian-Maastrichtian of New Jersey, a set of formations corresponds to a period of transgressions and regressions of the Atlantic Ocean (e.g., Gallagher, 1993; Sugarman et al., 1995; Miller et al., 2004; Gallagher et al., 2014). The majority of these Cretaceous units are known for producing marine vertebrate and invertebrate fossils (Gallagher, 1993), although some, such as the Woodbury and New Egypt formations, are notable for producing some of the first partial dinosaur skeletons from the Americas (e.g., Leidy, 1858; Cope, 1866; Gallagher, 1993, Weishampel and Young, 1996; Gallagher, 1997). One of the most fossiliferous of these formations is the Mount Laurel Formation, which is either uppermost Campanian or lowermost Maastrichtian (Miller et al., 2004) and in New Jersey has produced the remains of several groups of dinosaurs, including hadrosaurs, tyrannosaurs, and ornithomimosaurids (Gallagher, 1993; Gallagher, 2014). Because of the sheer diversity of the community represented in the Mt. Laurel,

the formation serves as a window into Maastrichtian eastern North American faunas. However,
the terrestrial fossils it produces are often eroded postcranial fragments (Gallagher, 2014).

Here, I describe some theropod teeth from the Mt. Laurel Formation of New Jersey.
These include a large tooth assignable to a mid-sized (est. 6—8 m) tyrannosauroid and a smaller,
heavily recurved one assignable to a smallish (est. 2—3 m) dromaeosaurid. These teeth are
among the most diagnostic records of theropods from the Mt. Laurel Formation, allowing for a
more precise understanding of the faunal composition and ecology of the eastern seaboard during
the Maastrichtian, a globally under-sampled time period (Le Loeuff, 2012; Brusatte et al., 2012).

The dromaeosaurid tooth, representing the youngest occurrence of a non-avian maniraptoran in
the American east, is very closely allied with Maastrichtian western North American
saurornitholestines, suggesting Laramidian maniraptorans dispersed into eastern North America
as early as the latest Campanian. The tyrannosauroid tooth is comparable to the dentition of
*Dryptosaurus aquilunguis*, providing the first strong morphological support for the presence of
*Dryptosaurus* or a closely related form in the Mt. Laurel ecosystem and representing a probable
additional record of phylogenetically-intermediate tyrannosaurs in eastern North America. These
specimens indicate that Appalachian vertebrate faunas were already becoming heterogenous
during the early Maastrichtian, five million years earlier than previous discoveries have
indicated. Along with other finds in North America, Europe, and Asia, these Mt. Laurel theropod
teeth indicate complex interchanges occurred across northern hemisphere only a few million
45 years before the end of the Mesozoic Era.

**Results.**

Geological setting.

Both theropod teeth described here were collected from sediments of the Mount Laurel
Formation (Gallagher, 1993; pers. obs.), a marine deposit that represents a regression of the
Atlantic Ocean during the Late Cretaceous period and is the oldest unit included in the
Monmouth Group (Gallagher, 1993; Miller et al., 2004). The tyrannosauroid tooth described
here, NJSM GP 12456, was recovered from Big Brook (Fig. 1A), a highly fossiliferous locality
famous for producing an extensive marine fauna (Gallagher, 1993; Weishampel and Young,
1996). At Big Brook, the stratigraphic column is exposed along the banks, with the Mt. Laurel
smoothly transitioning from the underlying Wenonah Formation such that the border between the
two are indistinguishable (Gallagher, 1993). The contact between the Mt. Laurel and the
overlying Navesink Formation is highly irregular (Miller et al., 2004; Gallagher et al., 2014). The
23 Mt. Laurel Formation appears as gray to dark brown, pebbly quartz sands. The Big Brook
tyrannosauroid tooth (Fig. 1E—H) is unusual among the terrestrial vertebrate teeth collected
from the site in possessing a well-preserved enamel surface. Whereas other terrestrial vertebrate
fossils from Big Brook are known for being heavily water-worn and lacking morphological
details, NJSM GP 12456 preserves both its outermost enamel layer and many of its denticles.

NJSM GP 22949, the dromaeosaurid tooth, was recovered from Mt. Laurel deposits in
Burlington County, New Jersey (Fig. 1A). In this area, which makes up a portion of the
southwestern-most range of the Monmouth Group, the sands of the Mt. Laurel are more
glauconitic than farther north and are intermixed with iron compounds (Gallagher, 1993). The
thickness of this unit is also far greater to the southwest of its range (e.g., Gallagher, 1993).

Tyrannosauroid tooth.

Systematic Paleontology.
Dinosauria Owen 1842

Theropoda Marsh 1881

Coelurosauria von Huene 1921

Tyrannosauroida Osborn 1905

Tyrannosauroida indet., cf. *Dryptosaurus aquilunguis* Cope 1866

Material.

New Jersey State Museum collections (NJSM) GP 14256, the partial tooth of a large theropod
dinosaur (Fig. 1E—H).

Locality and Horizon.

25 Mt. Laurel Formation sediments at Big Brook, Monmouth County, New Jersey, latest Campanian
to early Maastrichtian (Gallagher, 1993; Miller et al., 2004; Gallagher et al., 2014).

Identification.

NJSN GP 14256 (Fig. 1E—H) closely resembles the dentition of tyrannosauroid
theropods in several respects. The tooth resembles those of adult tyrannosauroids in its size,
which is closely comparable to tyrannosaur crowns known from both western and eastern North
America (Currie et al., 1990; Brochu, 2003; Samman et al., 2005; Smith, 2005; Brusatte et al.,
2011; Williamson and Brusatte, 2014). In addition to its size, the Mt. Laurel tooth resembles
those of tyrannosaurs among theropods known from Late Cretaceous North America in
possessing packed denticles (2–2.5/mm) on its distal carina (15+ mm), the presence of denticles
along both carinae, the straightness of its distal margin, the presence of numerous transverse
undulations (density = 2/mm) on its main surface, and its smooth but irregular surface texture
(Fig 1E—H)(Currie et al., 1990; Brochu, 2003; Samman et al., 2005; Smith, 2005; Brusatte et

al., 2011; Williamson and Brusatte, 2014). However, despite the size of the Mt. Laurel tooth,
which indicates the ontogenetic maturity of the individual it represents, NJSM GP 14256 is
notably unlike the teeth of tyrannosaurids, which possess incrassate teeth with D-shaped basal
cross-sections (e.g., Williamson and Brusatte, 2014). Instead, NJSM GP 14256 shows is
highly mediolaterally compressed and possesses a lens-shaped basal cross-section, indicative it
came from a more basal tyrannosaur outside Tyrannosauridae. Among large Late Cretaceous
tyrannosaurs, only *Dryptosaurus aquilunguis* from the Maastrichtian New Egypt Formation of
New Jersey is known to possess this combination of features in its teeth (e.g., Brusatte et al.,
2011). NJSM GP 14256 is also closely comparable with the mediolaterally compressed teeth of
*Dryptosaurus aquilunguis* in its dimensions, curvature, and prominence and density of its
denticles (1.7–1.9 mm in *Dryptosaurus*) and enamel crenulations (Brusatte et al., 2011; pers.
obs. of YPM PU 22208). Given the Mt. Laurel tooth's very close spatiotemporal proximity to the
holotype of *Dryptosaurus*, I suggest the former belongs to *D. aquilunguis* or a close relative.

The principle components analysis including NJSM GP 14256 in a modified version of
the dataset of Smith et al. (2005) found this tooth to cluster more closely to those of
tyrannosauroid theropods than to dromaeosaurids and troodontids (Fig. 2). Furthermore,
phylogenetic analysis of the tooth within the dataset of Hendrickx and Mateus (2014) found
NJSM GP 14256 to be the sister taxon of *Tyrannosaurus rex* in a clade united by four characters.
These are characters 94 (biconvex apical denticles present on distal carinae of lateral teeth), 100
(subequal number of denticles apically than at mid-crown portion of distal carinae on lateral
teeth), 103 (interdenticular space between mid-crown denticles on distal carinae of lateral teeth
broad), and 105 (interdenticular sulci between mid-crown denticles on distal carinae of lateral

teeth present, long, and well-developed)(Hendrickx and Mateus, 2014). The clade comprised of
NJSM GP 14256 and other derived tyrannosaurs (*Alioramus*, *Tyrannosaurus*, “*Raptorex*”) is
united by the presence of a sub-symmetric crown with a centrally-positioned distal carina in
distal view (char. 83). The strict consensus tree (tree length = 688, consistency index = 0.340,
retention index = 0.561) is shown in Fig. 1B.

Description.

NJSM GP 14256 (Fig. 1) is the apical half of the tooth of a theropod dinosaur.
Measurements of the specimen may be found in Table 1. The tooth is well-preserved for a
terrestrial fossil collected from one of the marine deposits of the Cretaceous Atlantic Coastal
Plain, preserving details of the outer enamel layer and denticle morphology. Unfortunately, the
basal half of the crown and the entirety of the root of the tooth are not preserved. This is
probably due to erosion, as the tooth is broken transversely and heavily rounded at its preserved
base (Fig. 1E—F).

The tooth displays the ziphodont condition in being labiolingually compressed and only
slightly recurved. The preserved mesial carina is slightly convex, whereas the distal carina is
straightened along its entire run. The labial and lingual portions of the enamel are well-preserved
(Fig. 1G), bearing developed transverse undulations that develop out of the distal margin of the
tooth to become bands (Brusatte et al. 2007). In NJSM GP 14256, these undulations (= marginal
bands) are relatively strongly developed, although they are less prominent than in
carcharodontosaurids (e.g., Sereno et al., 1996; Brusatte et al., 2007). The labial and lingual
surfaces of the tooth are slightly convex, as in most other theropod dinosaurs (Hendrickx et al.,

2015a). The apex of the tooth bears a slight wear facet on its lingual surface. The tooth is lenticular in basal cross-section.

The distal carina preserves many denticles (Fig. 1E—F, H), which are small, dense (6/mm), and apicobasally straightened. The denticles are and are interspersed with diminutive interdenticular sulci (the “blood grooves” of Currie et al. (1990)). These are encompassed by the apical ends of the denticles. These denticles maintain a similar density along

Table 1. Measurements of teeth described in this study (in/per mm).

Specimen	CH	CBL	CBW	AL	CA	MB	MC	MA	DB	DC	DA
NJSM GP 12456	15 (est. 25)	8.99	4.90						2	2	2.5
NJSM GP 22949	15.5	8.0	2.0	18.00	55.5				6	5	5

the entirety of the distal carina. However, their density may have changed along the missing portion of the tooth. The mesial carina preserves a few denticles, although these are too eroded for much morphological description. These denticles appear to be similar in size to those on the distal carina.

Dromaeosaurid tooth.

Systematic Paleontology.

Dinosauria Owen 1842

Theropoda Marsh 1881

Coelurosauria von Huene 1921

Maniraptora Gauthier 1986

Dromaeosauridae Matthew & Brown 1922

Sauornitholestinae Sues 1978

Sauornitholestinae indet.

Material.

NJSM GP 22949, the tooth of a mid-sized theropod dinosaur.

Locality and Horizon.

18 Mt. Laurel Formation sediments in Burlington County, New Jersey, latest Campanian to early
Maastrichtian (Gallagher, 1993; Miller et al., 2004; Gallagher et al., 2014).

Identification.

NJSM GP 22949 is identified as the lateral tooth of a dromaeosaurid theropod based on
the following combination of features: (1) its extreme apicobasal curvature created by its
concave distal carina and distally offset apex, (2) the presence of apically hooked distal denticles,
(3) the absence of mesiodistal constriction along the crown base, and (4) distal denticles that
decrease in size towards the apex of the tooth (Fig. 3A—E)(Currie et al., 1990; Turner et al.,
2012; Larson and Currie, 2013; Williamson and Brusatte, 2014). NJSM GP 22949 is smaller than
the majority of Appalachian theropod teeth assigned to tyrannosauroids, in which crown heights
surpass 50 mm (Schwimmer et al., 1993; Carr et al., 2005; Brusatte et al., 2011; Denton et al.,
2011; Schwimmer et al., 2015). However, the tooth is somewhat larger than most North
American dromaeosaurid teeth, which are often less than 10 mm in height (Ostrom, 1969; Currie
et al., 1990; Currie and Varricchio, 2004; Larson and Currie, 2013; Williamson and Brusatte,
2014; Wick et al., 2015). *Dakotaraptor*, which possessed crowns up to ~25 mm high, represents
the exception among Maastrichtian forms (Palma et al., 2015). NJSM GP 22949 is also

distinguished from Appalachian tyrannosauroids in lacking subquadrangular distocentral
denticles (Hendrickx et al., 2015). The first principle components analysis on the Mt. Laurel
dromaeosaurid tooth supports this hypothesis by placing the specimen closer to the convex hulls
formed by dromaeosaurid teeth than to those formed by tyrannosauroids (Fig. 4).

NJSM GP 22949 is notable for being very similar to the teeth of western North American
saurornitholestine dromaeosaurids (Fig. 3A—E, G)(Larson and Currie, 2013) and unlike those
previously discovered from the American east (Kiernan and Schwimmer, 2004; Schwimmer et
al., 2015; Brownstein, 2018a). Teeth assigned to “cf. *Saurornitholestes*” have been described
from the Cretaceous of the southeastern United States (Kiernan and Schwimmer, 2004;
Schwimmer et al., 2015), but these are extremely small (< 6 mm), far less recurved than NJSM
GP 22949, and have proportionally large denticles that are more strongly apically hooked (e.g.,
fig. 1 in Kiernan and Schwimmer, 2004). One tooth from Alabama measuring 4.9 mm in crown
height and preserving 7 distal denticles and 8 mesial denticles per mm is less recurved than
NJSM GP 22949 and far less slender (see Kiernan and Schwimmer, 2004). Some
‘saurornitholestine’ teeth from South Carolina (Schwimmer et al., 2015) also lack the ‘slender’
condition in NJSM GP 22949, where the width of the heavily recurved tooth crown steadily
shortens towards the apex to produce a tall, mesiodistally shortened crown. All these teeth are
very similar to dromaeosaurid crowns described from the mid-Cretaceous of Idaho
(Krumenacker et al., 2016) and Utah (Frederickson et al., 2018). Teeth from the Ellisdale site of
New Jersey include dromaeosaurid crowns (Denton et al., 2011; Brownstein, 2018c). However,
large crowns from Ellisdale are not ‘slender’ like NJSM GP 22949, are much larger in size, and
bear distal denticles that are considerably more apically hooked than the Mt. Laurel tooth and

western saurornitholestines (Brownstein, 2018c). A dromaeosaurid tooth from North Carolina is
slightly larger than NJSM GP 22949, but is less recurved and far less slender (Brownstein,
2018b).

When compared to dromaeosaurid teeth from outside eastern North America, NJSM GP
22949 most closely resembles the teeth of western North American saurornitholestine
dromaeosaurids. Unlike the western North American velociraptorine *Acheroraptor temertyorum*
(Evans et al., 2013) and the Mongolian *Velociraptor mongoliensis* (Turner et al., 2012; Fig. 3F),
NJSM GP 22949 lacks strongly developed striations along its crown surface, is more strongly
recurved, and is far more slender. In contrast to *Utahraptor ostrommaysorum*, *Dromaeosaurus*
*albertensis*, and ‘dromaeosaurine’ teeth from western North America, the mesial carina in NJSM
GP 22949 is not twisted onto the mesiolingual face of the crown, the distal denticles are apically
hooked, and the tooth is more strongly recurved (Fig. 3H—I; Turner et al., 2012; Larson and
Currie, 2013). The Mt. Laurel tooth is also smaller, much more strongly recurved, and possesses
denticles more apically hooked than those of the giant Maastrichtian dromaeosaurid
*Dakotaraptor steini* (DePalma et al., 2015). NJSM GP 22949 lacks the ‘figure-8’ basal cross-
section seen in the teeth of *Deinonychus* (Ostrom, 1969; Brownstein, 2018b; pers. obs.).
Although the strongly recurved maxillary teeth of *Deinonychus* (Ostrom, 1969; Turner et al.,
2012; pers. obs.) are somewhat comparable with NJSM GP 22949, the differing basal cross-
sections among these specimens and the slightly asymmetrical morphology of the teeth in
*Deinonychus* distinguish *D. antirrhopus* and the Mt. Laurel form.

NJSM GP 22949 closely resembles the teeth of western North American Maastrichtian
saurornitholestines in having a slender, tall outline in labial and lingual views (the “Lancian”

saurornitholestine morphotype of Larson and Currie, 2013). The tooth is closely comparable with
the crowns of the juvenile saurornitholestine ‘*Bambiraptor feinbergi*,’ which are extremely
recurved and slender and possess apically-hooked denticles (Fig. 3G). A principle components
analysis of the Late Cretaceous western North American paravian tooth dataset of Larson and
Currie (2013) found NJSM GP 22949 to nest within the convex hulls formed by four tooth
morphotypes (Saurornitholestinae, Dromaeosaurinae, *Zapsalis*, and *Atrociraptor*). However,
unlike in teeth assigned to *Dromaeosaurus* or other ‘dromaeosaurines,’ the mesial carina in
NJSM GP 22949 is not twisted onto the mesiolingual face of the crown and the distal denticles
are apically hooked (Fig. 3H—I; Larson and Currie, 2013). NJSM GP 22949 is also quantifiably
unlike YPM VPPU.021397, the large dromaeosaurid tooth from the Campanian of North
Carolina (Brownstein, 2018b), plotting far from the southeastern North American specimen in
both morphometric analysis in which these teeth were included (Fig. 4). Thus, NJSM GP 22949
is identifiable as the crown of a dromaeosaurid with affinities to coeval western
saurornitholestines.

36 37 Description.

NJSM GP 22949 is the complete crown of a dromaeosaurid dinosaur. Measurements of
this specimen are in Table 1. This tooth is heavily recurved, displaying the zipodont condition
present in the teeth of other Appalachian theropods (e.g., Brusatte et al. 2011; Schwimmer et al.,
2015). The crown is also mediolaterally compressed and possesses an ovoid basal cross-section.
In distal view, the middle portion of tooth is convex labially and distally, although the crown
becomes labiolingually straightened towards its apex. The labial and lingual surfaces are
flattened, and the lack of a root attached to this crown indicates it was shed. Although both the

mesial and distal carinae are preserved, the mesial denticles have been eroded away. Some
portions of the tooth crown are cracked, and the outer enamel layer is poorly preserved towards
the distal end of the specimen. Small portions of the middle of the crown are missing. The distal
profile of NJSM GP 22949 is strongly concave and more developed in basal cross-section than
the mesial. The preserved portions of the outer enamel layer are smooth, although at the apex
several slightly developed ridges appear. These ridges could represent features of the original
morphology of the tooth or be damage from feeding or taphonomic processes.

The distal carina preserves a large number of apically hooked denticles that become
smaller towards the apex of the crown. These denticles are separated by interdenticular sulci that,
along with the serrations, project slightly onto the tooth surface. Unfortunately, the shape and
density of the mesial denticles could not be determined, as the mesial carina is heavily eroded in
NJSM GP 22949.

**Discussion.**

The two theropod teeth described here add to one of the most complete Maastrichtian
faunas from eastern North America. The dromaeosaurid tooth NJSM GP 22949 is the first
occurrence of this clade in the Mount Laurel Formation and more generally Maastrichtian of
eastern North America. Until now, only tyrannosauroid and ornithomimosaur remains have been
reported from this area (Gallagher, 1993; Weishampel & Young, 1996; Gallagher, 1997; Brusatte
et al., 2011; Brusatte et al., 2012; Gallagher et al., 2014), with the latest records of
dromaeosaurids in the American east hailing from mid-Campanian units in the Carolinas
(Schwimmer et al., 2015; Brownstein, 2018b) and the Ellisdale site of New Jersey (Denton et al.,
2011). Although NJSM GP 22949 is most comparable to the crowns of mid-sized to largish

dromaeosaurids like *Deinonychus antirrhopus* and *Dakotaraptor steini* (Ostrom, 1969; DePalma
et al., 2015) and small tyrannosauroids (Denton et al., 2011; Williamson and Brusatte, 2014;
Schwimmer et al., 2015) in its dimensions, it is most closely allied with dromaeosaurids in the
morphometric analyses conducted (Fig. 4) and in several key features of its morphology. Indeed,
this tooth is of great biogeographical significance for being the youngest record of a non-avian
maniraptoran in the eastern half of the North American continent, greatly increasing theropod
diversity in this area.

The tyrannosauroid tooth NJSM GP 14256 supports the presence of *Dryptosaurus*-like
tyrannosauroids in the early Maastrichtian of New Jersey. Isolated teeth and postcranial
fragments from the Mount Laurel were previously assigned to *Dryptosaurus* sp. based on little
more than their geographic proximity to the site where the holotype of this taxon was recovered
(Gallagher, 1993; Weishampel & Young, 1996; Gallagher, 1997), but no detailed description of
early Maastrichtian tyrannosauroids from New Jersey has appeared in the literature. The well-
preserved nature of NJSM GP 14256 thus allows for the formal recognition of the presence of
intermediate-grade tyrannosauroids in the earliest Maastrichtian of the Atlantic Coastal Plain.
Furthermore, the excellent condition of the outer enamel layer of NJSM GP 12456 allows for
further documentation of the dental anatomy of Appalachian tyrannosauroids, the isolated teeth
of which are often found highly abraded among stream deposits (e.g., Weishampel & Young,
1996). The presence of non-tyrannosaurid tyrannosauroids in the Mt. Laurel Formation was
expected, given the presence of *Dryptosaurus aquilunguis* and non-tyrannosauroid tyrannosaurs
of similar phylogenetic position in both the middle-late Maastrichtian Navesink and New Egypt
formations (Gallagher, 1993; Brusatte et al., 2011) and Campanian Marshalltown Formation

(Denton et al., 2011). However, that NJSM GP 14256, originally discovered in 1984, is only
described now attests to the understudied nature of these deposits.

The late recognition of dromaeosaurids in the Maastrichtian sediments of New Jersey is
somewhat more intriguing, given that the dinosaurs of the Mt. Laurel and other Cretaceous units
in the Atlantic Coastal Plain have been studied for over a century and a half (e.g., Leidy, 1858;
Cope, 1866; Gallagher, 1993; Weishampel & Young, 1996; Gallagher, 1997). Teeth from
locations like the Ellisdale site of the Marshalltown Formation of New Jersey originally assigned
to tyrannosaurs have more recently been reclassified as the crowns of dromaeosaurids (Gallagher
et al., 1986; Gallagher, 1993, 1997; Denton et al., 2011), so it is entirely possible that the lack of
diversity in dinosaur faunas from the Maastrichtian of the Atlantic Coastal Plain reflects
systematic misidentification of these isolated fossils. Only further work on Appalachian fossils
will allow for more comprehensive revision of the identification of fossils from this area.

On a larger biogeographical scale, the two teeth described here are important for
illuminating early trends to faunal homogenization in the latest Cretaceous of the northern
hemisphere. The trend in faunal composition for the majority of the Cretaceous was evolution in
isolation, which coincided with the breakup of supercontinents like Gondwana and Laurasia and
the inundation of smaller landmasses like North America and Europe (e.g., Russell, 1995; Sereno
et al., 1996; Csiki et al., 2010; Sampson et al., 2010; Csiki-Sava et al., 2015). In particular, the
faunas of Appalachia and Laramidia, which became isolated from each other as the Western
Interior Seaway flooded the middle of North America (e.g., Russell, 1995; Roberts and
Kirschbaum, 1997; Schwimmer, 2002; Sampson et al., 2010), have been recognized as highly
distinctive (e.g., Gallagher, 1993; Weishampel and Young, 1996; Gallagher, 1997; Sampson et

al., 2010; Denton et al., 2011; Schwimmer et al., 2015). However, in the past thirty years, a
handful of discoveries from the eastern margin of the North American continent have indicated
some faunal interchange occurred between Laramidian and Appalachian dinosaur communities
during the latest Maastrichtian. These include the tooth of a ceratopsid from the latest
Maastrichtian of Mississippi and the forelimb material of possible lambeosaurines from the
Maastrichtian of New Jersey and earliest Maastrichtian of Nunavut, Canada (Gallagher, 1995;
Gallagher, 1997; Farke and Phillips, 2017). The discovery of a dromaeosaurid tooth in the latest
Campanian-earliest Maastrichtian Mt. Laurel Formation with affinities to western North
American saurornitholestine dromaeosaurids is particularly interesting in the context of these
other finds, indicating such faunal dispersals could have occurred as early as the late Campanian.
At the same time, Appalachian faunas continued to harbor endemic forms like intermediate
tyrannosauroids, represented by *Dryptosaurus aquilunguis* and comparable forms through the
Campanian-Maastrichtian transition in the Atlantic Coastal Plain (Gallagher, 1993; Brusatte et
al., 2011; this paper).

Faunal homogenization in the last 10 million years of the Mesozoic seems to have
occurred throughout the northern hemisphere. The presence of *Tyrannosaurus rex* in the
Maastrichtian of the western United States and Canada seems to represent a dispersal of Asian
tyrannosaurids into the Americas (Brusatte et al., 2010; Brusatte and Carr, 2016). A dispersal
event between Asian and North Americans faunas during the Maastrichtian may have occurred
over Beringia (Fiorillo, 2008), given the similarity of Maastrichtian polar faunas from Russia,
Alaska, and lower latitudes in the United States and Canada (e.g., Weishampel et al., 2004;
Gangloff, 2012; Godefroit et al., 2012; Le Loeuff, 2012). The new teeth illuminate the presence

of this trend in the Campanian-Maastrichtian of the Atlantic Coastal Plain, showing that the
transition from a solely endemic Appalachian fauna to one including immigrant genera occurred
during the beginning of the closure of the Western Interior Seaway (Schwimmer, 2002; Farke
and Phillips, 2017; Brownstein, 2018a).

**Conclusions.**

Two theropod teeth from the Campanian-Maastrichtian Mount Laurel Formation of New
Jersey preserve the detailed morphology of their denticles and enamel surfaces, allowing for their
assignment to dromaeosaurids and tyrannosauroids. The dromaeosaurid tooth, which plots with
western North American saurornitholestine teeth in principle components and discriminant
analyses, is the youngest record of a non-avian maniraptoran from eastern North America and the
first from the Maastrichtian of the American east. Along with the tyrannosauroid tooth, which is
the first specimen to strongly suggest the presence of *Dryptosaurus aquilunguis* or a closely
related tyrannosauroid in the Mt. Laurel ecosystem, this dromaeosaurid crown provides new
information in faunal interchanges in the latest Cretaceous of the northern hemisphere, pushing
back the dispersal of western North American taxa into the eastern half of the continent into the
Campanian-Maastrichtian boundary.

**Methods.**

Measurements and nomenclature.

Measurements of both teeth were taken in accordance with the methodology of Smith et al.
(2005) and Larson and Currie (2013). The dimensions of the Mt. Laurel teeth were determined
using digital calipers. I follow the nomenclature of Hendrickx et al. (2015) when describing the
two teeth on which this paper focuses.

Principle components analyses.

In order to provide statistical support for the assignment of the two described teeth to
specific groups of theropod dinosaurs, I included them in principle components analyses
conducted in the program PAST v. 3.18 (Hammer et al., 2001). In order to assess the
morphological similarity of the Mt. Laurel teeth to theropod clades present in the Cretaceous of
the northern hemisphere, I used a modified version (Brownstein, 2018b) of the dataset of Smith
et al. (2005) that includes tooth data on tyrannosauroids, troodontids, and dromaeosaurids. A
principle components analysis was run on this dataset, which included data on fifteen
measurements: crown height (CH), crown base length (CBL), crown base width (CBW),
apicobasal length (length of the tooth along the longest apicobasal axis), serration density per
5mm for the basal (MB), mid-crown (MC), and apical (MA) mesial carina, and the same for the
distal carina (DB, DC, DA). An additional principle components analysis (PCA) was conducted
using the dataset of Larson and Currie (2013) in order to better assess the similarity of the Mt.
Laurel dromaeosaurid specimen to other North American paravian teeth. This principle
components analysis assessed for five measurements: CH, CBL, CBW, and the mesial (MD) and
distal (DD) denticles per millimeter. The summary statistics and loadings from the results of the
PCAs conducted are included in the Supplementary Information.

Discriminant analyses.

To further assess the affinities of the Mt. Laurel dromaeosaurid tooth, I performed a discriminant
analysis on the tooth datasets of Smith et al. (2005, modified in Brownstein, 2018b) and Larson
and Currie (2013). This analysis creates a morphospace by maximally separating objects sorted

into pre-determined groups. This analysis was also run in PAST v. 3.18 (Hammer et al., 2001),
and the summary statistics can be found in the supplementary information.

**Phylogenetic analysis.**

To provide additional support for the referral of the incomplete tyrannosauroid tooth to that
family, I coded the specimen for the phylogenetic matrix of Hendrickx and Mateus (2014), a
dataset of theropod dentition that includes 64 taxa/specimens coded for 141 characters. The
matrix was entered into the phylogenetics program TNT 1.5 (Goloboff and Catalano, 2016) for a
phylogenetic analysis. The matrix was first analyzed using the “New Technology Search,” with
default parameters for ratchet, tree drift, tree fuse, and sectorial search. A total of 10 trees of
length 688 were retained. These topologies were then subjected to traditional (TBR) branch
swapping, which allows for a more extensive exploration of each tree island. This found over
>99,999 most parsimonious topologies of 688. These were summarized in a strict consensus
topology.

**Acknowledgements.**

I thank Dana Ehret for access to the collections of the New Jersey State Museum, Carl Mehling
for access to the collections of the American Museum of Natural History, and Dan Brinkman for
access to the collections of the Yale Peabody Museum of Natural History. I thank the editor
Howard Falcon-Lang, Christophe Hendrickx, and three anonymous reviewers for their
comments, which greatly improved the quality of this paper.

**Availability of data and material.**

All data is available in the Supplementary Information.

**Competing interests.**

The author declares no competing interests.

**Funding.**

The author received no funding for this work.

**References.**

Alvarez, L.W., Alvarez, W., Asaro, F., Michel, H.V. 1980. Extraterrestrial cause for the
Cretaceous–Tertiary extinction. *Science* **208**:1095–1108.

Brochu, C.A. 2003. Osteology of *Tyrannosaurus rex*: insights from a nearly complete skeleton
and high-resolution computed tomographic analysis of the skull. *Society of Vertebrate*
*Paleontology Memoir* **7**:1–138.

Brownstein, C.D. 2018a. The biogeography and ecology of the Cretaceous non-avian dinosaurs
of Appalachia. *Palaeontologia Electronica* **21.1.5A**:1–56.

Brownstein, C.D. 2018b. A large dromaeosaurid from North Carolina. *Cretaceous Research* **92**:1
—7.

Brownstein, C.D. 2018c. The distinctive theropod assemblage of the Ellisdale site of New Jersey
and its implications for North American dinosaur ecology and evolution during the Cretaceous.
*Journal of Paleontology* **92**(5):1115—1129.

Brusatte, S.L., Benson, R.B., Norell, M.A. 2011. The Anatomy of *Dryptosaurus aquilunguis*
(Dinosauria: Theropoda) and a Review of its Tyrannosauroid Affinities. *American Museum*
*Novitates* **3717**:1–53.

Brusatte, S.L., Benson, R.B.J., Carr, T.D., Williamson, T.E., and Sereno, P.C. 2007. The
systematic utility of theropod enamel wrinkles. *Journal of Vertebrate Paleontology* **27**:
1052-1056.

Brusatte, S.L., Butler, R.J., Barrett, P.M., Carrano, M.T., Evans, D.C., Lloyd, G.T., Mannion,
P.D., Norell, M.A., Peppe, D.J., Upchurch, P., and Williamson, T. E. 2015. The extinction of the
dinosaurs. *Biological Reviews* **90**:628–642.

Brusatte, S.L., Butler, R.J., Prieto-Márquez, A., Norell, M.A. 2012. Dinosaur morphological
diversity and the end-Cretaceous extinction. *Nature Communications* **3**:804.

Brusatte, S.L., Carr, T.D. 2016. The phylogeny and evolutionary history of tyrannosauroid
dinosaurs. *Scientific Reports* **6**(1):20252.

Brusatte, S.L., Norell, M.A., Carr T.D., Erickson, G.M., Hutchinson, J.R., Balanoff, A.M.,
Bever, G.S., Choiniere, J.N., Makovicky, P.J., Xu X. 2010. Tyrannosaur paleobiology: new
research on ancient exemplar organisms. *Science* **329**(5998):1481-1485.

Burns, M.E., and Ebersole, J.A. 2016. Juvenile Appalachian nodosaur material (Nodosauridae,
Ankylosauridae) from the lower Campanian lower Mooreville Chalk of Alabama. Abstracts and
Posters Session IV, Society of Vertebrate Paleontology, 76th Annual Meeting.

Carr, T.D., Williamson, T.E., Schwimmer, D.R. 2005. A new genus and species of tyrannosauroid
from the Late Cretaceous (middle Campanian) Demopolis Formation of Alabama. *Journal of*
*Vertebrate Paleontology* **25**(1):119–143.

Cope, E.D. 1866. Discovery of a gigantic dinosaur in the Cretaceous of New Jersey. *Proceedings*
*of the Academy of Natural Sciences of Philadelphia* **18**: 275–279.

Csiki-Sava, Z., Buffetaut, E., Ósi, A., Pereda-Suberbiola, X., and Brusatte, S.L. 2015. Island life
in the Cretaceous - Faunal composition, biogeography, evolution, and extinction of land-living
vertebrates on the Late Cretaceous European archipelago. *ZooKeys* **469**:1–161.

Csiki, Z., Vremir, M., Brusatte, S.L., Norell, M. A. 2010. An aberrant island-dwelling theropod
dinosaur from the Late Cretaceous of Romania. *Proceedings of the National Academy of*
*Sciences* **107**(35):15357–15361.

Currie P.J., Rigby J.K., Sloan R.E., 1990, Theropod teeth from the Judith River Formation of
Southern Alberta, Canada. In: Carpenter, K., and Currie, P.J., eds: *Dinosaur Systematics:*
*Approaches and Perspectives*. Cambridge University Press, p. 107–125.

Currie, P. J. and Varricchio, D.J. 2004. A new dromaeosaurid from the Horseshoe Canyon
Formation (Upper Cretaceous) of Alberta, Canada. In: Currie, P.J., Koppelhus, E.B., Shugar,
10 M.A., and Wright, J.L., eds: *Feathered Dragons*. Indianapolis: Indiana University Press. p. 112–
132.

Denton R.K., O’Neill R.C., Grandstaff B.S., and Parris D.C. 2011. The Ellisdale Site (Late
Cretaceous, Campanian) - is there a rationale for an “Ellisdalean” land faunal age? *Geological*
*Society of America Abstracts with Programs* **43**:85.

DePalma R.A., Burnham D.A., Martin L.D., Larson P.L., and Bakker R.T. 2015. The First Giant
Raptor (Theropoda: Dromaeosauridae) from the Hell Creek Formation. *Paleontological*
*Contributions* **14**:1–16.

Farke, A.A. and Phillips, G.E. 2017. The first reported ceratopsid dinosaur from eastern North
America (Owl Creek Formation, Upper Cretaceous, Mississippi, USA). *PeerJ* **5**:e3342.

Fiorillo, A.R. 2008. Cretaceous dinosaurs of Alaska: Implications for the origins of Beringia. In:
Blodgett, R.B., and Stanley, G. eds: *The Terrane Puzzle: new perspectives on paleontology and*
*stratigraphy from the North American Cordillera. Geological Society of America Special Paper*
**442**: 313-326.

Frederickson, J., Engel, M.H., Cifelli, R. 2018. Niche Partitioning in Theropod Dinosaurs: Diet
and Habitat Preference in Predators from the Uppermost Cedar Mountain Formation (Utah,
U.S.A.). *Scientific Reports* **8**(1):17872.

Gallagher, W.B. 1993. The Cretaceous-Tertiary mass extinction event in North Atlantic Coastal
Plain. *The Mosasaur* **5**:75-154.

Gallagher, W.B. 1995. Evidence of juvenile dinosaurs and dinosaurian growth stages in the Late
Cretaceous deposits of the Atlantic Coastal Plain. *Bulletin of the New Jersey Academy of Science*
**40**:5–8.

Gallagher, W.B. 1997. When Dinosaurs Roamed New Jersey. Rutgers University Press, New
Brunswick, New Jersey.

Gallagher, W.B., Camburn, J., Camburn, S., and Hanczaryk, P.A. 2014. Taphonomy of a Late
Campanian Fossil Assemblage at Marlboro, Monmouth County, New Jersey. *The Mosasaur*
**8**:53–68.

Gangloff, R.A. Dinosaurs under the Aurora. Bloomington, Indiana University Press, 2012.

Gauthier, J. 1986. Saurischian monophyly and the origin of birds. *Memoirs of the California*
*Academy of Sciences* **8**:1–55.

Godefroit, P., Bolotsky, Y.L., Lauters, P. 2012. A New Saurolophine Dinosaur from the Latest
Cretaceous of Far Eastern Russia. *PLoS ONE* **7**(5):e36849.

Goloboff, P., Catalano, S., 2016. TNT version 1.5, including full implementation of phylogenetic
morphometrics. *Cladistics* **32**(3): 221-238.

Hammer, Ø., Harper, D.A.T., Ryan, P. D. 2001. Paleontological statistics software package for
education and data analysis. *Palaeontologia Electronica* **4**:1–9.

Hendrickx, C., Mateus, O., 2014. Abelisauridae (Dinosauria: Theropoda) from the Late Jurassic
of Portugal and dentition-based phylogeny as a contribution for the identification of isolated
theropod teeth. *Zootaxa* **3759**(1):1–74.

Hendrickx, C., Mateus, O., Araújo, R., 2015. A proposed terminology of theropod teeth
(Dinosauria, Saurischia). *Journal of Vertebrate Paleontology* **35**(5):e982797.

Kiernan, K. and Schwimmer, D.R. 2004. First record of a velociraptorine theropod (Tetanurae,
Dromaeosauridae) from the eastern Gulf Coastal United States. *The Mosasaur* **7**:89–93.

Krumenacker, L.J., Simon, D.J., Scofield, G., and Varricchio, D.J. 2017. Theropod dinosaurs
from the Albian-Cenomanian Wayan Formation of eastern Idaho. *Historical Biology* **29**(2):
170-186.

Langston, W. 1960. The vertebrate fauna of the Selma Formation of Alabama, part VI: the
dinosaurs. *Fieldiana: Geological Memoirs* **3**(5):315-359.

Larson, D.W., Currie, P.J. 2013. Multivariate Analyses of Small Theropod Dinosaur Teeth and
Implications for Paleocological Turnover through Time. *PLoS ONE* **8**(1): e54329.

Le Loeuff, J. 2012. Paleobiogeography and biodiversity of Late Maastrichtian dinosaurs: how
many dinosaur species went extinct at the Cretaceous-Tertiary boundary? *Bulletin de la Société*
*Géologique de France* **183**:547–559.

Leidy J. 1858. Hadrosaurus foulkii, a new saurian from the Cretaceous of New Jersey, related
to Iguanodon. *Proceedings of the Academy of Natural Sciences* **10**:213-218.

Marsh, O.C. 1881. Principal characters of American Jurassic dinosaurs. Part V. *American*
*Journal of Sciences series 3* **21**:417–423.

Matthew W.D., and Brown, B. 1922. The family Deinodontidae, with notice of a new genus from
the Cretaceous of Alberta. *Bulletin of the American Museum of Natural History* **46**:367–385.

Miller, H.W. 1967. Cretaceous vertebrates from Phoebus Landing, North Carolina. *Proceedings*
*of the Academy of Natural Sciences* **119**:219–239.

Miller, K.G., Sugarman, P.J., Browning, J.V., Kominz, M.A., Olsson, R.K., Feigenson, M.D., and
Hernandez, J.C. 2004. Upper Cretaceous sequences and sea-level history, New Jersey Coastal
Plain. *Bulletin of the Geological Society of America* **116**(3):368–393.

Ostrom, J.H. 1969. Osteology of *Deinonychus antirrhopus*, an unusual theropod from the Lower
Cretaceous of Montana. *Peabody Museum of Natural History Bulletin* **30**:1–165.

Owen, R. 1842. Report on British Fossil Reptiles, Pt. II. *Report of the British Association for the*
*Advancement of Science* **11**:60–204.

Prieto-Márquez, A., Erickson, G.M., and Ebersole, J.A. 2016. A primitive hadrosaurid from
southeastern North America and the origin and early evolution of 'duck-billed' dinosaurs. *Journal*
*of Vertebrate Paleontology* **36**(2):e1054495.

Prieto-Márquez, A., Weishampel, D.B., and Horner, J.R. 2006. The dinosaur *Hadrosaurus foulkii*,
from the Campanian of the East Coast of North America, with a reevaluation of the genus. *Acta*
*Palaeontologica Polonica* **51**(1):77-98.

Roberts, L.N.R. and Kirschbaum, M.A. 1995. Paleogeography of the Late Cretaceous of the
Western Interior of middle North America—Coal distribution and sediment accumulation. *U.S.*
*Geological Survey Professional Paper* **1561**:1–115.

Russell, D.A. 1995. China and the lost worlds of the dinosaurian era. *Historical Biology* **10**(1):3–
12.

Sakamoto, M., Benton, M.J., and Venditti, C. 2016. Dinosaurs in decline tens of millions of years
before their final extinction. *Proceedings of the National Academy of Sciences* **113**:5036–5040.

Samman T., Powell G.L., Currie P.J., and Hills, L.V. 2005. Morphometry of the teeth of western
North American tyrannosaurids and its applicability to quantitative classification. *Acta*
*Palaeontologica Polonica* **50**:757–776.

Sampson, S.D., Loewen, M.A., Farke, A.A., Roberts, E.M., Forster, C.A., Smith, J.A., and Titus,
40 A.L. 2010. New horned dinosaurs from Utah provide evidence for intracontinental dinosaur
endemism. *PLoS ONE* **5**(9):e12292.

Sarjeant, W.A.S., Currie, P.J. 2001. The “Great Extinction” that never happened: The demise of
the dinosaurs considered. *Canadian Journal of Earth Sciences* **38**(2):239–247.

Schwimmer, D.R., Sanders, A.E., Erickson, B.R., and Weems, R.E. 2015. A Late Cretaceous
dinosaur and reptile assemblage from South Carolina, USA. *Transactions of the American*
*Philosophical Society* **105**(2):1–157.

Schwimmer, D.R., Williams, G.D., Dobie, J.L., and Siesser, W.G. 1993. Late Cretaceous
dinosaurs from the Blufftown Formation in western Georgia and eastern Alabama. *Journal of*
*Vertebrate Paleontology* **67**(2):288–296.

Sereno, P.C., Dutheil, D.B., Iarochene, M., Larsson, H.C.E., Lyon, G.H., Magwene, P.M., Sidor,
C.A., Varricchio, D.J., Wilson, J.A. Predatory dinosaurs from the Sahara and Late Cretaceous
faunal differentiation. *Science* **272**:986–991.

Sloan, R.E., Rigby, J.K. Jr, Van Valen, L.M., Gabriel, D. 1986. Gradual dinosaur extinction and
simultaneous ungulate radiation in the hell creek formation. *Science* **232**(4750):629–633.

Smith, J.B. 2005. Heterodonty in *Tyrannosaurus rex*: Implications for the taxonomic and
systematic utility of theropod dentitions. *Journal of Vertebrate Paleontology* **25**:865–887.

Smith, J.B., Vann, D.R., Dodson, P. 2005. Dental morphology and variation in theropod
dinosaurs: implications for the taxonomic identification of isolated teeth. *The Anatomical Record*
**285A**:699–736.

Sues, H.-D .1978. A new small theropod dinosaur from the Judith River Formation (Campanian)
of Alberta Canada. *Zoological Journal of the Linnean Society* **62**:381–400.

Sugarman, P.J., Miller, K.G., Burky, D., and Feigenson, M.D. 1995. Uppermost Campanian-
Maastrichtian strontium isotopic; biostratigraphic and sequence stratigraphic framework of the
New Jersey Coastal Plain. *Geological Society of America Bulletin* **107**:19-37.

Turner A.H., Makovicky P.J., and Norell M.A. 2012. A Review of Dromaeosaurid Systematics
and Paravian Phylogeny. *Bulletin of the American Museum of Natural History* **371**:1–206.

von Huene F. 1914. Saurischia et Ornithischia Triadica (“Dinosuaria” Triadica).
*Animalia. Fossilium Catalogus* **4**:1-21.

Weishampel, D.B. 2006. Another look at the dinosaurs of the East Coast of North America, p.
129-168. In ‘Coletivo Arqueológico-Paleontológico Salense, (eds.), Actas III Jornadas
Dinosaurios Entorno. Salas de los Infantes, Burgos, Spain.

Weishampel, D.B. and Young, L. 1996. *Dinosaurs of the East Coast*. Johns Hopkins University
Press, Baltimore, Maryland, USA.

Weishampel, D.B., Barrett, P.M., Coria, R.A., Loeuff, J.L., Xing, X., Xijin, Z., Sahni, A.,
Gomani, E.M.P., and Noto, C.R. 2004. Dinosaur Distribution. In: Weishampel, D.B., Dodson, P.,

and Osmólska, H. eds: The Dinosauria, 2nd Edition. University of California Press, Berkeley,
California, USA. p. 517-617.

Wick, S.L., Lehman, T.M., Brink, A.A. 2015. A theropod tooth assemblage from the lower Aguja
Formation (early Campanian) of West Texas, and the roles of small theropod and varanoid lizard
mesopredators in a tropical predator guild. *Palaeogeography, Palaeoclimatology,*
*Palaeoecology* **418**:229.

Williamson T.E., and Brusatte S.L. 2014. Small Theropod Teeth from the Late Cretaceous of the
San Juan Basin, Northwestern New Mexico and Their Implications for Understanding Late
Cretaceous Dinosaur Evolution. *PLoS ONE* **9**:e93190.

Figure 1. Locality information, Maastrichtian eastern North American dinosaurs, and Mount Laurel tyrannosauroid tooth anatomy. (A) map of New Jersey showing the location of Burlington and Monmouth counties and the Big Brook site (pink dot), (B) possible partial Lambeosaurine forelimb, (C) cast of the manual ungual of *Dryptosaurus aquilunguis*, and (D) pedal palanx of an ornithomimosaur. NJSM GP 14256 in labial (E—F) distal (G), and basal (H) views with a closeup of the enamel wrinkles (I) and distal denticles (J). Map is public domain, access to photograph the cast of the manual ungual of *Dryptosaurus aquilunguis* courtesy of the Yale Peabody Museum (peabody.yale.edu).

Figure 2. Support for the assignment of NJSM GP 12456 in Tyrannosauroidae. (A) Principle components analysis of coelurosaurian teeth including a large dromaeosaurid tooth from North Carolina and the Mt. Laurel tyrannosauroid tooth (Smith et al., 2005; Brownstein, 2018b). Principle component 1 accounted for 91.677% of variance, whereas principle component 2 accounted for 4.0981%. (B) Phylogenetic topology of theropod teeth, with Tyrannosauroidae highlighted in red.

Figure 3. Anatomy of the Mt. Laurel dromaeosaurid tooth. NJSM GP 22949 in labial (A), lingual (B), basal (C), and distal (D) views, with a closeup (E) of the distal denticles. (F), teeth of *Velociraptor*, (G) dentary and teeth of “*Bambiraptor*,” (H) premaxilla and teeth of *Utahraptor*, and (I) dentary and teeth of *Dromaeosaurus*. r, enamel ridge; tc, twisted mesial carina.

Figure 4. Principle components and discriminant analyses of the Mt. Laurel dromaeosaurid tooth. (A), Principle components analysis of coelurosaurian teeth including a large dromaeosaurid tooth from North Carolina and the Mt. Laurel dromaeosaurid tooth (Smith et al., 2005; Brownstein, 2018b). Principle component 1 accounted for 91.722% of variance, whereas principle component 2 accounted for 4.0932%. (B), Principle components analysis of North American paravian teeth including the Mt. Laurel dromaeosaurid tooth (Larson and Currie, 2013). Principle component 1 accounted for (C), Discriminant analysis of North American paravian teeth including the Mt. Laurel dromaeosaurid tooth (Larson and Currie, 2013).

Appendix B

Cover Letter

July 10, 2019

Dear editor of *Royal Society Open Science*,

I would like to resubmit to you my manuscript “New records of theropods from the latest Cretaceous of New Jersey and the Maastrichtian Appalachian fauna” for consideration at your journal.

In revising my manuscript, I have paid special attention to two major comments given by the reviewers. Firstly, I’ve removed the PCA of the dataset of Smith et al. (2005) with the tyrannosauroid tooth included on the advice of reviewer 2, provided additional justification for why the tooth is from a tyrannosauroid and probably not a tyrannosaurid, and provided additional information on the results of the phylogenetic analysis.

Secondly, I’ve redone the PCA and discriminant analyses including the dromaeosaurid tooth such that the analyses of the Larson and Currie dataset include the Tar Heel tooth and the discriminant analysis of the Smith et al. dataset does not include characters that could not be scored for the Mt. Laurel tooth.

Finally, I’ve removed the biogeographic speculations in line with the comments of the reviewers and editor. I agree that these were presented far too strongly, and I have reorganized the manuscript to act as a faunal characterization. I’ve also revised the manuscript in accordance with the many minor comments presented in reviewer 2’s annotated PDF.

I hope the manuscript is now suitable for publication, and thank you for your reviews.

Regards,

Chase Brownstein
Research Associate,
Collections & Exhibitions, Stamford Museum & Nature Center
Stamford, Connecticut, United States

Responses to Reviewers.

Chase D. Brownstein

However, your data and analyses either do not support your main claim or are insufficient to do so. Starting with your dataset, your measurements of NJSMP GP 12456 for DB, DC, and DA do not appear congruent with the specimen figured in Fig. 1J, though do seem to match the scale of Fig. 1E. In Fig. 1J, measurements closer to 4 would appear to be more accurate than 2, 2, and 2.5. The scale bars for the photos and your measurements should be double-checked.

Done. I've double checked the scale bars to make sure they are correct. Please see the revised figure.

You also have a substantial amount of missing data for your new specimens in your dataset, and you haven't discussed how this was handled in your PCA. The default in PAST is to fill missing data with column average substitution, which in the Smith et al. (2005) dataset will bias all of your specimens towards the tyrannosaur part of your plot (as there are more tyrannosaurs in the dataset than any other group). Given that all three of the new teeth plot in areas of your plots where no other theropod teeth plot (partway between dromaeosaurs and tyrannosaurs), it appears that column average substitution of your dataset is having a substantial contribution to the position of your specimens on these plots. As well, eastern taxa are notably excluded. Dryptosaurus is not plotted in Fig. 2A, and the Tar Heel dromaeosaurid is not plotted in Fig. 4B and 4C, making the argument that either of these teeth have eastern or western affinities insufficiently supported by these analyses. Additions of those specimens and the exclusion of variables with missing data are necessary for these plots to be informative.

I appreciate this comment, and I have added measurements on the Tar Heel form to the datasets for analysis of the dromaeosaurid tooth. I've gone on the suggestion of reviewer 2 and not performed morphometric work on the tyrannosaur tooth. The affinities of that tooth to tyrannosaurs are still supported by several morphological comparisons and the phylogenetic analysis. I've also removed measurements that are not known for the new teeth to better perform the analysis. Moreover, instead of performing principle components analyses, I've performed a discriminant analysis of the Smith et al. (2005) dataset with characters unknown for the dromaeosaurid tooth removed to better support the referrals. Because the only character in the Larson and Currie dataset unknown for the dromaeosaurid tooth is that of the mesial denticle count, I left the dataset as is for that morphometric analysis.

*In the phylogenetic analysis, you only list shared derived characters between your tyrannosaur specimen and *T. rex*, but do not list shared derived characters that unite all tyrannosauroids. However, even this analysis does not support your argument that the specimen is a non-tyrannosaurid tyrannosauroid with affinities with *Dryptosaurus*. Also, one of the four characters you do list, 94) biconvex apical distal denticles, is not mentioned in your description at all nor is it visible in Fig. 1.*

I have added the shared derived characters that unite all tyrannosauroids to the paper's text and noted the presence of biconvex denticles along the entirety of the carina.

As well, Fig. 1B, 1C, and 1D depict fossils that are not described in the manuscript. If these specimens are figured in this manuscript, they should be described systematically in the text.

These fossils have been described in previous papers and are there to give the reader a visual map of what fossils are known from the Maastrichtian of NJ.

Alternatively, these figures could easily be excluded from the manuscript. The entire datasets used for analysis, including the Smith et al. (2005) and Larson and Currie (2013) should be reproduced in the supplemental data for ease of replication.

Done. I've added the datasets to the Supplemental Data.

This description is of specialist interest given the new records from eastern North America, but any discussion of faunal interchange is premature. In your description, you have demonstrated that you have teeth of a dromaeosaurid and a tyrannosauroid from this formation, but your analyses in their current form are insufficient to demonstrate that these teeth have any biogeographic affinities.

Done. I've modified the paper to focus on the significance of these teeth for contributing to our understanding of Appalachian faunas, and removed the biogeographic speculations.

Reviewer: 2

Comments to the Author(s)

This paper is important for the acknowledgement of the taxa found in this New Jersey formation. Outside of this finding there is much more difficulty in acquiring significance. This problem of lack of greater significance is not the fault of the author but the nature of the Appalachian dinosaur record. Fossiliferous formations are poorly dated, fossils are not terribly abundant, and when known are typically in rough condition. Because there is not a proper framework for the dinosaurs, determining faunal interchange is problematic...you need to know what is around when, and the organism's phylogenetic relationships. Without this it is difficult to say anything concrete about paleobiology. Especially, in these days when doing rigorous biogeographic analyses is so easy. In my opinion you can't really say anything about faunal interchange until a solid, dated fossil record is assembled.

I agree. See my comment above.

First, I firmly do not think that a phylogenetic analysis should be used on a single tooth, much less a single partial tooth. The phylogenetic analysis of teeth is not a proper use of phylogenetics in my opinion at any rate. To my mind this part of the paper should be eliminated.

This part of the paper was recommended by another researcher who looked at my paper. As it follows in line with many recent studies (e.g., Hendrickx et al., 2014; Young et al., 2014), I have kept it for the sake of making this paper comparable to other, recent works.

Second, the author should use a more recent ordinal analysis of teeth. This is not a deal breaker, but the primary dataset chosen is 13 years old (or 5 yrs old in the case of Larson and Currie) and many others have been produced since then.

Since the Larson and Currie dataset continues to be the one used the most often for morphometric analysis of Cretaceous North American theropod teeth (see Evans et al., 2013; Williamson and Brusatte, 2014 for examples), I am uncertain what the reviewer here refers to. The Hendrickx et al. (2015) theropod tooth dataset only represents an extension of the Smith et al. (2005) dataset that includes the Larson and Currie dataset and removes a number of continuous characters.

Third, the tyrannosaur tooth should absolutely be removed from the PCA analysis. It is a fragmentary tooth, therefore any measurements taken are not going to be the correct measurement and will provide erroneous results. Or there will be too little data to say much about.

Done. I've removed the tooth.

Fourth, the dromaeosaurid tooth falls outside the dromaeosaurids in the Smith/Brownstein matrix. And is just part of the miasma of teeth in the Larson and Currie matrix. I would suggest pulling individual taxa from the analyses and talk about comparisons with them instead of consistently saying it is a Saurornitholestesine theropod. Remember that the PCA is using data you input. If there is more information supporting your taxonomic claim that is not included in the quantitative data, then you should discuss this in the text. In PAST you can change the symbol and the color, meaning that you can label individual dinosaur taxa and the higher level taxonomic groups at the same time. This should be done to help you see which species the Tar Heel and Mt Laurel specimens are mostly similar to.

Done. I've instead performed discriminant analyses on the datasets and taking up your suggestion of visualizing the resulting plots using axes 2 and 3. The Mt. Laurel tooth plots firmly within the hull formed by the teeth of *Velociraptor* in the first DA and firmly with

Fifth, this is very important! When using PCA on variables that are measurements, the first axis is almost invariably a reflection of size, not shape. Therefore, you need to show axes 2 and 3. You are welcome to retain axis 1 and 2, but 2 and 3 must be shown in addition. You may also want to consider Centering or z-transforming your data because it can help remove the effect of size, bring outliers closer together, and reflect shape better.

Done.

Appendix C**ROYAL SOCIETY
OPEN SCIENCE****New records of theropods from New Jersey inform faunal
interchange in Maastrichtian North America**

Journal:	Royal Society Open Science
Manuscript ID	RSOS-191206
Article Type:	Research
Date Submitted by the Author:	10-Jul-2019
Complete List of Authors:	Brownstein, Chase; Stamford Museum and Nature Center,
Subject:	Palaeontology < EARTH SCIENCES, evolution < BIOLOGY, ecology < BIOLOGY
Keywords:	Dinosaur, Biogeography, Appalachia, Cretaceous
Subject Category:	Biology (whole organism)

**Author-supplied statements**

Relevant information will appear here if provided.

***Ethics***

*Does your article include research that required ethical approval or permits?:*

This article does not present research with ethical considerations

*Statement (if applicable):*

CUST_IF_YES_ETHICS :No data available.

***Data***

*It is a condition of publication that data, code and materials supporting your paper are made publicly*
*available. Does your paper present new data?:*

Yes

*Statement (if applicable):*

The data is included in the supplementary material.

***Conflict of interest***

I/We declare we have no competing interests

*Statement (if applicable):*

CUST_STATE_CONFLICT :No data available.

***Authors' contributions***

I am the only author on this paper

*Statement (if applicable):*

CUST_AUTHOR_CONTRIBUTIONS_TEXT :No data available.

Cover Letter

July 10, 2019

Dear editor of *Royal Society Open Science*,

I would like to resubmit to you my manuscript “New records of theropods from the latest Cretaceous of New Jersey and the Maastrichtian Appalachian fauna” for consideration at your journal.

In revising my manuscript, I have paid special attention to two major comments given by the reviewers. Firstly, I’ve removed the PCA of the dataset of Smith et al. (2005) with the tyrannosauroid tooth included on the advice of reviewer 2, provided additional justification for why the tooth is from a tyrannosauroid and probably not a tyrannosaurid, and provided additional information on the results of the phylogenetic analysis.

Secondly, I’ve redone the PCA and discriminant analyses including the dromaeosaurid tooth such that the analyses of the Larson and Currie dataset include the Tar Heel tooth and the discriminant analysis of the Smith et al. dataset does not include characters that could not be scored for the Mt. Laurel tooth.

Finally, I’ve removed the biogeographic speculations in line with the comments of the reviewers and editor. I agree that these were presented far too strongly, and I have reorganized the manuscript to act as a faunal characterization. I’ve also revised the manuscript in accordance with the many minor comments presented in reviewer 2’s annotated PDF.

I hope the manuscript is now suitable for publication, and thank you for your reviews.

Regards,

Chase Brownstein
Research Associate,
Collections & Exhibitions, Stamford Museum & Nature Center
Stamford, Connecticut, United States

Responses to Reviewers.

Chase D. Brownstein

However, your data and analyses either do not support your main claim or are insufficient to do so. Starting with your dataset, your measurements of NJSM GP 12456 for DB, DC, and DA do not appear congruent with the specimen figured in Fig. 1J, though do seem to match the scale if Fig. 1E. In Fig. 1J, measurements closer to 4 would appear to be more accurate than 2, 2, and 2.5. The scale bars for the photos and your measurements should be double-checked.

Done. I've double checked the scale bars to make sure they are correct. Please see the revised figure.

You also have a substantial amount of missing data for your new specimens in your dataset, and you haven't discussed how this was handled in your PCA. The default in PAST is to fill missing data with column average substitution, which in the Smith et al. (2005) dataset will bias all of your specimens towards the tyrannosaur part of your plot (as there are more tyrannosaurs in the dataset than any other group). Given that all three of the new teeth plot in areas of your plots where no other theropod teeth plot (partway between dromaeosaurs and tyrannosaurs), it appears that column average substitution of your dataset is having a substantial contribution to the position of your specimens on these plots. As well, eastern taxa are notably excluded. Dryptosaurus is not plotted in Fig. 2A, and the Tar Heel dromaeosaurid is not plotted in Fig. 4B and 4C, making the argument that either of these teeth have eastern or western affinities insufficiently supported by these analyses. Additions of those specimens and the exclusion of variables with missing data are necessary for these plots to be informative.

I appreciate this comment, and I have added measurements on the Tar Heel form to the datasets for analysis of the dromaeosaurid tooth. I've gone on the suggestion of reviewer 2 and not performed morphometric work on the tyrannosaur tooth. The affinities of that tooth to tyrannosaurs are still supported by several morphological comparisons and the phylogenetic analysis. I've also removed measurements that are not known for the new teeth to better perform the analysis. Moreover, instead of performing principle components analyses, I've performed a discriminant analysis of the Smith et al. (2005) dataset with characters unknown for the dromaeosaurid tooth removed to better support the referrals. Because the only character in the Larson and Currie dataset unknown for the dromaeosaurid tooth is that of the mesial denticle count, I left the dataset as is for that morphometric analysis.

*In the phylogenetic analysis, you only list shared derived characters between your tyrannosaur specimen and *T. rex*, but do not list shared derived characters that unite all tyrannosauroids. However, even this analysis does not support your argument that the specimen is a non-tyrannosaurid tyrannosauroid with affinities with *Dryptosaurus*. Also, one of the four characters you do list, 94) biconvex apical distal denticles, is not mentioned in your description at all nor is it visible in Fig. 1.*

I have added the shared derived characters that unite all tyrannosauroids to the paper's
text and noted the presence of biconvex denticles along the entirety of the carina.

*As well, Fig. 1B, 1C, and 1D depict fossils that are not described in the manuscript. If*
*these specimens are figured in this manuscript, they should be described systematically*
*in the text.*

These fossils have been described in previous papers and are there to give the reader a
visual map of what fossils are known from the Maastrichtian of NJ.

*Alternatively, these figures could easily be excluded from the manuscript. The entire*
*datasets used for analysis, including the Smith et al. (2005) and Larson and Currie*
*(2013) should be reproduced in the supplemental data for ease of replication.*

Done. I've added the datasets to the Supplemental Data.

*This description is of specialist interest given the new records from eastern North*
*America, but any discussion of faunal interchange is premature. In your description, you*
*have demonstrated that you have teeth of a dromaeosaurid and a tyrannosauroid from*
*this formation, but your analyses in their current form are insufficient to demonstrate that*
*these teeth have any biogeographic affinities.*

Done. I've modified the paper to focus on the significance of these teeth for contributing
to our understanding of Appalachian faunas, and removed the biogeographic
speculations.

Reviewer: 2

Comments to the Author(s)

*This paper is important for the acknowledgement of the taxa found in this New Jersey*
*formation. Outside of this finding there is much more difficulty in acquiring significance.*
*This problem of lack of greater significance is not the fault of the author but the nature of*
*the Appalachian dinosaur record. Fossiliferous formations are poorly dated, fossils are*
*not terribly abundant, and when known are typically in rough condition. Because there is*
*not a proper framework for the dinosaurs, determining faunal interchange is*
*problematic...you need to know what is around when, and the organism's phylogenetic*
*relationships. Without this it is difficult to say anything concrete about paleobiology.*
*Especially, in these days when doing rigorous biogeographic analyses is so easy. In my*
*opinion you can't really say anything about faunal interchange until a solid, dated fossil*
*record is assembled.*

I agree. See my comment above.

*First, I firmly do not think that a phylogenetic analysis should be used on a single tooth,*
*much less a single partial tooth. The phylogenetic analysis of teeth is not a proper use*
*of phylogenetics in my opinion at any rate. To my mind this part of the paper should be*
*eliminated.*

This part of the paper was recommended by another researcher who looked at my
paper. As it follows in line with many recent studies (e.g., Hendrickx et al., 2014; Young
et al., 2014), I have kept it for the sake of making this paper comparable to other, recent
works.

*Second, the author should use a more recent ordinal analysis of teeth. This is not a deal*
*breaker, but the primary dataset chosen is 13 years old (or 5 yrs old in the case of*
*Larson and Currie) and many others have been produced since then.*

Since the Larson and Currie dataset continues to be the one used the most often for
morphometric analysis of Cretaceous North American theropod teeth (see Evans et al.,
2013; Williamson and Brusatte, 2014 for examples), I am uncertain what the reviewer
here refers to. The Hendrickx et al. (2015) theropod tooth dataset only represents an
extension of the Smith et al. (2005) dataset that includes the Larson and Currie dataset
and removes a number of continuous characters.

*Third, the tyrannosaur tooth should absolutely be removed from the PCA analysis. It is a*
*fragmentary tooth, therefore any measurements taken are not going to be the correct*
*measurement and will provide erroneous results. Or there will be too little data to say*
*much about.*

Done. I've removed the tooth.

*Fourth, the dromaeosaurid tooth falls outside the dromaeosaurids in the Smith/*
*Brownstein matrix. And is just part of the miasma of teeth in the Larson and Currie*
*matrix. I would suggest pulling individual taxa from the analyses and talk about*
*comparisons with them instead of consistently saying it is a Saurornitholestesine*
*theropod. Remember that the PCA is using data you input. If there is more information*
*supporting your taxonomic claim that is not included in the quantitative data, then you*
*should discuss this in the text. In PAST you can change the symbol and the color,*
*meaning that you can label individual dinosaur taxa and the higher level taxonomic*
*groups at the same time. This should be done to help you see which species the Tar*
*Heel and Mt Laurel specimens are mostly similar to.*

Done. I've instead performed discriminant analyses on the datasets and taking up your
suggestion of visualizing the resulting plots using axes 2 and 3. The Mt. Laurel tooth
plots firmly within the hull formed by the teeth of *Velociraptor* in the first DA and firmly
with

*Fifth, this is very important! When using PCA on variables that are measurements, the*
*first axis is almost invariably a reflection of size, not shape. Therefore, you need to show*
*axes 2 and 3. You are welcome to retain axis 1 and 2, but 2 and 3 must be shown in*
*addition. You may also want to consider Centering or z-transforming your data because*
*it can help remove the effect of size, bring outliers closer together, and reflect shape*
*better.*

Done.

**New records of theropods from the latest Cretaceous of New Jersey and the Maastrichtian**

**Appalachian fauna**

Chase Doran Brownstein

Research Associate, Dept. of Collections and Exhibitions, Stamford Museum and Nature Center,

Stamford, CT, chasethedinosaur@gmail.com

Abstract.

The faunal changes that occurred in the few million years before the Cretaceous-Paleogene
extinction are of much interest to vertebrate paleontologists. Western North America preserves
arguably the best fossil record from this time, whereas terrestrial vertebrate fossils from the
eastern portion of the continent are usually limited to isolated, eroded postcranial remains.
Examination of fragmentary specimens from the American east, which was isolated for the
majority of the Cretaceous as the landmass Appalachia, is therefore important for better
understanding dinosaur diversity at the end of the Mesozoic. Here, I report on two theropod teeth
from the Mount Laurel Formation, a lower-middle Maastrichtian unit from northeastern North
America. One of these preserves in fine-detail the structure of the outer enamel and resembles
the dentition of the tyrannosauroid *Dryptosaurus aquilunguis* among latest Cretaceous forms in
being heavily mediolaterally compressed and showing many moderately developed enamel
crenulations. Along with previously reported tyrannosauroid material from the Mt. Laurel and
overlying Cretaceous units, this fossil supports the presence of non-tyrannosaurid
tyrannosauroids in the Campanian-Maastrichtian of eastern North America. The other tooth is
assignable to a dromaeosaurid and represents both the youngest occurrence of a non-avian
maniraptoran in eastern North America and the first from the Maastrichtian reported east of the
Mississippi. This tooth, which belonged to a medium-sized dromaeosaurid based on size
comparisons with the teeth of taxa for which skeletons are known, increases the diversity of the
Maastrichtian dinosaur fauna of Appalachia. Along with previously reported dromaeosaurid
teeth, the Mt. Laurel specimen supports the presence of mid-sized to large dromaeosaurids in
eastern North America throughout the Cretaceous. This indicates that dromaeosaurids

approaching or exceeding 3 meters in length were more common in the Late Cretaceous than
previously thought.

Keywords: Dinosaur; Appalachia; Cretaceous; Fauna ; Tyrannosaur; Teeth

**Introduction.**

The extinction of the non-avian dinosaurs at the end of the Mesozoic Era is a topic that
has continued to intrigue vertebrate paleontologists (e.g., Alvarez et al., 1980; Sloan et al., 1986;
Sarjeant and Currie, 2001; Le Loeuff 2012, Brusatte et al., 2012, 2015; Sakamoto et al., 2016).
However, a poor global terrestrial record from the Maastrichtian has hindered attempts to assess
the diversity dynamics of important groups like the Dinosauria during this period (e.g., Le
Loeuff, 2012; Brusatte et al., 2015). Western North America preserves arguably the most well-
characterized vertebrate record from the last 20 million years of the Mesozoic Era (Brusatte et
al., 2015), whereas that from the eastern portion of the continent is far more obscure. During the
majority of the Late Cretaceous, eastern and western North America were separated, the former
existing as a rectangular landmass called Appalachia. Appalachian dinosaur faunas included
intermediate-grade tyrannosauroids (Carr et al., 2005; Brusatte et al., 2011), basal hadrosaurids
and derived non-hadrosaurid hadrosauroids (Langston, 1960; Prieto-Márquez et al., 2006; Prieto-
Márquez et al., 2016), nodosaurids (Gallagher, 1993; Burns and Ebersole, 2016), and
ornithomimosaur (Miller, 1967; Schwimmer et al., 1993; Weishampel and Young, 1996;
Schwimmer et al., 2015; Brownstein, 2018a).

Despite the amount of knowledge of Cretaceous faunal change to be gleaned from the
fossil record of Appalachia, the assemblages of this landmass have remained fundamentally
understudied since the mid-19th century (e.g., Weishampel & Young, 1996; Gallagher, 1997;

Weishampel, 2006; Brownstein, 2018a). The scarcity of terrestrial sedimentary units known from
the eastern half of the United States has also contributed to the obscurity of Appalachian faunas
compared to western North American ones (Weishampel and Young, 1996; Gallagher, 1997;
Carr et al., 2005; Weishampel, 2006; Brusatte et al., 2011). Only in the past few years have
indications of faunal changes in the latest Cretaceous (late Campanian-Maastrichtian) of the
American east come, and all from isolated, fragmentary finds. Although a ceratopsian tooth from
the uppermost Maastrichtian of Mississippi (Farke and Phillips, 2017) and possible
lambeosaurine bones from the upper Maastrichtian of New Jersey (Gallagher, 1993) have
revealed that faunal exchanges probably occurred between Appalachia and Laramidia following
the regression of the Western Interior Seaway in the latest Campanian-earliest Maastrichtian, the
timing of these events remains poorly constrained. Further sampling of enigmatic assemblages
from the Maastrichtian, such as those of the eastern United States, is therefore important for
understanding faunal change in latest Mesozoic North America.

In the Campanian-Maastrichtian of New Jersey, a set of formations corresponds to a
period of transgressions and regressions of the Atlantic Ocean (e.g., Gallagher, 1993; Sugarman
et al., 1995; Miller et al., 2004; Gallagher et al., 2014). The majority of these Cretaceous units
are known for producing marine vertebrate and invertebrate fossils (Gallagher, 1993), although
some, such as the Woodbury and New Egypt formations, are notable for producing some of the
first partial dinosaur skeletons from the Americas (e.g., Leidy, 1858; Cope, 1866; Gallagher,
1993, Weishampel and Young, 1996; Gallagher, 1997). One of the most fossiliferous of these
formations is the Mount Laurel Formation, which is either uppermost Campanian or lowermost
Maastrichtian (Miller et al., 2004) and in New Jersey has produced the remains of several groups
of dinosaurs, including hadrosaurs, tyrannosaurs, and ornithomimosaur (Gallagher, 1993;

Gallagher, 2014). Because of the sheer diversity of the community represented in the Mt. Laurel,
the formation serves as a window into Campanian-Maastrichtian eastern North American faunas.
However, the terrestrial fossils it produces are often eroded postcranial fragments (Gallagher,
2014).

Here, I describe some theropod teeth from the Mt. Laurel Formation of New Jersey.
These include a large tooth assignable to a 6—8m tyrannosauroid and a smaller, heavily
recurved one assignable to a 3—4 m dromaeosaurid. These teeth are among the most diagnostic
records of theropods from the Mt. Laurel Formation, allowing for a more precise understanding
of the faunal composition and ecology of the eastern seaboard during the Maastrichtian, a
globally under-sampled time period (Le Loeuff, 2012; Brusatte et al., 2012).

**Results.**

Geological setting.

Both theropod teeth described here were collected from sediments of the Mount Laurel
Formation (Gallagher, 1993; pers. obs.), a marine deposit that represents a regression of the
Atlantic Ocean during the Late Cretaceous period and is the oldest unit included in the
Monmouth Group (Gallagher, 1993; Miller et al., 2004). The tyrannosauroid tooth described
here, NJSM GP 12456, was recovered from Big Brook (Fig. 1A), a highly fossiliferous locality
famous for producing an extensive marine fauna (Gallagher, 1993; Weishampel and Young,
1996). At Big Brook, the stratigraphic column is exposed along the banks, with the Wenonah
Formation grading into the Mt. Laurel such that the border between the two are indistinguishable
(Gallagher, 1993). The contact between the Mt. Laurel and the overlying Navesink Formation is
an unconformity (Miller et al., 2004; Gallagher et al., 2014). The Mt. Laurel Formation appears
as gray to dark brown, pebbly quartz sands. The Big Brook tyrannosauroid tooth (Fig. 1E—H) is

unusual among the terrestrial vertebrate teeth collected from the site in possessing a well-
preserved enamel surface. Whereas other terrestrial vertebrate fossils from Big Brook are known
for being heavily water-worn and lacking morphological details, NJSM GP 12456 preserves both
its outermost enamel layer and many of its denticles.

NJSM GP 22949, the dromaeosaurid tooth, was recovered from Mt. Laurel deposits in
Burlington County, New Jersey (Fig. 1A). In this area, which makes up a portion of the
southwestern-most range of the Monmouth Group, the sands of the Mt. Laurel are more
glauconitic than farther north and are intermixed with iron compounds (Gallagher, 1993). The
thickness of this unit is also far greater to the southwest of its range (e.g., Gallagher, 1993).

Tyrannosauroid tooth.

Dinosauria Owen 1842

Theropoda Marsh 1881

Coelurosauria von Huene 1921

Tyrannosauroida Osborn 1905

Tyrannosauroida indet.

Material.

New Jersey State Museum collections (NJSM) GP 14256, the partial tooth of a large theropod
dinosaur (Fig. 1E—H).

Locality and Horizon.

134 Mt. Laurel Formation sediments at Big Brook, Monmouth County, New Jersey, latest
Campanian to early Maastrichtian (Gallagher, 1993; Miller et al., 2004; Gallagher et al., 2014).
Identification.

NJSM GP 14256 (Fig. 1E—H) closely resembles the dentition of tyrannosauroid
theropods in several respects. The tooth resembles those of adult tyrannosauroids in its size,
which is closely comparable to tyrannosaur crowns known from both western and eastern North
America (Currie et al., 1990; Brochu, 2003; Samman et al., 2005; Smith, 2005; Brusatte et al.,
2011; Williamson and Brusatte, 2014). In addition to its size, the Mt. Laurel tooth resembles
those of tyrannosaurs to the exclusion of other theropods known from Late Cretaceous North
America in possessing a combination of packed denticles (2–2.5/mm) on its distal carina (15+
144 mm), the presence of denticles along both carinae, its slight, rather than pronounced, curvature,
the presence of numerous transverse undulations (density = 2/mm) on its main surface, the
presence of slightly biconvex denticle outlines for denticles all along the tooth (Fig. 1J), and its
smooth but slightly irregular surface texture (Fig 1E—H)(Currie et al., 1990; Brochu, 2003;
Samman et al., 2005; Smith, 2005; Brusatte et al., 2011; Williamson and Brusatte, 2014).
However, despite the size of the Mt. Laurel tooth, NJSM GP 14256 is notably unlike the teeth of
tyrannosaurids, for which incrassate teeth are a synapomorphy (e.g., Williamson and Brusatte,
2014; Brusatte and Carr, 2016). Instead, NJSM GP 14256 shows is highly mediolaterally
compressed and possesses a lens-shaped basal cross-section, indicative it came from a
tyrannosauroid outside Tyrannosauridae. Among large Late Cretaceous tyrannosaurs, only
*Dryptosaurus aquilunguis* from the Maastrichtian New Egypt Formation of New Jersey is known
to possess a combination of high mediolateral compression and tyrannosaurid-like features of the
denticles and tooth surface (e.g., Brusatte et al., 2011). NJSM GP 14256 is also comparable with
the mediolaterally compressed teeth of *Dryptosaurus aquilunguis* in its dimensions, curvature,
and enamel crenulations (Brusatte et al., 2011; pers. obs. of YPM PU 22208). Given the Mt.

Laurel tooth's very close spatiotemporal proximity to the holotype of *Dryptosaurus*, I suggest the
tooth belongs to a closely related form.

Given the number of measurements unable to be taken from this tooth, I did not include it
in a morphometric analysis. However, phylogenetic analysis of the tooth within the dataset of
Hendrickx and Mateus (2014) found NJSM GP 14256 to be the sister taxon of *Tyrannosaurus*
*rex* in a clade united by four characters. These are characters 94 (biconvex apical denticles
present on distal carinae of lateral teeth), 100 (subequal number of denticles apically than at mid-
crown portion of distal carinae on lateral teeth), 103 (interdenticular space between mid-crown
denticles on distal carinae of lateral teeth broad), and 105 (interdenticular sulci between mid-
crown denticles on distal carinae of lateral teeth present, long, and well-developed)(Hendrickx
and Mateus, 2014). The clade comprised of NJSM GP 14256 and other derived tyrannosaurs
(*Alioramus*, *Tyrannosaurus*, “*Raptorex*”) is united by the presence of a sub-symmetric crown
with a centrally-positioned distal carina in distal view (char. 83). Characters uniting the
tyrannosauroid clade include 3, 5, 19, 27, 37, 38, 41, and 48 in the list of Hendrickx and Mateus
(2014). The strict consensus tree (tree length = 688, consistency index = 0.340, retention index =
0.561) is shown in Fig. 1B.

Description.

NJSM GP 14256 (Fig. 1) is the apical half of the tooth of a theropod dinosaur.
Measurements of the specimen may be found in Table 1. The tooth is well-preserved for a
terrestrial fossil collected from one of the marine deposits of the Cretaceous Atlantic Coastal
Plain, preserving details of the outer enamel layer and denticle morphology. Unfortunately, the
basal half of the crown and the entirety of the root of the tooth are not preserved. This is

probably due to erosion, as the tooth is broken transversely and heavily rounded at its preserved
 base (Fig. 1E—F).

The tooth displays the ziphodont condition in being labiolingually compressed and only
 slightly recurved. The preserved mesial carina is slightly convex, whereas the distal carina is
 vertical along its entire run. The labial and lingual portions of the enamel are well-preserved
 (Fig. 1G), bearing transverse undulations that develop out of the distal margin of the tooth to
 become bands (Brusatte et al. 2007). In NJSM GP 14256, these undulations (= marginal bands)
 are relatively strongly developed, although they are less prominent than in carcharodontosaurids
 (e.g., Sereno et al., 1996; Brusatte et al., 2007). The labial and lingual surfaces of the tooth are
 slightly convex, as in most other theropod dinosaurs (Hendrickx et al., 2015a). The apex of the
 tooth bears a slight wear facet on its lingual surface. The tooth is lenticular in basal cross-section.

The distal carina preserves many denticles (Fig. 1E—F, H), which are small, dense
 (6/mm), and apicobasally straightened. The denticles are interspersed with diminutive
 interdenticular sulci (Currie et al., 1990). These are encompassed by the apical ends of the
 denticles. These denticles maintain a similar density along

Table 1. Measurements of teeth described in this study (in/per mm).

Specimen	CH	CBL	CBW	AL	CA	DB	DC	DA
NJSM GP 12456	15 (est. 25)	8.99	4.90	N/A	N/A	2	2	2.5
NJSM GP 22949	15.5	8.0	2.0	18.00	55.5	6	5	5

the entirety of the distal carina. However, their density may have changed along the missing

portion of the tooth. The mesial carina preserves a few denticles, although these are too eroded

for much morphological description. These denticles appear to be similar in size to those on the
distal carina.

Dromaeosaurid tooth.

Dinosauria Owen 1842

Theropoda Marsh 1881

Coelurosauria von Huene 1921

Maniraptora Gauthier 1986

Dromaeosauridae Matthew & Brown 1922

Saurornitholestinae Sues 1978

cf. Saurornitholestinae indet.

Material.

NJSM GP 22949, well-preserved, complete isolated tooth.

Locality and Horizon.

212 Mt. Laurel Formation sediments in Burlington County, New Jersey, latest Campanian to early

Maastrichtian (Gallagher, 1993; Miller et al., 2004; Gallagher et al., 2014).

Identification.

NJSM GP 22949 is identified as the lateral tooth of a dromaeosaurid theropod based on

the following combination of features: (1) its extreme apicobasal curvature created by its

concave distal carina and distally offset apex, (2) the presence of apically hooked distal denticles,

(3) the absence of mesiodistal constriction along the crown base, and (4) distal denticles that

decrease in size towards the apex of the tooth (Fig. 3A—E)(Currie et al., 1990; Turner et al.,

2012; Larson and Currie, 2013; Williamson and Brusatte, 2014). NJSM GP 22949 is smaller

than the majority of Appalachian theropod teeth assigned to tyrannosauroids, in which crown

heights surpass 50 mm (Schwimmer et al., 1993; Carr et al., 2005; Brusatte et al., 2011; Denton
et al., 2011; Schwimmer et al., 2015). However, the tooth is notably larger than most North
American dromaeosaurid teeth, which are often less than 10 mm in height and mostly measure
around 5 mm in that dimension (Ostrom, 1969; Currie et al., 1990; Currie and Varricchio, 2004;
Larson and Currie, 2013; Williamson and Brusatte, 2014; Wick et al., 2015). *Dakotaraptor*,
which possessed crowns up to ~25 mm high, represents the exception among Maastrichtian
dromaeosaurids, the teeth of which usually are below 10 mm in height (Larson and Currie, 2013;
DePalma et al., 2015). Instead, the Mt. Laurel dromaeosaurid tooth is more comparable to the
teeth of the 3—4 meter *Deinonychus* and an indeterminate specimen from the Tar Heel
Formation of North Carolina in its dimensions (Ostrom, 1969; Brownstein, 2018b). NJSMP GP
22949 is also distinguished from Appalachian tyrannosauroids in lacking subquadrangular
distocentral denticles (Hendrickx et al., 2015). The first discriminant analysis on the Mt. Laurel
dromaeosaurid tooth supports this hypothesis by placing the specimen within the convex hull
formed by the teeth of *Velociraptor* and not in the convex hull formed by the teeth of
tyrannosaurs or troodontids (Fig. 4A).

NJSMP GP 22949 is notable for being similar to the teeth of western North American
saurornitholestine dromaeosaurids (Fig. 3A—E, G)(Larson and Currie, 2013) and somewhat
unlike those previously discovered from the American east (Kiernan and Schwimmer, 2004;
Schwimmer et al., 2015; Brownstein, 2018a). Teeth assigned to *Saurornitholestes* have been
described from the Cretaceous of the southeastern United States (Kiernan and Schwimmer, 2004;
Schwimmer et al., 2015). These teeth are extremely small (< 6 mm), far less recurved than NJSMP
GP 22949, and have proportionally large denticles that are more strongly apically hooked (e.g.,
fig. 1 in Kiernan and Schwimmer, 2004). One tooth from Alabama measuring 4.9 mm in crown

height and preserving 7 distal denticles and 8 mesial denticles per mm is less recurved than
NJSM GP 22949 and far less elongate in labial and lingual views (see Kiernan and Schwimmer,
2004). Some saurornitholestine teeth from South Carolina (Schwimmer et al., 2015) also lack the
‘slender’ condition in NJSM GP 22949, where the mesiodistal width of the heavily recurved
tooth crown is much smaller than the crown height. Teeth from the Ellisdale site of New Jersey
include dromaeosaurid crowns (Denton et al., 2011; Brownstein, 2018c). However, large crowns
from Ellisdale are not ‘slender’ like NJSM GP 22949, are larger in size, and bear distal denticles
that are considerably more apically hooked than the Mt. Laurel tooth and western
saurornitholestines (Brownstein, 2018c). A dromaeosaurid tooth from North Carolina is slightly
larger than NJSM GP 22949, but is less recurved and far less slender (Brownstein, 2018b).

When compared to dromaeosaurid teeth from outside eastern North America, NJSM GP
22949 most closely resembles the teeth of western North American saurornitholestine
dromaeosaurids. Despite the fact that NJSM GP 22949 was placed in the convex hull formed by
the teeth of *Velociraptor* in the first discriminant analysis conducted, the tooth is unlike those of
the western North American velociraptorine *Acheroraptor temertyorum* (Evans et al., 2013) or
the Mongolian *Velociraptor mongoliensis* (Turner et al., 2012; Fig. 3F) in lacking strongly
developed striations along its crown surface, in being more strongly recurved, and in being far
more slender (lower CBL/CH value)(Fig. 2A—B, F)(Smith et al., 2005; Evans et al., 2013). In
contrast to *Dromaeosaurus albertensis*, and ‘dromaeosaurine’ teeth from western North
America, the mesial carina in NJSM GP 22949 is not twisted onto the mesiolingual face of the
crown, the distal denticles are apically hooked, and the tooth is more strongly recurved (Fig.
3H—I; Turner et al., 2012; Larson and Currie, 2013). The Mt. Laurel tooth is far less robust and
has far less developed carinae than the teeth of *Utahraptor* (Fig. 2H). The Mt. Laurel tooth is

also smaller, much more strongly recurved, and possesses denticles more apically hooked than
those of the giant Maastrichtian dromaeosaurid *Dakotaraptor steini* (DePalma et al., 2015).
NJSM GP 22949 lacks the ‘figure-8’ basal cross-section seen in the teeth of *Deinonychus*
(Ostrom, 1969; Brownstein, 2018b; pers. obs.). Although the strongly recurved maxillary teeth
of *Deinonychus* (Ostrom, 1969; Turner et al., 2012; pers. obs.) are somewhat comparable with
NJSM GP 22949, the differing basal cross-sections among these specimens and the slightly
asymmetrical morphology of the teeth in *Deinonychus* distinguish *D. antirrhopus* and the Mt.
Laurel form. The discriminant analysis of the Larson and Currie (2013) dataset supports
saurornitholestine affinities for NJSM GP 22949, classifying the tooth as a saurornitholestine
crown (Supplementary Information).

NJSM GP 22949 resembles the teeth of western North American Maastrichtian
saurornitholestines in having a slender, tall outline in labial and lingual views (the “Lancian”
saurornitholestine morphotype of Larson and Currie, 2013). The tooth is closely comparable with
the crowns of the juvenile saurornitholestine ‘*Bambiraptor feinbergi*,’ which are extremely
recurved and slender and possess apically-hooked denticles (Fig. 3G). A discriminant analysis of
the Late Cretaceous western North American paravian tooth dataset of Larson and Currie (2013)
found NJSM GP 22949 to nest within the convex hulls formed by four tooth morphotypes
(Saurornitholestinae, Dromaeosaurinae, *Zapsalis*, and *Atrociraptor*). NJSM GP 22949 is also
quantifiably unlike YPM VPPU.021397, the large dromaeosaurid tooth from the Campanian of
North Carolina (Brownstein, 2018b), plotting far from the southeastern North American
specimen in both morphometric analysis in which these teeth were included (Fig. 4). Thus,
NJSM GP 22949 is most comparable to the crowns of a saurornitholestine-like dromaeosaurid.
Saurornitholestines are small-bodied dromaeosaurids (Turner et al., 2012), and so NJSM GP

22949 is important for indicating members of this group may have achieved relatively large body
sizes for dromaeosaurs.

**Description.**

NJSM GP 22949 is the complete crown of a dromaeosaurid dinosaur. Measurements of
this specimen are in Table 1. This tooth is heavily recurved, displaying the ziphodont condition.
The crown possesses an ovoid basal cross-section. In distal view, the middle portion of tooth is
convex labially, although the crown becomes labiolingually straightened towards its apex. The
labial and lingual surfaces are flattened, and the lack of a root attached to this crown indicates it
was shed. Although both the mesial and distal carinae are preserved, the mesial denticles have
been mostly eroded away, and precise denticle counts for the mesial carina are unable to be
taken. Some portions of the tooth crown are cracked, and the outer enamel layer is poorly
preserved towards the distal end of the specimen. Small portions of the middle of the crown are
missing. The distal profile of NJSM GP 22949 is strongly concave. The preserved portions of the
outer enamel layer are smooth, although at the apex several slightly developed ridges appear.
These ridges could represent features of the original morphology of the tooth or be damage from
feeding or taphonomic processes. The distal carina preserves a large number of apically hooked
denticles that become smaller towards the apex of the crown. These denticles are separated by
interdenticular sulci that, along with the serrations, project slightly onto the tooth surface.
Unfortunately, the shape and density of the mesial denticles could not be determined, as the
mesial carina is heavily eroded in NJSM GP 22949.

**Discussion.**

The two theropod teeth described here add to one of the most complete Maastrichtian
faunas from eastern North America. The dromaeosaurid tooth NJSM GP 22949 is

biogeographically significance for being the first occurrence of this clade in the Mount Laurel
Formation and more generally Maastrichtian of eastern North America. Until now,
tyrannosauroids and ornithomimosaurids were the only known theropods from the late Campanian-
Maastrichtian of this area (Gallagher, 1993; Weishampel & Young, 1996; Gallagher, 1997;
Brusatte et al., 2011; Brusatte et al., 2012; Gallagher et al., 2014), with the latest records of
dromaeosaurids in the American east hailing from mid-Campanian units in the Carolinas
(Schwimmer et al., 2015; Brownstein, 2018b) and the Ellisdale site of New Jersey (Denton et al.,
2011). Although NJSM GP 22949 is most comparable to the crowns of mid-sized to largish
dromaeosaurids like *Deinonychus antirrhopus* and *Dakotaraptor steini* (Ostrom, 1969; DePalma
et al., 2015) and to small tyrannosauroids (Denton et al., 2011; Williamson and Brusatte, 2014;
Schwimmer et al., 2015) in its dimensions, it is most closely allied with dromaeosaurids in the
morphometric analyses conducted (Fig. 4) and in many key features of its morphology.

The tyrannosauroid tooth NJSM GP 14256 supports the presence of *Dryptosaurus*-like
tyrannosauroids in the early Maastrichtian of New Jersey. Isolated teeth and postcranial material
from the Mount Laurel were previously assigned to *Dryptosaurus* sp. based on little more than
their geographic proximity to the site where the holotype of this taxon was recovered
(Gallagher, 1993; Weishampel & Young, 1996; Gallagher, 1997), but no detailed description of
late Campanian to early Maastrichtian tyrannosauroids from New Jersey has appeared in the
literature. The well-preserved nature of NJSM GP 14256 thus allows for the formal recognition
of the presence of non-tyrannosauroid tyrannosauroids in the latest Campanian to earliest
Maastrichtian of the Atlantic Coastal Plain. Furthermore, the excellent condition of the outer
enamel layer of NJSM GP 12456 allows for further documentation of the dental anatomy of
Appalachian tyrannosauroids, the isolated teeth of which are often found highly abraded among

stream deposits (e.g., Weishampel & Young, 1996). The presence of non-tyrannosaurid
tyrannosauroids in the Mt. Laurel Formation is expected given the presence of *Dryptosaurus*
*aquilunguis* and non-tyrannosauroid tyrannosaurs of similar phylogenetic position in both the
middle-late Maastrichtian Navesink and New Egypt formations (Gallagher, 1993; Brusatte et al.,
2011) and early Campanian Marshalltown Formation (Denton et al., 2011). However, that NJSM
GP 14256, originally discovered in 1984, is only described now attests to the understudied nature
of these deposits.

The late recognition of dromaeosaurids in the Maastrichtian sediments of New Jersey is
notable, given that the dinosaurs of the Mt. Laurel and other Cretaceous units in the Atlantic
Coastal Plain have been studied for over a century and a half (e.g., Leidy, 1858; Cope, 1866;
Gallagher, 1993; Weishampel & Young, 1996; Gallagher, 1997). Teeth from locations like the
Ellisdale site of the Marshalltown Formation of New Jersey originally assigned to tyrannosaurs
have more recently been reclassified as the crowns of dromaeosaurids (Gallagher et al., 1986;
Gallagher, 1993, 1997; Denton et al., 2011), so it is entirely possible that the lack of diversity in
dinosaur faunas from the Maastrichtian of the Atlantic Coastal Plain reflects systematic
misidentification of these isolated fossils. Only further work on Appalachian fossils will allow
for more comprehensive revision of the identification of fossils from this area.

During the Cretaceous, terrestrial faunas became more regionalized as the breakup of
supercontinents like Gondwana and Laurasia and the inundation of smaller landmasses like
North America and Europe occurred (e.g., Russell, 1995; Sereno et al., 1996; Csiki et al., 2010;
Sampson et al., 2010; Csiki-Sava et al., 2015). In particular, the faunas of Appalachia and
Laramidia, which became isolated from each other as the Western Interior Seaway flooded the
middle of North America (e.g., Russell, 1995; Roberts and Kirschbaum, 1997; Schwimmer,

2002; Sampson et al., 2010), have been recognized as highly distinctive (e.g., Gallagher, 1993;
Weishampel and Young, 1996; Gallagher, 1997; Sampson et al., 2010; Denton et al., 2011;
Schwimmer et al., 2015). In the past thirty years, a handful of discoveries from the eastern
margin of North America have indicated some faunal interchange occurred between Laramidian
and Appalachian dinosaur communities during the latest Maastrichtian. These include the tooth
of a ceratopsid from the latest Maastrichtian of Mississippi and the fragmentary forelimb
material of possible lambeosaurines from the Maastrichtian of New Jersey and earliest
Maastrichtian of Nunavut, Canada (Gallagher, 1995; Gallagher, 1997; Farke and Phillips, 2017).
At the same time, Appalachian faunas continued to harbor endemic forms like intermediate
tyrannosauroids, represented by *Dryptosaurus aquilunguis* and comparable forms through the
Campanian-Maastrichtian transition in the Atlantic Coastal Plain (Gallagher, 1993; Brusatte et
al., 2011; this paper). Faunal interchange in the last 10 million years of the Mesozoic seems to
have occurred throughout the northern hemisphere. Phylogenetic evidence strongly posits that
the presence of *Tyrannosaurus rex* in the Maastrichtian of the western United States and Canada
represents a dispersal of Asian tyrannosaurids into the Americas (Brusatte et al., 2010; Brusatte
and Carr, 2016). A dispersal event between Asian and North American faunas during the
Maastrichtian may have occurred over Beringia (Fiorillo, 2008), given the similarity of
Maastrichtian polar faunas from Russia, Alaska, and lower latitudes in the United States and
Canada (e.g., Weishampel et al., 2004; Gangloff, 2012; Godefroit et al., 2012; Le Loeuff, 2012).
Along with previous discoveries, the tyrannosauroid tooth described here supports the ‘refugium’
model for eastern North America, wherein taxa more closely allied with middle Cretaceous
forms (e.g., non-tyrannosaurid tyrannosauroids like *Dryptosaurus* and *Appalachiosaurus*)(Carr et

al., 2005; Brusatte et al., 2011; Brusatte and Carr, 2016) persisted in relative isolation as more
derived forms evolved in Laurasia.

**Conclusions.**

Two theropod teeth from the Campanian-Maastrichtian Mount Laurel Formation of New
Jersey are described in detail. The dromaeosaurid tooth, which plots with western North
American saurornitholestine teeth in principle components and discriminant analyses, is the
youngest record of a non-avian maniraptoran from eastern North America and the first from the
latest Campanian-Maastrichtian of the American east. This tooth provides another record of a
mid-sized to large dromaeosaurid in the Cretaceous of eastern North America. However, this
tooth is more allied with those of saurornitholestines and velociraptorines than with
*Deinonychus*, dromaeosaurines, or largish dromaeosaurid teeth previously described from
Appalachia, tentatively suggesting that several types of dromaeosaurids might have grown to
relatively large sizes in the Cretaceous of the eastern United States and indicating mid-sized to
largish dromaeosaurids were a usual component of Appalachian faunas. The tyrannosauroid
tooth is the first specimen to suggest the presence of *Dryptosaurus aquilunguis* or a closely
related tyrannosauroid in the Mt. Laurel ecosystem, further supporting the refugium model for
Appalachian vertebrate evolution.

**Methods.**

Measurements and nomenclature.

Measurements of both teeth were taken in accordance with the methodology of Smith et al.

(2005) and Larson and Currie (2013). The dimensions of the Mt. Laurel teeth were determined

using digital calipers. I follow the nomenclature of Hendrickx et al. (2015) when describing the

two teeth on which this paper focuses.

Principle components analyses.

In order to provide support for the assignment of the dromaeosaurid tooth to a specific
group of theropod dinosaurs, I included it in principle components and discriminant analyses
conducted in the program PAST v. 3.18 (Hammer et al., 2001). In order to assess the
morphological similarity of the Mt. Laurel teeth to theropod clades present in the Cretaceous of
the northern hemisphere, I used a modified version (Brownstein, 2018b) of the dataset of Smith
et al. (2005) that includes tooth data on tyrannosauroids, troodontids, and dromaeosaurids. A
principle components analysis was run on this dataset, which included data on fifteen
measurements: crown height (CH), crown base length (CBL), crown base width (CBW),
apicobasal length (length of the tooth along the longest apicobasal axis), and serration density
415 per 5mm for the basal (DB), mid-crown (DC), and apical (DA) distal carina. An additional
416 principle components analysis (PCA) was conducted using the dataset of Larson and Currie
(2013) in order to better assess the similarity of the Mt. Laurel dromaeosaurid specimen to other
North American paravian teeth. This principle components analysis assessed for five
measurements: CH, CBL, CBW, and the mesial (MD) and distal (DD) denticles per millimeter.
The summary statistics and loadings from the results of the PCAs conducted are included in the
Supplementary Information.

Discriminant analyses.

To further assess the affinities of the Mt. Laurel dromaeosaurid tooth, I performed a discriminant
analysis on the tooth datasets of Smith et al. (2005, modified in Brownstein, 2018b) and Larson
and Currie (2013). This analysis creates a morphospace by maximally separating objects sorted
into pre-determined groups. This analysis was also run in PAST v. 3.18 (Hammer et al., 2001),
and the loadings and confusion matrices can be found in the supplementary information.

**Phylogenetic analysis.**

To provide additional support for the referral of the incomplete tyrannosauroid tooth to that

family, I coded the specimen for the phylogenetic matrix of Hendrickx and Mateus (2014), a

dataset of theropod dentition that includes 64 taxa/specimens coded for 141 characters. The

matrix was entered into the phylogenetics program TNT 1.5 (Goloboff and Catalano, 2016) for a

phylogenetic analysis. The matrix was first analyzed using the “New Technology Search,” with

default parameters for ratchet, tree drift, tree fuse, and sectorial search. A total of 10 trees of

length 688 were retained. These topologies were then subjected to traditional (TBR) branch

swapping, which allows for a more extensive exploration of each tree island. This found

over >99,999 most parsimonious topologies of 688. These were summarized in a strict consensus

topology.

**Acknowledgements.**

I thank Dana Ehret for access to the collections of the New Jersey State Museum, Carl Mehling

for access to the collections of the American Museum of Natural History, and Dan Brinkman for

access to the collections of the Yale Peabody Museum of Natural History. I thank Howard

Falcon-Lang, Christophe Hendrickx, Terry Gates, Kevin Padian, and four anonymous reviewers

for their comments, which greatly improved the quality of this paper.

**Availability of data and material.**

All data is available in the Supplementary Information.

**Competing interests.**

The author declares no competing interests.

**Funding.**

The author received no funding for this work.

**References.**

Alvarez, L.W., Alvarez, W., Asaro, F., Michel, H.V. 1980. Extraterrestrial cause for the
Cretaceous-Tertiary extinction. *Science* **208**:1095–1108.

Brochu, C.A. 2003. Osteology of *Tyrannosaurus rex*: insights from a nearly complete skeleton
and high-resolution computed tomographic analysis of the skull. *Society of Vertebrate
Paleontology Memoir* **7**:1–138.

Brownstein, C.D. 2018a. The biogeography and ecology of the Cretaceous non-avian dinosaurs
of Appalachia. *Palaeontologia Electronica* **21.1.5A**:1–56.

Brownstein, C.D. 2018b. A large dromaeosaurid from North Carolina. *Cretaceous Research*
**92**:1–7.

Brownstein, C.D. 2018c. The distinctive theropod assemblage of the Ellisdale site of New Jersey
and its implications for North American dinosaur ecology and evolution during the Cretaceous.
*Journal of Paleontology* **92**(5):1115—1129.

Brusatte, S.L., Benson, R.B., Norell, M.A. 2011. The Anatomy of *Dryptosaurus aquilunguis*
(Dinosauria: Theropoda) and a Review of its Tyrannosauroid Affinities. *American Museum
Novitates* **3717**:1–53.

Brusatte, S.L., Benson, R.B.J., Carr, T.D., Williamson, T.E., and Sereno, P.C. 2007. The
systematic utility of theropod enamel wrinkles. *Journal of Vertebrate Paleontology* **27**: 1052-
1056.
Brusatte, S.L., Butler, R.J., Barrett, P.M., Carrano, M.T., Evans, D.C., Lloyd, G.T., Mannion,
P.D., Norell, M.A., Peppe, D.J., Upchurch, P., and Williamson, T. E. 2015. The extinction of the
dinosaurs. *Biological Reviews* **90**:628–642.
Brusatte, S.L., Butler, R.J., Prieto-Márquez, A., Norell, M.A. 2012. Dinosaur morphological
diversity and the end-Cretaceous extinction. *Nature Communications* **3**:804.
Brusatte, S.L., Carr, T.D. 2016. The phylogeny and evolutionary history of tyrannosauroid
dinosaurs. *Scientific Reports* **6**(1):20252.
Brusatte, S.L., Norell, M.A., Carr T.D., Erickson, G.M., Hutchinson, J.R., Balanoff, A.M.,
Bever, G.S., Choiniere, J.N., Makovicky, P.J., Xu X. 2010. Tyrannosaur paleobiology: new
research on ancient exemplar organisms. *Science* **329**(5998):1481-1485.
Burns, M.E., and Ebersole, J.A. 2016. Juvenile Appalachian nodosaur material (Nodosauridae,
Ankylosauridae) from the lower Campanian lower Mooreville Chalk of Alabama. Abstracts and
Posters Session IV, Society of Vertebrate Paleontology, 76th Annual Meeting.

Carr, T.D., Williamson, T.E., Schwimmer, D.R. 2005. A new genus and species of
tyrannosauroid from the Late Cretaceous (middle Campanian) Demopolis Formation of
Alabama. *Journal of Vertebrate Paleontology* **25**(1):119–143.
Cope, E.D. 1866. Discovery of a gigantic dinosaur in the Cretaceous of New Jersey. *Proceedings*
*of the Academy of Natural Sciences of Philadelphia* **18**: 275–279.
Csiki-Sava, Z., Buffetaut, E., Ósi, A., Pereda-Suberbiola, X., and Brusatte, S.L. 2015. Island life
in the Cretaceous - Faunal composition, biogeography, evolution, and extinction of land-living
vertebrates on the Late Cretaceous European archipelago. *ZooKeys* **469**:1–161.
Csiki, Z., Vremir, M., Brusatte, S.L., Norell, M. A. 2010. An aberrant island-dwelling theropod
dinosaur from the Late Cretaceous of Romania. *Proceedings of the National Academy of*
*Sciences* **107**(35):15357–15361.
Currie P.J., Rigby J.K., Sloan R.E., 1990, Theropod teeth from the Judith River Formation of
Southern Alberta, Canada. In: Carpenter, K., and Currie, P.J., eds: *Dinosaur Systematics:*
*Approaches and Perspectives*. Cambridge University Press, p. 107–125.
Currie, P. J. and Varricchio, D.J. 2004. A new dromaeosaurid from the Horseshoe Canyon
Formation (Upper Cretaceous) of Alberta, Canada. In: Currie, P.J., Koppelhus, E.B., Shugar,
517 M.A., and Wright, J.L., eds: *Feathered Dragons*. Indianapolis: Indiana University Press. p. 112–
518 132.

Denton R.K., O'Neill R.C., Grandstaff B.S., and Parris D.C. 2011. The Ellisdale Site (Late
Cretaceous, Campanian) - is there a rationale for an "Ellisdalean" land faunal age? *Geological*
*Society of America Abstracts with Programs* **43**:85.
DePalma R.A., Burnham D.A., Martin L.D., Larson P.L., and Bakker R.T. 2015. The First Giant
Raptor (Theropoda: Dromaeosauridae) from the Hell Creek Formation. *Paleontological*
*Contributions* **14**:1–16.
Farke, A.A. and Phillips, G.E. 2017. The first reported ceratopsid dinosaur from eastern North
America (Owl Creek Formation, Upper Cretaceous, Mississippi, USA). *PeerJ* **5**:e3342.
Fiorillo, A.R. 2008. Cretaceous dinosaurs of Alaska: Implications for the origins of Beringia. In:
Blodgett, R.B., and Stanley, G. eds: The Terrane Puzzle: new perspectives on paleontology and
stratigraphy from the North American Cordillera. *Geological Society of America Special Paper*
**442**: 313-326.
Frederickson, J., Engel, M.H., Cifelli, R. 2018. Niche Partitioning in Theropod Dinosaurs: Diet
and Habitat Preference in Predators from the Uppermost Cedar Mountain Formation (Utah,
U.S.A.). *Scientific Reports* **8**(1):17872.
Gallagher, W.B. 1993. The Cretaceous-Tertiary mass extinction event in North Atlantic Coastal
Plain. *The Mosasaur* **5**:75-154.

Gallagher, W.B. 1995. Evidence of juvenile dinosaurs and dinosaurian growth stages in the Late
Cretaceous deposits of the Atlantic Coastal Plain. *Bulletin of the New Jersey Academy of Science*
**40**:5–8.
Gallagher, W.B. 1997. When Dinosaurs Roamed New Jersey. Rutgers University Press, New
Brunswick, New Jersey.
Gallagher, W.B., Camburn, J., Camburn, S., and Hanczaryk, P.A. 2014. Taphonomy of a Late
Campanian Fossil Assemblage at Marlboro, Monmouth County, New Jersey. *The Mosasaur*
**8**:53–68.
Gangloff, R.A. Dinosaurs under the Aurora. Bloomington, Indiana University Press, 2012.
Gauthier, J. 1986. Saurischian monophyly and the origin of birds. *Memoirs of the California*
*Academy of Sciences* **8**:1–55.
Godefroit, P., Bolotsky, Y.L., Lauters, P. 2012. A New Saurolophine Dinosaur from the Latest
Cretaceous of Far Eastern Russia. *PLoS ONE* **7**(5):e36849.
Goloboff, P., Catalano, S., 2016. TNT version 1.5, including full implementation of phylogenetic
morphometrics. *Cladistics* **32**(3): 221-238.

Hammer, Ø., Harper, D.A.T., Ryan, P. D. 2001. Paleontological statistics software package for
education and data analysis. *Palaeontologia Electronica* **4**:1–9.
Hendrickx, C., Mateus, O., 2014. Abelisauridae (Dinosauria: Theropoda) from the Late Jurassic
of Portugal and dentition-based phylogeny as a contribution for the identification of isolated
theropod teeth. *Zootaxa* **3759**(1):1–74.
Hendrickx, C., Mateus, O., Araújo, R., 2015. A proposed terminology of theropod teeth
(Dinosauria, Saurischia). *Journal of Vertebrate Paleontology* **35**(5):e982797.
Kiernan, K. and Schwimmer, D.R. 2004. First record of a velociraptorine theropod (Tetanurae,
Dromaeosauridae) from the eastern Gulf Coastal United States. *The Mosasaur* **7**:89–93.
Krumenacker, L.J., Simon, D.J., Scofield, G., and Varricchio, D.J. 2017. Theropod dinosaurs
from the Albian-Cenomanian Wayan Formation of eastern Idaho. *Historical Biology* **29**(2): 170-
186.
Langston, W. 1960. The vertebrate fauna of the Selma Formation of Alabama, part VI: the
dinosaurs. *Fieldiana: Geological Memoirs* **3**(5):315-359.
Larson, D.W., Currie, P.J. 2013. Multivariate Analyses of Small Theropod Dinosaur Teeth and
Implications for Paleoecological Turnover through Time. *PLoS ONE* **8**(1): e54329.

Le Loeuff, J. 2012. Paleobiogeography and biodiversity of Late Maastrichtian dinosaurs: how
many dinosaur species went extinct at the Cretaceous-Tertiary boundary? *Bulletin de la Société*
*Géologique de France* **183**:547–559.
Leidy J. 1858. Hadrosaurus foulkii, a new saurian from the Cretaceous of New Jersey, related
to Iguanodon. *Proceedings of the Academy of Natural Sciences* **10**:213-218.
Marsh, O.C. 1881. Principal characters of American Jurassic dinosaurs. Part V. *American*
*Journal of Sciences series 3* **21**:417–423.
Matthew W.D., and Brown, B. 1922. The family Deinodontidae, with notice of a new genus
from the Cretaceous of Alberta. *Bulletin of the American Museum of Natural History* **46**:367–
385.
Miller, H.W. 1967. Cretaceous vertebrates from Phoebus Landing, North Carolina. *Proceedings*
*of the Academy of Natural Sciences* **119**:219–239.
Miller, K.G., Sugarman, P.J., Browning, J.V., Kominz, M.A., Olsson, R.K., Feigenson, M.D.,
and Hernandez, J.C. 2004. Upper Cretaceous sequences and sea-level history, New Jersey
Coastal Plain. *Bulletin of the Geological Society of America* **116**(3):368–393.
Ostrom, J.H. 1969. Osteology of Deinonychus antirrhopus, an unusual theropod from the Lower
Cretaceous of Montana. *Peabody Museum of Natural History Bulletin* **30**:1–165.

Owen, R. 1842. Report on British Fossil Reptiles, Pt. II. *Report of the British Association for the*
*Advancement of Science* **11**:60–204.
Prieto-Márquez, A., Erickson, G.M., and Ebersole, J.A. 2016. A primitive hadrosaurid from
southeastern North America and the origin and early evolution of 'duck-billed' dinosaurs. *Journal*
*of Vertebrate Paleontology* **36**(2):e1054495.
Prieto-Márquez, A., Weishampel, D.B., and Horner, J.R. 2006. The dinosaur *Hadrosaurus*
*foulkii*, from the Campanian of the East Coast of North America, with a reevaluation of the
genus. *Acta Palaeontologica Polonica* **51**(1):77-98.
Roberts, L.N.R. and Kirschbaum, M.A. 1995. Paleogeography of the Late Cretaceous of the
Western Interior of middle North America-Coal distribution and sediment accumulation. *U.S.*
*Geological Survey Professional Paper* **1561**:1–115.
Russell, D.A. 1995. China and the lost worlds of the dinosaurian era. *Historical Biology* **10**(1):3–
12.
Sakamoto, M., Benton, M.J., and Venditti, C. 2016. Dinosaurs in decline tens of millions of
631 years before their final extinction. *Proceedings of the National Academy of Sciences* **113**:5036–
632 5040.

Samman T., Powell G.L., Currie P.J., and Hills, L.V. 2005. Morphometry of the teeth of western
North American tyrannosaurids and its applicability to quantitative classification. *Acta*
*Palaeontologica Polonica* **50**:757–776.
Sampson, S.D., Loewen, M.A., Farke, A.A., Roberts, E.M., Forster, C.A., Smith, J.A., and Titus,
639 A.L. 2010. New horned dinosaurs from Utah provide evidence for intracontinental dinosaur
endemism. *PLoS ONE* **5**(9):e12292.
Sarjeant, W.A.S., Currie, P.J. 2001. The “Great Extinction” that never happened: The demise of
the dinosaurs considered. *Canadian Journal of Earth Sciences* **38**(2):239–247.
Schwimmer, D.R., Sanders, A.E., Erickson, B.R., and Weems, R.E. 2015. A Late Cretaceous
dinosaur and reptile assemblage from South Carolina, USA. *Transactions of the American*
*Philosophical Society* **105**(2):1–157.
Schwimmer, D.R., Williams, G.D., Dobie, J.L., and Siesser, W.G. 1993. Late Cretaceous
dinosaurs from the Blufftown Formation in western Georgia and eastern Alabama. *Journal of*
*Vertebrate Paleontology* **67**(2):288–296.
Sereno, P.C., Dutheil, D.B., Iarochene, M., Larsson, H.C.E., Lyon, G.H., Magwene, P.M., Sidor,
C.A., Varricchio, D.J., Wilson, J.A. Predatory dinosaurs from the Sahara and Late Cretaceous
faunal differentiation. *Science* **272**:986–991.

Sloan, R.E., Rigby, J.K. Jr, Van Valen, L.M., Gabriel, D. 1986. Gradual dinosaur extinction and
simultaneous ungulate radiation in the hell creek formation. *Science* **232**(4750):629–633.
Smith, J.B. 2005. Heterodonty in *Tyrannosaurus rex*: Implications for the taxonomic and
systematic utility of theropod dentitions. *Journal of Vertebrate Paleontology* **25**:865–887.
Smith, J.B., Vann, D.R., Dodson, P. 2005. Dental morphology and variation in theropod
dinosaurs: implications for the taxonomic identification of isolated teeth. *The Anatomical Record*
**285A**:699–736.
Sues, H.-D .1978. A new small theropod dinosaur from the Judith River Formation (Campanian)
of Alberta Canada. *Zoological Journal of the Linnean Society* **62**:381–400.
Sugarman, P.J., Miller, K.G., Burky, D., and Feigenson, M.D. 1995. Uppermost Campanian-
Maastrichtian strontium isotopic; biostratigraphic and sequence stratigraphic framework of the
New Jersey Coastal Plain. *Geological Society of America Bulletin* **107**:19-37.
Turner A.H., Makovicky P.J., and Norell M.A. 2012. A Review of Dromaeosaurid Systematics
and Paravian Phylogeny. *Bulletin of the American Museum of Natural History* **371**:1–206.
von Huene F. 1914. Saurischia et Ornithischia Triadica (“Dinosuaria” Triadica).
*Animalia. Fossilium Catalogus* **4**:1-21.

Weishampel, D.B. 2006. Another look at the dinosaurs of the East Coast of North America, p.
129-168. In 'Coletivo Arqueológico-Paleontológico Salense, (eds.), Actas III Jornadas
Dinosaurios Entorno. Salas de los Infantes, Burgos, Spain.
Weishampel, D.B. and Young, L. 1996. Dinosaurs of the East Coast. Johns Hopkins University
Press, Baltimore, Maryland, USA.
Weishampel, D.B., Barrett, P.M., Coria, R.A., Loeuff, J.L., Xing, X., Xijin, Z., Sahni, A.,
Gomani, E.M.P., and Noto, C.R. 2004. Dinosaur Distribution. In: Weishampel, D.B., Dodson, P.,
and Osmólska, H. eds: The Dinosauria, 2nd Edition. University of California Press, Berkeley,
California, USA. p. 517-617.
Wick, S.L., Lehman, T.M., Brink, A.A. 2015. A theropod tooth assemblage from the lower
Aguja Formation (early Campanian) of West Texas, and the roles of small theropod and varanoid
lizard mesopredators in a tropical predator guild. *Palaeogeography, Palaeoclimatology,*
*Palaeoecology* **418**:229.
Williamson T.E., and Brusatte S.L. 2014. Small Theropod Teeth from the Late Cretaceous of the
San Juan Basin, Northwestern New Mexico and Their Implications for Understanding Late
Cretaceous Dinosaur Evolution. *PLoS ONE* **9**:e93190.

A

A

B

C

Appendix D

ROYAL SOCIETY OPEN SCIENCE

New records of theropods from New Jersey inform faunal interchange in Maastrichtian North America

Journal:	Royal Society Open Science
Manuscript ID	RSOS-191206
Article Type:	Research
Date Submitted by the Author:	10-Jul-2019
Complete List of Authors:	Brownstein, Chase; Stamford Museum and Nature Center,
Subject:	Palaeontology < EARTH SCIENCES, evolution < BIOLOGY, ecology < BIOLOGY
Keywords:	Dinosaur, Biogeography, Appalachia, Cretaceous
Subject Category:	Biology (whole organism)

**Author-supplied statements**

Relevant information will appear here if provided.

***Ethics***

*Does your article include research that required ethical approval or permits?:*

This article does not present research with ethical considerations

*Statement (if applicable):*

CUST_IF_YES_ETHICS :No data available.

***Data***

*It is a condition of publication that data, code and materials supporting your paper are made publicly*
*available. Does your paper present new data?:*

Yes

*Statement (if applicable):*

The data is included in the supplementary material.

***Conflict of interest***

I/We declare we have no competing interests

*Statement (if applicable):*

CUST_STATE_CONFLICT :No data available.

***Authors' contributions***

I am the only author on this paper

*Statement (if applicable):*

CUST_AUTHOR_CONTRIBUTIONS_TEXT :No data available.

Cover Letter

July 10, 2019

Dear editor of *Royal Society Open Science*,

I would like to resubmit to you my manuscript “New records of theropods from the latest Cretaceous of New Jersey and the Maastrichtian Appalachian fauna” for consideration at your journal.

In revising my manuscript, I have paid special attention to two major comments given by the reviewers. Firstly, I’ve removed the PCA of the dataset of Smith et al. (2005) with the tyrannosauroid tooth included on the advice of reviewer 2, provided additional justification for why the tooth is from a tyrannosauroid and probably not a tyrannosaurid, and provided additional information on the results of the phylogenetic analysis.

Secondly, I’ve redone the PCA and discriminant analyses including the dromaeosaurid tooth such that the analyses of the Larson and Currie dataset include the Tar Heel tooth and the discriminant analysis of the Smith et al. dataset does not include characters that could not be scored for the Mt. Laurel tooth.

Finally, I’ve removed the biogeographic speculations in line with the comments of the reviewers and editor. I agree that these were presented far too strongly, and I have reorganized the manuscript to act as a faunal characterization. I’ve also revised the manuscript in accordance with the many minor comments presented in reviewer 2’s annotated PDF.

I hope the manuscript is now suitable for publication, and thank you for your reviews.

Regards,

Chase Brownstein
Research Associate,
Collections & Exhibitions, Stamford Museum & Nature Center
Stamford, Connecticut, United States

Responses to Reviewers.

Chase D. Brownstein

However, your data and analyses either do not support your main claim or are insufficient to do so. Starting with your dataset, your measurements of NJSM GP 12456 for DB, DC, and DA do not appear congruent with the specimen figured in Fig. 1J, though do seem to match the scale of Fig. 1E. In Fig. 1J, measurements closer to 4 would appear to be more accurate than 2, 2, and 2.5. The scale bars for the photos and your measurements should be double-checked.

Done. I've double checked the scale bars to make sure they are correct. Please see the revised figure.

You also have a substantial amount of missing data for your new specimens in your dataset, and you haven't discussed how this was handled in your PCA. The default in PAST is to fill missing data with column average substitution, which in the Smith et al. (2005) dataset will bias all of your specimens towards the tyrannosaur part of your plot (as there are more tyrannosaurs in the dataset than any other group). Given that all three of the new teeth plot in areas of your plots where no other theropod teeth plot (partway between dromaeosaurs and tyrannosaurs), it appears that column average substitution of your dataset is having a substantial contribution to the position of your specimens on these plots. As well, eastern taxa are notably excluded. Dryptosaurus is not plotted in Fig. 2A, and the Tar Heel dromaeosaurid is not plotted in Fig. 4B and 4C, making the argument that either of these teeth have eastern or western affinities insufficiently supported by these analyses. Additions of those specimens and the exclusion of variables with missing data are necessary for these plots to be informative.

I appreciate this comment, and I have added measurements on the Tar Heel form to the datasets for analysis of the dromaeosaurid tooth. I've gone on the suggestion of reviewer 2 and not performed morphometric work on the tyrannosaur tooth. The affinities of that tooth to tyrannosaurs are still supported by several morphological comparisons and the phylogenetic analysis. I've also removed measurements that are not known for the new teeth to better perform the analysis. Moreover, instead of performing principle components analyses, I've performed a discriminant analysis of the Smith et al. (2005) dataset with characters unknown for the dromaeosaurid tooth removed to better support the referrals. Because the only character in the Larson and Currie dataset unknown for the dromaeosaurid tooth is that of the mesial denticle count, I left the dataset as is for that morphometric analysis.

*In the phylogenetic analysis, you only list shared derived characters between your tyrannosaur specimen and *T. rex*, but do not list shared derived characters that unite all tyrannosauroids. However, even this analysis does not support your argument that the specimen is a non-tyrannosaurid tyrannosauroid with affinities with *Dryptosaurus*. Also, one of the four characters you do list, 94) biconvex apical distal denticles, is not mentioned in your description at all nor is it visible in Fig. 1.*

I have added the shared derived characters that unite all tyrannosauroids to the paper's
text and noted the presence of biconvex denticles along the entirety of the carina.

*As well, Fig. 1B, 1C, and 1D depict fossils that are not described in the manuscript. If*
*these specimens are figured in this manuscript, they should be described systematically*
*in the text.*

These fossils have been described in previous papers and are there to give the reader a
visual map of what fossils are known from the Maastrichtian of NJ.

*Alternatively, these figures could easily be excluded from the manuscript. The entire*
*datasets used for analysis, including the Smith et al. (2005) and Larson and Currie*
*(2013) should be reproduced in the supplemental data for ease of replication.*

Done. I've added the datasets to the Supplemental Data.

*This description is of specialist interest given the new records from eastern North*
*America, but any discussion of faunal interchange is premature. In your description, you*
*have demonstrated that you have teeth of a dromaeosaurid and a tyrannosauroid from*
*this formation, but your analyses in their current form are insufficient to demonstrate that*
*these teeth have any biogeographic affinities.*

Done. I've modified the paper to focus on the significance of these teeth for contributing
to our understanding of Appalachian faunas, and removed the biogeographic
speculations.

Reviewer: 2

Comments to the Author(s)

*This paper is important for the acknowledgement of the taxa found in this New Jersey*
*formation. Outside of this finding there is much more difficulty in acquiring significance.*
*This problem of lack of greater significance is not the fault of the author but the nature of*
*the Appalachian dinosaur record. Fossiliferous formations are poorly dated, fossils are*
*not terribly abundant, and when known are typically in rough condition. Because there is*
*not a proper framework for the dinosaurs, determining faunal interchange is*
*problematic...you need to know what is around when, and the organism's phylogenetic*
*relationships. Without this it is difficult to say anything concrete about paleobiology.*
*Especially, in these days when doing rigorous biogeographic analyses is so easy. In my*
*opinion you can't really say anything about faunal interchange until a solid, dated fossil*
*record is assembled.*

I agree. See my comment above.

*First, I firmly do not think that a phylogenetic analysis should be used on a single tooth,*
*much less a single partial tooth. The phylogenetic analysis of teeth is not a proper use*
*of phylogenetics in my opinion at any rate. To my mind this part of the paper should be*
*eliminated.*

This part of the paper was recommended by another researcher who looked at my
paper. As it follows in line with many recent studies (e.g., Hendrickx et al., 2014; Young
et al., 2014), I have kept it for the sake of making this paper comparable to other, recent
works.

*Second, the author should use a more recent ordinal analysis of teeth. This is not a deal*
*breaker, but the primary dataset chosen is 13 years old (or 5 yrs old in the case of*
*Larson and Currie) and many others have been produced since then.*

Since the Larson and Currie dataset continues to be the one used the most often for
morphometric analysis of Cretaceous North American theropod teeth (see Evans et al.,
2013; Williamson and Brusatte, 2014 for examples), I am uncertain what the reviewer
here refers to. The Hendrickx et al. (2015) theropod tooth dataset only represents an
extension of the Smith et al. (2005) dataset that includes the Larson and Currie dataset
and removes a number of continuous characters.

*Third, the tyrannosaur tooth should absolutely be removed from the PCA analysis. It is a*
*fragmentary tooth, therefore any measurements taken are not going to be the correct*
*measurement and will provide erroneous results. Or there will be too little data to say*
*much about.*

Done. I've removed the tooth.

*Fourth, the dromaeosaurid tooth falls outside the dromaeosaurids in the Smith/*
*Brownstein matrix. And is just part of the miasma of teeth in the Larson and Currie*
*matrix. I would suggest pulling individual taxa from the analyses and talk about*
*comparisons with them instead of consistently saying it is a Saurornitholestesine*
*theropod. Remember that the PCA is using data you input. If there is more information*
*supporting your taxonomic claim that is not included in the quantitative data, then you*
*should discuss this in the text. In PAST you can change the symbol and the color,*
*meaning that you can label individual dinosaur taxa and the higher level taxonomic*
*groups at the same time. This should be done to help you see which species the Tar*
*Heel and Mt Laurel specimens are mostly similar to.*

Done. I've instead performed discriminant analyses on the datasets and taking up your
suggestion of visualizing the resulting plots using axes 2 and 3. The Mt. Laurel tooth
plots firmly within the hull formed by the teeth of *Velociraptor* in the first DA and firmly
with

*Fifth, this is very important! When using PCA on variables that are measurements, the*
*first axis is almost invariably a reflection of size, not shape. Therefore, you need to show*
*axes 2 and 3. You are welcome to retain axis 1 and 2, but 2 and 3 must be shown in*
*addition. You may also want to consider Centering or z-transforming your data because*
*it can help remove the effect of size, bring outliers closer together, and reflect shape*
*better.*

Done.

**New records of theropods from the latest Cretaceous of New Jersey and the Maastrichtian**

**Appalachian fauna**

Chase Doran Brownstein

Research Associate, Dept. of Collections and Exhibitions, Stamford Museum and Nature Center,

Stamford, CT, chasethedinosaur@gmail.com

Abstract.

The faunal changes that occurred in the few million years before the Cretaceous-Paleogene
extinction are of much interest to vertebrate paleontologists. Western North America preserves
arguably the best fossil record from this time, whereas terrestrial vertebrate fossils from the
eastern portion of the continent are usually limited to isolated, eroded postcranial remains.
Examination of fragmentary specimens from the ~~American east~~, which was isolated for the
majority of the Cretaceous as the landmass Appalachia, is therefore important for better
understanding dinosaur diversity at the end of the Mesozoic. Here, I report on two theropod teeth
from the Mount Laurel Formation, a lower-middle Maastrichtian unit from northeastern North
America. One of these preserves in fine-detail the structure of the outer enamel and resembles
the dentition of the tyrannosauroid *Dryptosaurus aquilunguis* among latest Cretaceous forms in
being heavily mediolaterally compressed and showing many moderately developed enamel
crenulations. Along with previously reported tyrannosauroid material from the Mt. Laurel and
overlying Cretaceous units, this fossil supports the presence of non-tyrannosaurid
tyrannosauroids in the Campanian-Maastrichtian of eastern North America. The other tooth is
assignable to a dromaeosaurid and represents both the youngest occurrence of a non-avian
maniraptoran in eastern North America and the first from the Maastrichtian reported east of the
Missisipi. This tooth, which belonged to a medium-sized dromaeosaurid based on size
comparisons with the teeth of taxa for which skeletons are known, increases the diversity of the
Maastrichtian dinosaur fauna of Appalachia. Along with previously reported dromaeosaurid
teeth, the Mt. Laurel specimen supports the presence of mid-sized to large dromaeosaurids in
eastern North America throughout the Cretaceous. This indicates that dromaeosaurids

approaching or exceeded  3 meters in length were more common in the Late Cretaceous than
previously thought.

Keywords: Dinosaur; Appalachia; Cretaceous; Fauna ; ~~Tyrannosaur~~; Teeth

**Introduction.**

The extinction of the non-avian dinosaurs at the end of the Mesozoic Era is a topic that
has continued to intrigue vertebrate paleontologists (e.g., Alvarez et al., 1980; Sloan et al., 1986;
Sarjeant and Currie, 2001; Le Loeuff 2012, Brusatte et al., 2012, 2015; Sakamoto et al., 2016).
However, a poor global terrestrial record from the Maastrichtian has hindered attempts to assess
the diversity dynamics of important groups like the Dinosauria during this period (e.g., Le
Loeuff, 2012; Brusatte et al., 2015). Western North America preserves arguably the most well-
characterized vertebrate record from the last 20 million years of the Mesozoic Era (Brusatte et
al., 2015), whereas that from the eastern portion of the continent is far more obscure. During the
majority of the Late Cretaceous, eastern and western North America were separated, the former
existing as a rectangular landmass called Appalachia. Appalachian dinosaur faunas included
intermediate-grade tyrannosauroids (Carr et al., 2005; Brusatte et al., 2011), basal hadrosaurids
and deri  non-hadrosaurid hadrosauroids (Langston, 1960; Prieto-Márquez et al., 2006; Prieto-
Márquez et al., 2016), nodosaurids (Gallagher, 1993; Burns and Ebersole, 2016), and
ornithomimosaur (Miller, 1967; Schwimmer et al., 1993; Weishampel and Young, 1996;
Schwimmer et al., 2015; Brownstein, 2018a).

Despite the amount of knowledge of Cretaceous faunal change to be gleaned from the
fossil record of Appalachia, the assemblages of this landmass have remained fundamentally
understudied since the mid-19th century (e.g., Weishampel & Young, 1996; Gallagher, 1997;

Weishampel, 2006; Brownstein, 2018a). The scarcity of terrestrial sedimentary units known from
the eastern half of the United States has also contributed to the obscurity of Appalachian faunas
compared to western North American ones (Weishampel and Young, 1996; Gallagher, 1997;
Carr et al., 2005; Weishampel, 2006; Brusatte et al., 2011). Only in the past few years have
indications of faunal changes in the latest Cretaceous (late Campanian-Maastrichtian) of the
~~American east~~ come, and all from isolated, fragmentary finds. Although a ceratopsian tooth from
the uppermost Maastrichtian of Mississippi (Farke and Phillips, 2017) and possible
lambeosaurine bones from the upper Maastrichtian of New Jersey (Gallagher, 1993) have
revealed that faunal exchanges probably occurred between Appalachia and Laramidia following
the regression of the Western Interior Seaway in the latest Campanian-earliest Maastrichtian, the
timing of these events remains poorly constrained. Further sampling of enigmatic assemblages
from the Maastrichtian, such as those of the eastern United States, is therefore important for
understanding faunal change in latest Mesozoic North America.

In the Campanian-Maastrichtian of New Jersey, a set of formations corresponds to a
period of transgressions and regressions of the Atlantic Ocean (e.g., Gallagher, 1993; Sugarman
et al., 1995; Miller et al., 2004; Gallagher et al., 2014). The majority of these Cretaceous units
are known for producing marine vertebrate and invertebrate fossils (Gallagher, 1993), although
some, such as the Woodbury and New Egypt formations, are notable for producing some of the
first partial dinosaur skeletons from the Americas (e.g., Leidy, 1858; Cope, 1866; Gallagher,
1993, Weishampel and Young, 1996; Gallagher, 1997). One of the most fossiliferous of these
formations is the Mount Laurel Formation, which is either uppermost Campanian or lowermost
Maastrichtian (Miller et al., 2004) and in New Jersey has produced the remains of several groups
of dinosaurs, including hadrosos, tyrannos, and ornithomimosaur (Gallagher, 1993;

Gallagher, 2014). Because of the sheer diversity of the community represented in the Mt. Laurel,
the formation serves as a window into Campanian-Maastrichtian eastern North American faunas.
However, the terrestrial fossils it produces are often eroded postcranial fragments (Gallagher,
2014).

Here, I describe some  theropod teeth from the Mt. Laurel Formation of New Jersey.
These include a large tooth assignable to a 6—8m tyrannosauroid and a smaller, heavily
recurved one assignable to a 3—4 m dromaeosaurid. These teeth are among the most diagnostic
records of theropods from the Mt. Laurel Formation, allowing for a more precise understanding
of the faunal composition and ecology of the eastern seaboard during the Maastrichtian, a
globally under-sampled time period (Le Loeuff, 2012; Brusatte et al., 2012).

**Results.**

Geological setting.

Both theropod teeth described here were collected from sediments of the Mount Laurel
Formation (Gallagher, 1993; pers. obs.), a marine deposit that represents a regression of the
Atlantic Ocean during the Late Cretaceous ~~period~~  and is the oldest unit included in the
Monmouth Group (Gallagher, 1993; Miller et al., 2004). The tyrannosauroid tooth described
here, NJSM GP 12456, was recovered from Big Bro  (Fig. 1A), a highly fossiliferous locality
famous for producing an extensive marine fauna (Gallagher, 1993; Weishampel and Young,
1996). At Big Brook, the stratigraphic column is exposed along the banks, with the Wenonah
Formation grading into the Mt. Lau  such that the border between the two are indistinguishable
(Gallagher, 1993). The contact between the Mt. Laurel and the overlying Navesink Formation is
an unconformity (Miller et al., 2004; Gallagher et al., 2014). The Mt. Laurel Formation appears
as gray to dark brown, pebbly quartz sands. The Big Brook tyrannosauroid tooth (Fig. 1E—H) is

unusual among the terrestrial vertebrate teeth collected from the site in possessing a well-
preserved enamel surface. When  other terrestrial vertebrate fossils from Big Brook are known
for being heavily water-worn and lacking morphological details, NJSM GP 12456 preserves both
its outermost enamel layer and many of its denticles.

NJSM GP 22949, the dromaeosaurid tooth, was recovered from Mt. Laurel deposits in
Burlington County, New Jersey (Fig. 1A). In this area, which makes up a portion of the
southwestern-most range of the Monmouth Group, the sands of the Mt. Laurel are more
glauconitic than farther north and are intermixed with iron compounds (Gallagher, 1993). The
thickness of this unit is also far greater to the southwest of its range (e.g., Gallagher, 1993).

Tyrannosauroid tooth.

Dinosauria Owen 1842

Theropoda Marsh 1881

Coelurosauria von Huene 1921

Tyrannosauroida Osborn 1905

Tyrannosauroida indet.

Material.

New Jersey State Museum collections (NJSM) GP 14256, the partial tooth of a large theropod
dinosaur (Fig. 1E—H).

Locality and Horizon.

134 Mt. Laurel Formation sediments at Big Brook, Monmouth County, New Jersey, latest

Campanian to early Maastrichtian (Gallagher, 1993; Miller et al., 2004; Gallagher et al., 2014).

Identification.

NJSM GP 14256 (Fig. 1E—H) closely resembles the dentition of tyrannosauroid
theropods in several respects. The tooth resembles those of ad¹¹ tyrannosauroids in its size,
which is closely comparable to tyrann¹²osaur crowns known from both western and eastern North
America (Currie et al., 1990; Brochu, 2003; Samman et al., 2005; Smith, 2005; Brusatte et al.,
2011; Williamson and Brusatte, 2014). In addition to its size, the Mt. Laurel tooth resembles
those of tyrann¹⁵ours to the exclusion of other theropods known from Late Cretaceous North
America in possessing a combination of packed denticles (2–2.5/mm) on its distal carina (15+
144 mm), the presence of denticles along both carinae, its slight, rather than pronounced, curvature,
the presence of numerous transverse undulations (density = 2/mm) on its main surface, the
presence of slightly biconvex denticle outlines for denticles all along the tooth (Fig. 1J), and its
smooth but slightly irregular surface texture (Fig 1E—H)(Currie et al., 1990; Brochu, 2003;
Samman et al., 2005; Smith, 2005; Brusatte et al., 2011; Williamson and Brusatte, 2014).
However, despite the size of the Mt. Laurel tooth, NJSM GP 14256 is notably unlike the teeth of
tyrannosaurids, for which incre¹⁶te teeth are a synapomorphy (e.g., Williamson and Brusatte,
2014; Brusatte and Carr, 2016). Instead, NJSM GP 14256 shows is highly mediolaterally
compressed and possesses a lens-shaped basal cross-section, indicative it came from a
tyrannosauroid outside Tyrannosauridae. Among large Late Cretaceous tyrann¹⁷ours, only
*Dryptosaurus aquilunguis* from the Maastrichtian New Egypt Formation of New Jersey is known
to possess a combination of high mediolateral compression and tyrannosaurid-like features of the
denticles and tooth surface (e.g., Brusatte et al., 2011). NJSM GP 14256 is also comparable with
the mediolaterally compressed teeth of *Dryptosaurus aquilunguis* in its dimensions, curvature,
and enamel crenulations (Brusatte et al., 2011; pers. obs. of YPM PU 22208). Given the Mt.

Laurel tooth's very close sp^{ecies} temporal proximity to the holotype of *Dryptosaurus*, I suggest the
tooth belongs to a closely related form.

Given the number of measurements unable to be taken from this tooth, I did not include it
in a morphometric analysis. However, phylogenetic analysis of the tooth within the dataset of
Hendrickx and Mateus (2014) found NJSM GP 14256 to be the sister taxon of *Tyrannosaurus*
*rex* in a clade united by four characters. These are characters 94 (biconvex apical denticles
present on distal carinae of lateral teeth), 100 (subequal number of denticles apically than at mid-
crown portion of distal carinae on lateral teeth), 103 (interdenticular space between mid-crown
denticles on distal carinae of lateral teeth broad), and 105 (interdenticular sulci between mid-
crown denticles on distal carinae of lateral teeth present, long, and well-developed)(Hendrickx
and Mateus, 2014). The clade comprised of NJSM GP 14256 and other de^{scribed} tyrannosaurs
(*Alioramus*, *Tyrannosaurus*, “*Raptorex*”) is united by the presence of a sub-symmetric crown
with a centrally-positioned distal carina in distal view (char. 83). Characters uniting the
tyrannosauroid clade include 3, 5, 19, 27, 37, 38, 41, and 48 in the list of Hendrickx and Mateus
(2014). The strict consensus tree (tree length = 688, consistency index = 0.340, retention index =
0.561) is shown in Fig. 1B.

Des^{cription}.

NJSM GP 14256 (Fig. 1) is the apical half of the tooth of a theropod dinosaur.
Measurements of the specimen may be found in Table 1. The tooth is well-preserved for a
terrestrial fossil collected from one of the marine deposits of the Cretaceous Atlantic Coastal
Plain, preserving details of the outer enamel layer and denticle morphology. Unfortunately, the
basal half of the crown and the entirety of the root of the tooth are not preserved. This is

probably due to erosion, as the tooth is broken transversely and heavily rounded at its preserved
 base (Fig. 1E—F).

The tooth displays the ziphodont condition in being labiolingually compressed and only
 slightly recurved. The preserved mesial carina is slightly convex, whereas the distal carina is
 vertical along its entire ~~run~~. The labial and lingual portions of the enamel are well-preserved
 (Fig. 1G), bearing transverse undulations that develop out of the distal margin of the tooth to
 become bands (Brusatte et al. 2007). In NJSM GP 14256, these undulations (= marginal bands)
 are relatively strongly developed, although they are less prominent than in carcharodontosaurids
 (e.g., Sereno et al., 1996; Brusatte et al., 2007). The labial and lingual surfaces of the tooth are
 slightly convex, as in most other theropod dinosaurs (Hendrickx et al., 2015a). The apex of the
 tooth bears a slight wear facet on its lingual surface. The tooth is lenticular in basal cross-section.

The distal carina preserves many denticles (Fig. 1E—F, H), which are small, dense
 (6/mm), and apicobasally straightened. The denticles are interspersed with diminutive
 interdenticular sulci (Currie et al., 1990). These are encompassed by the apical ends of the
 denticles. These denticles maintain a similar density along

Table 1. Measurements of teeth described in this study (in/per mm).

Specimen	CH	CBL	CBW	AL	CA	DB	DC	DA
NJSM GP 12456	15 (est. 25)	8.99	4.90	N/A	N/A	2	2	2.5
NJSM GP 22949	15.5	8.0	2.0	18.00	55.5	6	5	5

the entirety of the distal carina. However, their density may have changed along the missing
 portion of the tooth. The mesial carina preserves a few denticles, although these are too eroded

for much morphological description. These denticles appear to be similar in size to those on the
distal carina.

~~Dromaeosaurid tooth.~~

Dinosauria Owen 1842

Theropoda Marsh 1881

Coelurosauria von Huene 1921

Maniraptora Gauthier 1986

Dromaeosauridae Matthew & Brown 1922

Saurornitholestinae Sues 1978

~~cf. Saurornitholestinae indet.~~

Material.

NJSM GP 22949, well-preserved, complete isolated tooth.

Locality and Horizon.

212 Mt. Laurel Formation sediments in Burlington County, New Jersey, latest Campanian to early

Maastrichtian (Gallagher, 1993; Miller et al., 2004; Gallagher et al., 2014).

Identification.

NJSM GP 22949 is identified as the lateral tooth of a dromaeosaurid theropod based on

the following combination of features: (1) its extreme apicobasal curvature created by its

concave distal carina and distally offset apex, (2) the presence of apically hooked distal denticles,

(3) the absence of mesiodistal constriction along the crown base, and (4) distal denticles that

decrease in size towards the apex of the tooth (Fig. 3A—E)(Currie et al., 1990; Turner et al.,

2012; Larson and Currie, 2013; Williamson and Brusatte, 2014). NJSM GP 22949 is smaller

than the majority of Appalachian theropod teeth assigned to tyrannosauroids, in which crown

heights surpass 50 mm (Schwimmer et al., 1993; Carr et al., 2005; Brusatte et al., 2011; Denton
et al., 2011; Schwimmer et al., 2015). However, the tooth is notably larger than most North
American dromaeosaurid teeth, which are often less than 10 mm in height and mostly measure
around 5 mm in that dimension (Ostrom, 1969; Currie et al., 1990; Currie and Varricchio, 2004;
Larson and Currie, 2013; Williamson and Brusatte, 2014; Wick et al., 2015). *Dakotaraptor*,
which possessed crowns up to ~25 mm high, represents the exception among Maastrichtian
dromaeosaurids, the teeth of which usually are below 10 mm in height (Larson and Currie, 2013;
DePalma et al., 2015). Instead, the Mt. Laurel dromaeosaurid tooth is more comparable to the
teeth of the 3—4 meter *Deinonychus* and an indeterminate specimen from the Tar Heel
Formation of North Carolina in its dimensions (Ostrom, 1969; Brownstein, 2018b). NJSMP GP
22949 is also distinguished from Appalachian tyrannosauroids in lacking subquadrangular
distocentral denticles (Hendrickx et al., 2015). The first discriminant analysis on the Mt. Laurel
dromaeosaurid tooth supports this hypothesis by placing the specimen within the convex hull
formed by the teeth of *Velociraptor* and not in the convex hull formed by the teeth of
tyrannosaurs or troodontids (Fig. 4A).

NJSMP GP 22949 is notable for being similar to the teeth of western North American
saurornitholestine dromaeosaurids (Fig. 3A—E, G)(Larson and Currie, 2013) and somewhat
unlike those previously discovered from the American east (Kiernan and Schwimmer, 2004;
Schwimmer et al., 2015; Brownstein, 2018a). Teeth assigned to *Saurornitholestes* have been
described from the Cretaceous of the southeastern United States (Kiernan and Schwimmer, 2004;
Schwimmer et al., 2015). These teeth are extremely small (< 6 mm), far less recurved than NJSMP
GP 22949, and have proportionally large denticles that are more strongly apically hooked (e.g.,
fig. 1 in Kiernan and Schwimmer, 2004). One tooth from Aluma measuring 4.9 mm in crown

height and preserving 7 distal denticles and 8 mesial denticles per mm is less recurved than
NJSM GP 22949 and far less elongate in labial and lingual views (see Kiernan and Schwimmer,
2004). Some saurornitholestine teeth from South Carolina (Schwimmer et al., 2015) also lack the
‘slender’ condition in NJSM GP 22949, where the mesiodistal width of the heavily recurved
tooth crown is much smaller than the crown height. Teeth from the Ellisdale site of New Jersey
include dromaeosaurid crowns (Denton et al., 2011; Brownstein, 2018c). However, large crowns
from Ellisdale are not ‘slender’ like NJSM GP 22949, are larger in size, and bear distal denticles
that are considerably more apically hooked than the Mt. Laurel tooth and western
saurornitholestines (Brownstein, 2018c). A dromaeosaurid tooth from North Carolina is slightly
larger than NJSM GP 22949, but is less recurved and far less slender (Brownstein, 2018b).

When compared to dromaeosaurid teeth from outside eastern North America, NJSM GP
22949 most closely resembles the teeth of western North American saurornitholestine
dromaeosaurids. Despite the fact that NJSM GP 22949 was placed in the convex hull formed by
the teeth of *Velociraptor* in the first discriminant analysis conducted, the tooth is unlike those of
the western North American velociraptorine *Acheroraptor temertyorum* (Evans et al., 2013) or
the Mongolian *Velociraptor mongoliensis* (Turner et al., 2012; Fig. 3F) in lacking strongly
developed striations along its crown surface, in being more strongly recurved, and in being far
more slender (lower CBL/CH value)(Fig. 2A—B, F)(Smith et al., 2005; Evans et al., 2013). In
contrast to *Dromaeosaurus albertensis*, and ‘dromaeosaurine’ teeth from western North
America, the mesial carina in NJSM GP 22949 is not twisted onto the mesiolingual face of the
crown, the distal denticles are apically hooked, and the tooth is more strongly recurved (Fig.
3H—I; Turner et al., 2012; Larson and Currie, 2013). The Mt. Laurel tooth is far less robust and
has far less developed carinae than the teeth of *Utahraptor* (Fig. 2H). The Mt. Laurel tooth is

also smaller, much more strongly recurved, and possesses denticles more apically hooked than
those of the giant Maastrichtian dromaeosaurid *Dakotaraptor steini* (DePalma et al., 2015).
NJSM GP 22949 lacks the ‘figure-8’ basal cross-section seen in the teeth of *Deinonychus*
(Ostrom, 1969; Brownstein, 2018b; pers. obs.). Although the strongly recurved maxillary teeth
of *Deinonychus* (Ostrom, 1969; Turner et al., 2012; pers. obs.) are somewhat comparable with
NJSM GP 22949, the differing basal cross-sections among these specimens and the slightly
asymmetrical morphology of the teeth in *Deinonychus* distinguish *D. antirrhopus* and the Mt.
Laurel form. The discriminant analysis of the Larson and Currie (2013) dataset supports
saurornitholestine affinities for NJSM GP 22949, classifying the tooth as a saurornitholestine
crown (Supplementary Information).

NJSM GP 22949 resembles the teeth of western North American Maastrichtian
saurornitholestines in having a slender, tall outline in labial and lingual views (the “Lancian”
saurornitholestine morphotype of Larson and Currie, 2013). The tooth is closely comparable with
the crowns of the juvenile saurornitholestine ‘*Bambiraptor feinbergi*,’ which are extremely
recurved and slender and possess apically-hooked denticles (Fig. 3G). A discriminant analysis of
the Late Cretaceous western North American paravian tooth dataset of Larson and Currie (2013)
found NJSM GP 22949 to nest within the convex hulls formed by four tooth morphotypes
(Saurornitholestinae, Dromaeosaurinae, *Zapsalis*, and *Atrociraptor*). NJSM GP 22949 is also
quantifiably unlike YPM VPPU.021397, the large dromaeosaurid tooth from the Campanian of
North Carolina (Brownstein, 2018b), plotting far from the southeastern North American
specimen in both morphometric analysis in which these teeth were included (Fig. 4). Thus,
NJSM GP 22949 is most comparable to the crowns of a saurornitholestine-like dromaeosaurid.
Saurornitholestines are small-bodied dromaeosaurids (Turner et al., 2012), and so NJSM GP

22949 is important for indicating members of this group may have achieved relatively large body
sizes for dromaeosaurs.

Descript
NJSJ GP 22949 is the complete crown of a dromaeosaurid dinosaur. Measurements of
this specimen are in Table 1. This tooth is heavily recurved, displaying the zipodont condition.
The crown possesses an ovoid basal cross-section. In distal view, the middle portion of tooth is
convex labially, although the crown becomes labiolingually straightened towards its apex. The
labial and lingual surfaces are flattened, and the lack of a root attached to this crown indicates it
was shed. Although both the mesial and distal carinae are preserved, the mesial denticles have
been mostly eroded away, and precise denticle counts for the mesial carina are unable to be
taken. Some portions of the tooth crown are cracked, and the outer enamel layer is poorly
preserved towards the distal end of the specimen. Small portions of the middle of the crown are
missing. The distal profile of NJSJ GP 22949 is strongly concave. The preserved portions of the
outer enamel layer are smooth, although at the apex several slightly developed ridges appear.
These ridges could represent features of the original morphology of the tooth or be damage from
feeding or taphonomic processes. The distal carina preserves a large number of apically hooked
denticles that become smaller towards the apex of the crown. These denticles are separated by
interdenticular sulci that, along with the serrations, project slightly onto the tooth surface.
Unfortunately, the shape and density of the mesial denticles could not be determined, as the
mesial carina is heavily eroded in NJSJ GP 22949.

**Discussion.**

The two theropod teeth described here add to one of the most complete Maastrichtian
faunas from eastern North America. The dromaeosaurid tooth NJSJ GP 22949 is

biogeographically significance for being the first occurrence of this clade in the Mount Laurel
Formation and more generally Maastrichtian of eastern North America. Until now,
tyrannosauroids and ornithomimosaurids were the only known theropods from the late Campanian-
Maastrichtian of this area (Gallagher, 1993; Weishampel & Young, 1996; Gallagher, 1997;
Brusatte et al., 2011; Brusatte et al., 2012; Gallagher et al., 2014), with the latest records of
dromaeosaurids in the American east hailing from mid-Campanian units in the Carolinas
(Schwimmer et al., 2015; Brownstein, 2018b) and the Ellisdale site of New Jersey (Denton et al.,
2011). Although NJSM GP 22949 is most comparable to the crowns of mid-sized to largish
dromaeosaurids like *Deinonychus antirrhopus* and *Dakotaraptor steini* (Ostrom, 1969; DePalma
et al., 2015) and to small tyrannosauroids (Denton et al., 2011; Williamson and Brusatte, 2014;
Schwimmer et al., 2015) in its dimensions, it is most closely allied with dromaeosaurids in the
morphometric analyses conducted (Fig. 4) and in many key features of its morphology.

The tyrannosauroid tooth NJSM GP 14256 supports the presence of *Dryptosaurus*-like
tyrannosauroids in the early Maastrichtian of New Jersey. Isolated teeth and postcranial material
from the Mount Laurel were previously assigned to *Dryptosaurus* sp. based on little more than
their geographic proximity to the site where the holotype of this taxon was recovered
(Gallagher, 1993; Weishampel & Young, 1996; Gallagher, 1997), but no detailed description of
late Campanian to early Maastrichtian tyrannosauroids from New Jersey has appeared in the
literature. The well-preserved nature of NJSM GP 14256 thus allows for the formal recognition
of the presence of non-tyrannosauroid tyrannosauroids in the latest Campanian to earliest
Maastrichtian of the Atlantic Coastal Plain. Furthermore, the excellent condition of the outer
enamel layer of NJSM GP 12456 allows for further documentation of the dental anatomy of
Appalachian tyrannosauroids, the isolated teeth of which are often found highly abraded among

stream deposits (e.g., Weishampel & Young, 1996). The presence of non-tyrannosaurid
tyrannosauroids in the Mt. Laurel Formation is expected given the presence of *Dryptosaurus*
*aquilunguis* and non-tyrannosauroid tyrannosaurs of similar phylogenetic position in both the
middle-late Maastrichtian Navesink and New Egypt formations (Gallagher, 1993; Brusatte et al.,
2011) and early Campanian Marshalltown Formation (Denton et al., 2011). However, that NJSM
GP 14256, originally discovered in 1984, is only described now attests to the understudied nature
of these deposits.

The late recognition of dromaeosaurids in the Maastrichtian sediments of New Jersey is
notable, given that the dinosaurs of the Mt. Laurel and other Cretaceous units in the Atlantic
Coastal Plain have been studied for over a century and a half (e.g., Leidy, 1858; Cope, 1866;
Gallagher, 1993; Weishampel & Young, 1996; Gallagher, 1997). Teeth from locations like the
Ellisdale site of the Marshalltown Formation of New Jersey originally assigned to tyrannosaurs
have more recently been reclassified as the crowns of dromaeosaurids (Gallagher et al., 1986;
Gallagher, 1993, 1997; Denton et al., 2011), so it is entirely possible that the lack of diversity in
dinosaur faunas from the Maastrichtian of the Atlantic Coastal Plain reflects systematic
misidentification of these isolated fossils. Only further work on Appalachian fossils will allow
for more comprehensive revision of the identification of fossils from this area.

During the Cretaceous, terrestrial faunas became more regionalized as the breakup of
supercontinents like Gondwana and Laurasia and the inundation of smaller landmasses like
North America and Europe occurred (e.g., Russell, 1995; Sereno et al., 1996; Csiki et al., 2010;
Sampson et al., 2010; Csiki-Sava et al., 2015). In particular, the faunas of Appalachia and
Laramidia, which became isolated from each other as the Western Interior Seaway flooded the
middle of North America (e.g., Russell, 1995; Roberts and Kirschbaum, 1997; Schwimmer,

2002; Sampson et al., 2010), have been recognized as highly distinctive (e.g., Gallagher, 1993;
Weishampel and Young, 1996; Gallagher, 1997; Sampson et al., 2010; Denton et al., 2011;
Schwimmer et al., 2015). In the past thirty years, a handful of discoveries from the eastern
margin of North America have indicated some faunal interchange occurred between Laramidian
and Appalachian dinosaur communities during the latest Maastrichtian. These include the tooth
of a ceratopsid from the latest Maastrichtian of Mississippi and the fragmentary forelimb
material of possible lambeosaurines from the Maastrichtian of New Jersey and earliest
Maastrichtian of Nunavut, Canada (Gallagher, 1995; Gallagher, 1997; Farke and Phillips, 2017).
At the same time, Appalachian faunas continued to harbor endemic forms like intermediate
tyrannosauroids, represented by *Dryptosaurus aquilunguis* and comparable forms through the
Campanian-Maastrichtian transition in the Atlantic Coastal Plain (Gallagher, 1993; Brusatte et
al., 2011; this paper). ~~Faunal interchange in the last 10 million years of the Mesozoic seems to
have occurred throughout the northern hemisphere. Phylogenetic evidence strongly posits that
the presence of *Tyrannosaurus rex* in the Maastrichtian of the western United States and Canada
represents a dispersal of Asian tyrannosaurids into the Americas (Brusatte et al., 2010; Brusatte
and Carr, 2016). A dispersal event between Asian and North American faunas during the
Maastrichtian may have occurred over Beringia (Fiorillo, 2008), given the similarity of
Maastrichtian polar faunas from Russia, Alaska, and lower latitudes in the United States and
Canada (e.g., Weishampel et al., 2004; Gangloff, 2012; Godefroit et al., 2012; Le Locuff, 2012).~~
Along with previous discoveries, the tyrannosauroid tooth described here supports the ‘refugium’
model for eastern North America, wherein taxa more closely allied with middle Cretaceous
forms (e.g., non-tyrannosaurid tyrannosauroids like *Dryptosaurus* and *Appalachiosaurus*)(Carr et

al., 2005; Brusatte et al., 2011; Brusatte and Carr, 2016) persisted in relative isolation as more
deriv  forms evolved in Laurasia.

**Conclusions.**

Two theropod teeth from the Campanian-Maastrichtian Mount Laurel Formation of New
Jersey are described in detail. The dromaeosaurid tooth, which plots with western North
American saurornitholestine teeth in principle components and discriminant analyses, is the
youngest record of a non-avian maniraptoran from eastern North America and the first from the
latest Campanian-Maastrichtian of the American east. This tooth provides another record of a
mid-sized to large dromaeosaurid in the Cretaceous of eastern North America. However, this
tooth is more allied with those of saurornitholestines and velociraptorines than with
*Deinonychus*, dromaeosaurines, or largish dromaeosaurid teeth previously described from
Appalachia, tentatively suggesting that several types of dromaeosaurids might have grown to
relatively large sizes in the Cretaceous of the eastern United States and indicating mid-sized to
largish dromaeosaurids were a usual component of Appalachian faunas. The tyrannosauroid
tooth is the first specimen to suggest the presence of *Dryptosaurus aquilunguis* or a closely
related tyrannosauroid in the Mt. Laurel ecosystem, further supporting the refugium model for
Appalachian vertebrate evolution.

**Methods.**

Measurements and nomenclature.

Measurements of both teeth were taken in accordance with the methodology of Smith et al.
(2005) and Larson and Currie (2013). The dimensions of the Mt. Laurel teeth were determined
using digital calipers. I follow the nomenclature of Hendrickx et al. (2015) when describing the
two teeth on which this paper focuses.

Principle components analyses.

In order to provide support for the assignment of the dromaeosaurid tooth to a specific
group of theropod dinosaurs, I included it in principle components and discriminant analyses
conducted in the program PAST v. 3.18 (Hammer et al., 2001). In order to assess the
morphological similarity of the Mt. Laurel teeth to theropod clades present in the Cretaceous of
the northern hemisphere, I used a modified version (Brownstein, 2018b) of the dataset of Smith
et al. (2005) that includes tooth data on tyrannosauroids, troodontids, and dromaeosaurids. A
principle components analysis was run on this dataset, which included data on fifteen
measurements: crown height (CH), crown base length (CBL), crown base width (CBW),
apicobasal length (length of the tooth along the longest apicobasal axis), and serration density
415 per 5mm for the basal (DB), mid-crown (DC), and apical (DA) distal carina. An additional
416 principle components analysis (PCA) was conducted using the dataset of Larson and Currie
(2013) in order to better assess the similarity of the Mt. Laurel dromaeosaurid specimen to other
North American paravian teeth. This principle components analysis assessed for five
measurements: CH, CBL, CBW, and the mesial (MD) and distal (DD) denticles per millimeter.
The summary statistics and loadings from the results of the PCAs conducted are included in the
Supplementary Information.

Discriminant analyses.

To further assess the affinities of the Mt. Laurel dromaeosaurid tooth, I performed a discriminant
analysis on the tooth datasets of Smith et al. (2005, modified in Brownstein, 2018b) and Larson
and Currie (2013). This analysis creates a morphospace by maximally separating objects sorted
into pre-determined groups. This analysis was also run in PAST v. 3.18 (Hammer et al., 2001),
and the loadings and confusion matrices can be found in the supplementary information.

Phylogenetic analysis.

To provide additional support for the referral of the incomplete tyrannosauroid tooth to that
family, I coded the specimen for the phylogenetic matrix of Hendrickx and Mateus (2014), a
dataset of theropod dentition that includes 64 taxa/specimens coded for 141 characters. The
matrix was entered into the phylogenetics program TNT 1.5 (Goloboff and Catalano, 2016) for a
phylogenetic analysis. The matrix was first analyzed using the “New Technology Search,” with
default parameters for ratchet, tree drift, tree fuse, and sectorial search. A total of 10 trees of
length 688 were retained. These topologies were then subjected to traditional (TBR) branch
swapping, which allows for a more extensive exploration of each tree island. This found
over >99,999 most parsimonious topologies of 60. These were summarized in a strict consensus
topology.

**Acknowledgements.**

I thank Dana Ehret for access to the collections of the New Jersey State Museum, Carl Mehling
for access to the collections of the American Museum of Natural History, and Dan Brinkman for
access to the collections of the Yale Peabody Museum of Natural History. I thank Howard
Falcon-Lang, Christophe Hendrickx, Terry Gates, Kevin Padian, and four anonymous reviewers
for their comments, which greatly improved the quality of this paper.

**Availability of data and material.**

All data is available in the Supplementary Information.

**Competing interests.**

The author declares no competing interests.

**Funding.**

The author received no funding for this work.

**References.**

Alvarez, L.W., Alvarez, W., Asaro, F., Michel, H.V. 1980. Extraterrestrial cause for the
Cretaceous-Tertiary extinction. *Science* **208**:1095–1108.

Brochu, C.A. 2003. Osteology of *Tyrannosaurus rex*: insights from a nearly complete skeleton
and high-resolution computed tomographic analysis of the skull. *Society of Vertebrate
Paleontology Memoir* **7**:1–138.

Brownstein, C.D. 2018a. The biogeography and ecology of the Cretaceous non-avian dinosaurs
of Appalachia. *Palaeontologia Electronica* **21.1.5A**:1–56.

Brownstein, C.D. 2018b. A large dromaeosaurid from North Carolina. *Cretaceous Research*
**92**:1–7.

Brownstein, C.D. 2018c. The distinctive theropod assemblage of the Ellisdale site of New Jersey
and its implications for North American dinosaur ecology and evolution during the Cretaceous.
*Journal of Paleontology* **92**(5):1115—1129.

Brusatte, S.L., Benson, R.B., Norell, M.A. 2011. The Anatomy of *Dryptosaurus aquilunguis*
(Dinosauria: Theropoda) and a Review of its Tyrannosauroid Affinities. *American Museum
Novitates* **3717**:1–53.

Brusatte, S.L., Benson, R.B.J., Carr, T.D., Williamson, T.E., and Sereno, P.C. 2007. The
systematic utility of theropod enamel wrinkles. *Journal of Vertebrate Paleontology* **27**: 1052-
1056.
Brusatte, S.L., Butler, R.J., Barrett, P.M., Carrano, M.T., Evans, D.C., Lloyd, G.T., Mannion,
P.D., Norell, M.A., Peppe, D.J., Upchurch, P., and Williamson, T. E. 2015. The extinction of the
dinosaurs. *Biological Reviews* **90**:628–642.
Brusatte, S.L., Butler, R.J., Prieto-Márquez, A., Norell, M.A. 2012. Dinosaur morphological
diversity and the end-Cretaceous extinction. *Nature Communications* **3**:804.
Brusatte, S.L., Carr, T.D. 2016. The phylogeny and evolutionary history of tyrannosauroid
dinosaurs. *Scientific Reports* **6**(1):20252.
Brusatte, S.L., Norell, M.A., Carr T.D., Erickson, G.M., Hutchinson, J.R., Balanoff, A.M.,
Bever, G.S., Choiniere, J.N., Makovicky, P.J., Xu X. 2010. Tyrannosaur paleobiology: new
research on ancient exemplar organisms. *Science* **329**(5998):1481-1485.
Burns, M.E., and Ebersole, J.A. 2016. Juvenile Appalachian nodosaur material (Nodosauridae,
Ankylosauridae) from the lower Campanian lower Mooreville Chalk of Alabama. Abstracts and
Posters Session IV, Society of Vertebrate Paleontology, 76th Annual Meeting.

Carr, T.D., Williamson, T.E., Schwimmer, D.R. 2005. A new genus and species of
tyrannosauroid from the Late Cretaceous (middle Campanian) Demopolis Formation of
Alabama. *Journal of Vertebrate Paleontology* **25**(1):119–143.
Cope, E.D. 1866. Discovery of a gigantic dinosaur in the Cretaceous of New Jersey. *Proceedings*
*of the Academy of Natural Sciences of Philadelphia* **18**: 275–279.
Csiki-Sava, Z., Buffetaut, E., Ósi, A., Pereda-Suberbiola, X., and Brusatte, S.L. 2015. Island life
in the Cretaceous - Faunal composition, biogeography, evolution, and extinction of land-living
vertebrates on the Late Cretaceous European archipelago. *ZooKeys* **469**:1–161.
Csiki, Z., Vremir, M., Brusatte, S.L., Norell, M. A. 2010. An aberrant island-dwelling theropod
dinosaur from the Late Cretaceous of Romania. *Proceedings of the National Academy of*
*Sciences* **107**(35):15357–15361.
Currie P.J., Rigby J.K., Sloan R.E., 1990, Theropod teeth from the Judith River Formation of
Southern Alberta, Canada. In: Carpenter, K., and Currie, P.J., eds: *Dinosaur Systematics:*
*Approaches and Perspectives*. Cambridge University Press, p. 107–125.
Currie, P. J. and Varricchio, D.J. 2004. A new dromaeosaurid from the Horseshoe Canyon
Formation (Upper Cretaceous) of Alberta, Canada. In: Currie, P.J., Koppelhus, E.B., Shugar,
517 M.A., and Wright, J.L., eds: *Feathered Dragons*. Indianapolis: Indiana University Press. p. 112–
518 132.

Denton R.K., O'Neill R.C., Grandstaff B.S., and Parris D.C. 2011. The Ellisdale Site (Late
Cretaceous, Campanian) - is there a rationale for an "Ellisdalean" land faunal age? *Geological*
*Society of America Abstracts with Programs* **43**:85.
DePalma R.A., Burnham D.A., Martin L.D., Larson P.L., and Bakker R.T. 2015. The First Giant
Raptor (Theropoda: Dromaeosauridae) from the Hell Creek Formation. *Paleontological*
*Contributions* **14**:1–16.
Farke, A.A. and Phillips, G.E. 2017. The first reported ceratopsid dinosaur from eastern North
America (Owl Creek Formation, Upper Cretaceous, Mississippi, USA). *PeerJ* **5**:e3342.
Fiorillo, A.R. 2008. Cretaceous dinosaurs of Alaska: Implications for the origins of Beringia. In:
Blodgett, R.B., and Stanley, G. eds: The Terrane Puzzle: new perspectives on paleontology and
stratigraphy from the North American Cordillera. *Geological Society of America Special Paper*
**442**: 313-326.
Frederickson, J., Engel, M.H., Cifelli, R. 2018. Niche Partitioning in Theropod Dinosaurs: Diet
and Habitat Preference in Predators from the Uppermost Cedar Mountain Formation (Utah,
U.S.A.). *Scientific Reports* **8**(1):17872.
Gallagher, W.B. 1993. The Cretaceous-Tertiary mass extinction event in North Atlantic Coastal
Plain. *The Mosasaur* **5**:75-154.

Gallagher, W.B. 1995. Evidence of juvenile dinosaurs and dinosaurian growth stages in the Late
Cretaceous deposits of the Atlantic Coastal Plain. *Bulletin of the New Jersey Academy of Science*
**40**:5–8.
Gallagher, W.B. 1997. *When Dinosaurs Roamed New Jersey*. Rutgers University Press, New
Brunswick, New Jersey.
Gallagher, W.B., Camburn, J., Camburn, S., and Hanczaryk, P.A. 2014. Taphonomy of a Late
Campanian Fossil Assemblage at Marlboro, Monmouth County, New Jersey. *The Mosasaur*
**8**:53–68.
Gangloff, R.A. *Dinosaurs under the Aurora*. Bloomington, Indiana University Press, 2012.
Gauthier, J. 1986. Saurischian monophyly and the origin of birds. *Memoirs of the California*
*Academy of Sciences* **8**:1–55.
Godefroit, P., Bolotsky, Y.L., Lauters, P. 2012. A New Saurolophine Dinosaur from the Latest
Cretaceous of Far Eastern Russia. *PLoS ONE* **7**(5):e36849.
Goloboff, P., Catalano, S., 2016. TNT version 1.5, including full implementation of phylogenetic
morphometrics. *Cladistics* **32**(3): 221-238.

Hammer, Ø., Harper, D.A.T., Ryan, P. D. 2001. Paleontological statistics software package for
education and data analysis. *Palaeontologia Electronica* **4**:1–9.
Hendrickx, C., Mateus, O., 2014. Abelisauridae (Dinosauria: Theropoda) from the Late Jurassic
of Portugal and dentition-based phylogeny as a contribution for the identification of isolated
theropod teeth. *Zootaxa* **3759**(1):1–74.
Hendrickx, C., Mateus, O., Araújo, R., 2015. A proposed terminology of theropod teeth
(Dinosauria, Saurischia). *Journal of Vertebrate Paleontology* **35**(5):e982797.
Kiernan, K. and Schwimmer, D.R. 2004. First record of a velociraptorine theropod (Tetanurae,
Dromaeosauridae) from the eastern Gulf Coastal United States. *The Mosasaur* **7**:89–93.
Krumenacker, L.J., Simon, D.J., Scofield, G., and Varricchio, D.J. 2017. Theropod dinosaurs
from the Albian-Cenomanian Wayan Formation of eastern Idaho. *Historical Biology* **29**(2): 170-
186.
Langston, W. 1960. The vertebrate fauna of the Selma Formation of Alabama, part VI: the
dinosaurs. *Fieldiana: Geological Memoirs* **3**(5):315-359.
Larson, D.W., Currie, P.J. 2013. Multivariate Analyses of Small Theropod Dinosaur Teeth and
Implications for Paleoeological Turnover through Time. *PLoS ONE* **8**(1): e54329.

Le Loeuff, J. 2012. Paleobiogeography and biodiversity of Late Maastrichtian dinosaurs: how
many dinosaur species went extinct at the Cretaceous-Tertiary boundary? *Bulletin de la Société*
*Géologique de France* **183**:547–559.
Leidy J. 1858. Hadrosaurus foulkii, a new saurian from the Cretaceous of New Jersey, related
to Iguanodon. *Proceedings of the Academy of Natural Sciences* **10**:213-218.
Marsh, O.C. 1881. Principal characters of American Jurassic dinosaurs. Part V. *American*
*Journal of Sciences series 3* **21**:417–423.
Matthew W.D., and Brown, B. 1922. The family Deinodontidae, with notice of a new genus
from the Cretaceous of Alberta. *Bulletin of the American Museum of Natural History* **46**:367–
385.
Miller, H.W. 1967. Cretaceous vertebrates from Phoebus Landing, North Carolina. *Proceedings*
*of the Academy of Natural Sciences* **119**:219–239.
Miller, K.G., Sugarman, P.J., Browning, J.V., Kominz, M.A., Olsson, R.K., Feigenson, M.D.,
and Hernandez, J.C. 2004. Upper Cretaceous sequences and sea-level history, New Jersey
Coastal Plain. *Bulletin of the Geological Society of America* **116**(3):368–393.
Ostrom, J.H. 1969. Osteology of Deinonychus antirrhopus, an unusual theropod from the Lower
Cretaceous of Montana. *Peabody Museum of Natural History Bulletin* **30**:1–165.

Owen, R. 1842. Report on British Fossil Reptiles, Pt. II. *Report of the British Association for the*
*Advancement of Science* **11**:60–204.
Prieto-Márquez, A., Erickson, G.M., and Ebersole, J.A. 2016. A primitive hadrosaurid from
southeastern North America and the origin and early evolution of 'duck-billed' dinosaurs. *Journal*
*of Vertebrate Paleontology* **36**(2):e1054495.
Prieto-Márquez, A., Weishampel, D.B., and Horner, J.R. 2006. The dinosaur *Hadrosaurus*
*foulkii*, from the Campanian of the East Coast of North America, with a reevaluation of the
genus. *Acta Palaeontologica Polonica* **51**(1):77-98.
Roberts, L.N.R. and Kirschbaum, M.A. 1995. Paleogeography of the Late Cretaceous of the
Western Interior of middle North America-Coal distribution and sediment accumulation. *U.S.*
*Geological Survey Professional Paper* **1561**:1–115.
Russell, D.A. 1995. China and the lost worlds of the dinosaurian era. *Historical Biology* **10**(1):3–
12.
Sakamoto, M., Benton, M.J., and Venditti, C. 2016. Dinosaurs in decline tens of millions of
631 years before their final extinction. *Proceedings of the National Academy of Sciences* **113**:5036–
632 5040.

Samman T., Powell G.L., Currie P.J., and Hills, L.V. 2005. Morphometry of the teeth of western
North American tyrannosaurids and its applicability to quantitative classification. *Acta*
*Palaeontologica Polonica* **50**:757–776.
Sampson, S.D., Loewen, M.A., Farke, A.A., Roberts, E.M., Forster, C.A., Smith, J.A., and Titus,
639 A.L. 2010. New horned dinosaurs from Utah provide evidence for intracontinental dinosaur
endemism. *PLoS ONE* **5**(9):e12292.
Sarjeant, W.A.S., Currie, P.J. 2001. The “Great Extinction” that never happened: The demise of
the dinosaurs considered. *Canadian Journal of Earth Sciences* **38**(2):239–247.
Schwimmer, D.R., Sanders, A.E., Erickson, B.R., and Weems, R.E. 2015. A Late Cretaceous
dinosaur and reptile assemblage from South Carolina, USA. *Transactions of the American*
*Philosophical Society* **105**(2):1–157.
Schwimmer, D.R., Williams, G.D., Dobie, J.L., and Siesser, W.G. 1993. Late Cretaceous
dinosaurs from the Blufftown Formation in western Georgia and eastern Alabama. *Journal of*
*Vertebrate Paleontology* **67**(2):288–296.
Sereno, P.C., Dutheil, D.B., Iarochene, M., Larsson, H.C.E., Lyon, G.H., Magwene, P.M., Sidor,
C.A., Varricchio, D.J., Wilson, J.A. Predatory dinosaurs from the Sahara and Late Cretaceous
faunal differentiation. *Science* **272**:986–991.

Sloan, R.E., Rigby, J.K. Jr, Van Valen, L.M., Gabriel, D. 1986. Gradual dinosaur extinction and
simultaneous ungulate radiation in the hell creek formation. *Science* **232**(4750):629–633.
Smith, J.B. 2005. Heterodonty in *Tyrannosaurus rex*: Implications for the taxonomic and
systematic utility of theropod dentitions. *Journal of Vertebrate Paleontology* **25**:865–887.
Smith, J.B., Vann, D.R., Dodson, P. 2005. Dental morphology and variation in theropod
dinosaurs: implications for the taxonomic identification of isolated teeth. *The Anatomical Record*
**285A**:699–736.
Sues, H.-D .1978. A new small theropod dinosaur from the Judith River Formation (Campanian)
of Alberta Canada. *Zoological Journal of the Linnean Society* **62**:381–400.
Sugarman, P.J., Miller, K.G., Burky, D., and Feigenson, M.D. 1995. Uppermost Campanian-
Maastrichtian strontium isotopic; biostratigraphic and sequence stratigraphic framework of the
New Jersey Coastal Plain. *Geological Society of America Bulletin* **107**:19-37.
Turner A.H., Makovicky P.J., and Norell M.A. 2012. A Review of Dromaeosaurid Systematics
and Paravian Phylogeny. *Bulletin of the American Museum of Natural History* **371**:1–206.
von Huene F. 1914. Saurischia et Ornithischia Triadica (“Dinosuaria” Triadica).
*Animalia. Fossilium Catalogus* **4**:1-21.

Weishampel, D.B. 2006. Another look at the dinosaurs of the East Coast of North America, p.
129-168. In 'Coletivo Arqueológico-Paleontológico Salense, (eds.), Actas III Jornadas
Dinosaurios Entorno. Salas de los Infantes, Burgos, Spain.
Weishampel, D.B. and Young, L. 1996. Dinosaurs of the East Coast. Johns Hopkins University
Press, Baltimore, Maryland, USA.
Weishampel, D.B., Barrett, P.M., Coria, R.A., Loeuff, J.L., Xing, X., Xijin, Z., Sahni, A.,
Gomani, E.M.P., and Noto, C.R. 2004. Dinosaur Distribution. In: Weishampel, D.B., Dodson, P.,
and Osmólska, H. eds: The Dinosauria, 2nd Edition. University of California Press, Berkeley,
California, USA. p. 517-617.
Wick, S.L., Lehman, T.M., Brink, A.A. 2015. A theropod tooth assemblage from the lower
Aguja Formation (early Campanian) of West Texas, and the roles of small theropod and varanoid
lizard mesopredators in a tropical predator guild. *Palaeogeography, Palaeoclimatology,*
*Palaeoecology* **418**:229.
Williamson T.E., and Brusatte S.L. 2014. Small Theropod Teeth from the Late Cretaceous of the
San Juan Basin, Northwestern New Mexico and Their Implications for Understanding Late
Cretaceous Dinosaur Evolution. *PLoS ONE* **9**:e93190.

A

A

B

C

Appendix E

Cover Letter

September 23, 2019

Dear editor of *Royal Society Open Science*,

I would like to resubmit to you my manuscript “New records of theropods from the latest Cretaceous of New Jersey and the Maastrichtian Appalachian fauna” for consideration at your journal.

In revising my work, I’ve paid special attention to the stylistic issues noted by the reviewers. Detailed responses may be found below. I hope the manuscript is now suitable for publication, and thank you for your reviews.

Regards,

Chase Brownstein
Research Associate,
Collections & Exhibitions, Stamford Museum & Nature Center
Stamford, Connecticut, United States

Responses to Reviewers:

Reviewer 1

First and foremost--the phylogenetic analysis of teeth. You must use the super matrix of Hendrickx and Mateo, not just the tooth matrix. Please recode and rerun the analysis. Also, you should add Dryptosaurus and potentially one other tyrannosaurid to the matrix so that you might get some resolution as to the phylogenetic affinities of your tooth. Without this there is little that can be said except it is a tyrannosauroid tooth.

Although I appreciate this comment, I want to note that the matrix of Hendrickx and Mateus I use in the paper was recommended by Dr. Hendrickx in a previous review of this manuscript. Also, beyond general phylogenetic placement, the conclusions to be drawn about the evolutionary relations of teeth from this type of analysis are really tenuous. The reviewer is correct in saying that little else besides the tooth's tyrannosauroid affinities can be deduced from this tree. Therefore, adding new taxa to the matrix wouldn't provide any strong new phylogenetic conclusions. As the other reviewers have not raised issues with this, I've chosen to keep the phylogeny as is.

Put some or all of the Discriminant Analysis results in the paper, not the SM. We need to see the data.

Done. I've made the relevant corrects.

The last sentence of the abstract does not make sense. I suggest deleting it, but you can try to modify it.

Done. I've deleted this sentence.

Reviewer: 3

Comments to the Author(s)

This revision of the original paper is much better present and the interpretations are supported. I think it is nearly ready for publication. I made minor corrections to the MS on the PDF. Please be careful with the use of the clade names. Tyrannosaurids = members of Tyrannosauridae. I noted these throughout. Also, put the description before the identification of each tooth section.

Done. I've made the suggested edits. One edit I did not choose to comply with was the switching of the Description and Identification sections. A previous reviewer had me order the paper that way, and switching them does not drastically change the manuscript or increase clarity.

Reviewer: 4

I suggest publication with minor revision; in addition to the specific comments below, I think the author should more clearly couch the article in terms of the refugium versus exchange models. In the introduction and conclusion, he leans heavily on the exchange model and that needs to be balanced, in both places, by a skeptical assessment of the case for the exchange model. An explicit statement of the evidence in favor of the refugium hypothesis must also be included in the abstract and the title must be adjusted accordingly.

Done. I've more clearly stated that these are the two models. I've also made the other suggested corrections.

The author may know my identity: Thomas D. Carr